# Cell cycle-driven transcriptome maturation confers multilineage competence to cardiopharyngeal progenitors

Yelena Y Bernadskaya[1,2,5], Ariel Kuan[1,5], Andreas Tjärnberg[1,5], Jonas Brandenburg [3], Ping Zhang[4], Keira Wiechecki[1], Nicole Kaplan[1], Margaux Failla [1,3], Maria Bikou[1], Oliver Madilian[1], Noah Bruderer [3], Wei Wang [1,4,5✉] & Lionel Christiaen [1,3✉]

## Abstract

During development, stem and progenitor cells divide and transition through multipotent states to generate the diverse cell types by undergoing defined changes in biomolecular composition, which underlie the progressive loss of potency and acquisition of lineage-specific characteristics. For example, the cardiac and pharyngeal muscle programs are jointly primed in multipotent cardiopharyngeal progenitors, and segregate in distinct daughter cells only after cell division. Here, using the tunicate Ciona, we showed that multipotent cardiopharyngeal progenitors acquire the competence to produce distinct *Tbx1/10* (+) and (−) daughter cells shortly before mitosis, which is necessary for *Tbx1/10* activation. By combining transgene-based sample barcoding with single-cell RNA-sequencing (scRNA-seq), we uncovered transcriptome-wide dynamics in migrating cardiopharyngeal progenitors as cells progress through G1, S, and G2 phases. We refer to this process as "transcriptome maturation", and identified candidate mature genes, including the Rho GAP-coding gene *Depdc1b*, which peaks in late G2. Functional assays indicated that transcriptome maturation fosters cardiopharyngeal competence, in part through multilineage priming and by enabling asymmetric cell division that influences subsequent fate decisions, illustrating the concept of "behavioral competence". We show that both classic regulatory circuits and coupling with the G1-S transition drive transcriptome maturation, ensuring the timely deployment of lineage-specific programs.

**Keywords** Cell Cycle; Gene Regulation; Multipotency; Heart; Oriented Cell Division
**Subject Categories** Chromatin, Transcription & Genomics; Development; Stem Cells & Regenerative Medicine

## Introduction

Complex animals are characterized by dozens to hundreds of distinct cell types that emerge during embryogenesis and post-embryonic development (Aviv et al, 2017; Cao et al, 2019b; Levy et al, 2021). During this process, the developmental potential of successive generations of pluri- and multipotent progenitor cells is progressively restricted, while they acquire the competence to produce a few differentiated cell types (Moris et al, 2016). Changes in the biomolecular composition of cells underlie these developmental transitions, and differential transcriptional activity governing transcriptome dynamics is an established driver of fateful molecular transitions during development (Levine and Tjian, 2003; Levine and Davidson, 2005).

While certain differentiated cells, such as neurons and striated muscles, are typically post-mitotic, multipotent progenitor cells must divide to express their full potential and produce a variety of cell lineages. The molecular machinery driving cell cycle progression and division is well characterized, and largely conserved across developmental and phylogenetic lineages (Harashima et al, 2013; Pagano, 2013; Nurse, 2000). How the cell cycle interfaces with fate choices during development remains debated and is likely to be highly variable. On one hand, there is evidence that fate choices can occur independently of cell cycle progression (Nair et al, 2013; Kukreja et al, 2024), to the point that so-called "cell cycle genes" are often *regressed* out of single-cell genomics analyses aimed at charting development decisions (Hao et al, 2024; Wolf et al, 2018). On the other hand, the G1 phase of the cell cycle tends to increase the propensity for mammalian stem cells to activate fateful determinants and engage along a certain developmental path (Pauklin and Vallier, 2013; Yiangou et al, 2019; Soufi and Dalton, 2016; Dalton, 2015), and fate choices are coupled with cell cycle progression during hematopoiesis (Passegué et al, 2005), while certain neuroblasts in *Drosophila* (Otsuki and Brand, 2018; El-Danaf et al, 2023; Doe, 2017), and early blastomeres in ascidian and *C. elegans* embryos appear to change fate with every division.

[1]Department of Biology, New York University, New York, NY, USA. [2]Department of Biology, College of Natural Sciences, University of Massachusetts - Amherst, Amherst, MA, USA. [3]Michael Sars Centre, University of Bergen, Bergen, Norway. [4]Ocean University of China, College of Marine Life Sciences, MoE Key Laboratory of Marine Genetics and Breeding, Fang Zongxi Center for Marine EvoDevo, 266003 Qingdao, China. [5]These authors contributed equally: Yelena Y Bernadskaya, Ariel Kuan, Andreas Tjärnberg, Wei Wang. ✉E-mail: ww8898@ouc.edu.cn; Lionel.Christiaen@uib.no

Conversely, cell fate choices have been shown to directly impact the cell cycle in a variety of organisms. For example, developmental regulation of the phosphatase Cdc25 contributes to coordinating mitotic patterns in the early fly embryo (Edgar et al, 1994; Edgar and O'Farrell, 1989; Di Talia and Wieschaus, 2012; Momen-Roknabadi et al, 2016), and differential beta-catenin activity, or *Cdc25* or *Cdkn1* expression accounts for blastomere and lineage-specific timing of cell division in ascidians (Ogura and Sasakura, 2016; Dumollard et al, 2013; Kobayashi et al, 2022).

The concept of multipotent cardiopharyngeal progenitors emerged as a compelling paradigm to account for the shared cardiac and craniofacial congenital defects observed in various conditions, such as the Di George/22q11 deletion syndrome, which is often caused by large deletions that remove a copy of the *TBX1* gene (Guner-Ataman et al, 2018; Cirino et al, 2020; Swedlund and Lescroart, 2020). This T-box transcription factor-coding gene is expressed early in progenitor cells for both the anterior second heart field and branchiomeric skeletal muscles, where it is required for proper heart and head muscle development (Rana et al, 2014; Nomaru et al, 2021; Adachi et al, 2020). Notably, tunicates of the ascidian genus *Ciona* possess a well-defined cardiopharyngeal lineage, which emerges from *Mesp+* mesodermal progenitors as is the case in mammals, and produces a *Gata4/5/6*-positive and *Tbx1*-negative first heart lineage, as well as the hallmark *Tbx1/10+* multipotent progenitors for the second heart and pharyngeal muscle lineages (Kaplan et al, 2015; Wang et al, 2019) (Fig. 1A). The ascidian cardiopharyngeal lineage develops in a highly stereotyped fashion, providing a single cell resolution view of the interplay between cardiopharyngeal multipotency, cell divisions and heart vs. pharyngeal muscle fate choices.

Here, we leveraged the unique features and experimental amenability of the *Ciona* embryo to study the acquisition of multilineage competence in cardiopharyngeal progenitors. We describe their transcriptome maturation, identifying a mature state that confers competence to divide in an oriented and asymmetric fashion, and produce distinct *Tbx1/10*(+) second multipotent cardiopharyngeal progenitors and *Tbx1/10*(−) first heart precursor cells. We characterized the regulation of multipotent progenitor maturation, and identified distinct levels of coupling between cell cycle progression and progenitor maturation and fate choices.

## Results

### Mitosis is necessary but not sufficient for *Tbx1/10* activation in cardiopharyngeal progenitors

The conserved cardiopharyngeal determinant *Tbx1/10* is activated after division of multipotent cardiopharyngeal progenitors (aka Trunk Ventral Cells, TVCs), following collective migration (Wang et al, 2013; Razy-Krajka et al, 2018). These oriented and unequal cleavages are coupled with asymmetric Fibroblast Growth Factor (FGF)- Microtubule Associated Protein Kinase (MAPK) signaling (Razy-Krajka et al, 2018), which positions the first heart progenitors medially and the *Tbx1/10+* second-generation multipotent progenitors (aka second trunk ventral cells, STVCs) laterally (Fig. 1A–E'). As is the case in vertebrates, Tbx1/10 promotes pharyngeal muscle specification, in part by antagonizing the cardiac fate (Wang et al, 2013; Tolkin and Christiaen, 2016; Song et al,

2022; Razy-Krajka et al, 2018). It is thus essential that *Tbx1/10* be activated after division of multipotent progenitors, to allow for the emergence of the first heart lineage, which produces the majority of cardiomyocytes in Ciona (Wang et al, 2019).

We previously showed that division of multipotent cardiopharyngeal progenitors is necessary for *Tbx1/10* activation, which is not detected following lineage-specific inhibition of either G1-S or G2-M transition by misexpression of Cdkn1 or Wee1, respectively (Razy-Krajka et al, 2018). Here we confirmed these results, and determined that *Tbx1/10* expression does not recover in division-inhibited cardiopharyngeal progenitors (Fig. 1C–G), unlike *Ebf*, the pharyngeal muscle determinant that is merely delayed by mitosis inhibition (Razy-Krajka et al, 2018). When only one of the two cardiopharyngeal progenitors inherited the *Foxf*-enhancer-driven Wee1-expressing plasmid, which typically occurs in ~40% of embryos due to mosaic incorporation of electroporated transgenes (Gline et al, 2015), only the unaffected cell induced *Tbx1/10* expression, indicating that cell division is required cell-autonomously (Fig. 1F,G'; Appendix Fig. S1). Mitosis is thus necessary for *Tbx1/10* activation in cardiopharyngeal progenitors.

Next, we sought to induce precocious division of cardiopharyngeal progenitors to test whether mitosis also suffices to trigger *Tbx1/10* expression. To this aim, we reused the minimal *Foxf* cardiopharyngeal enhancer (Beh et al, 2007) to misexpress Cdc25, a conserved phosphatase that antagonizes the Wee1 kinase and promotes entry into mitosis (Donzelli and Draetta, 2003). Consistent with a stereotyped and synchronous developmental progression, control cardiopharyngeal progenitors typically divide between 13 and 14 h post-fertilization (hpf) at 18 °C (FABA stage 23 and 24; Hotta et al, 2007) (Fig. 1A–E). By contrast, Cdc25 overexpression (Cdc25$^{OE}$) caused progenitor cells to divide as early as 11 hpf in ~20% of embryos, completing mitosis and proceeding to the second division ~2 h earlier than the equivalent cells in control embryos (Fig. 1H,I), albeit with substantial defects in orientation and asymmetry (Appendix Fig. S2).

Having established that Cdc25$^{OE}$ can induce precocious division of multipotent cardiopharyngeal progenitors, we assayed the sufficiency of mitosis for *Tbx1/10* activation (Fig. 1H,I'). In controls, the proportions of embryos with *Tbx1/10+* cells become equivalent to the percentage of embryos with divided cells within approximately 30 min, suggesting that mitosis is rapidly followed by the onset of *Tbx1/10* transcription, as assayed by intronic probes that detect nascent transcripts (Wang et al, 2013) (Fig. 1D',E'). By contrast, while Cdc25$^{OE}$ caused precocious mitosis, *Tbx1/10* expression was first detected approximately 2 h later, starting at around the same time as in control embryos (Fig. 1H,I'; Appendix Fig. S3). Careful examination of *Tbx1/10* expression in Cdc25$^{OE}$ cells indicated a slight reduction in expression and a sustained pattern of lateral expression after oriented divisions (Appendix Fig. S4). Taken together, these data indicate that mitosis is necessary but not sufficient to trigger *Tbx1/10* expression in the cardiopharyngeal lineage.

To better understand cell cycle progression and the Cdc25$^{OE}$ phenotype, we used the PCNA::GFP marker, a helicase that localizes to DNA replication forks, forming conspicuous nuclear dots during S-phase (Ogura and Sasakura, 2016) (Fig. 1J–L). PCNA::GFP nuclear dots tend to increase in size and become less numerous as replication forks coalesce and genome duplication approaches completion (Schönenberger et al, 2015) (Fig. 1J,K), a feature that we leveraged to evaluate S-phase progression, by

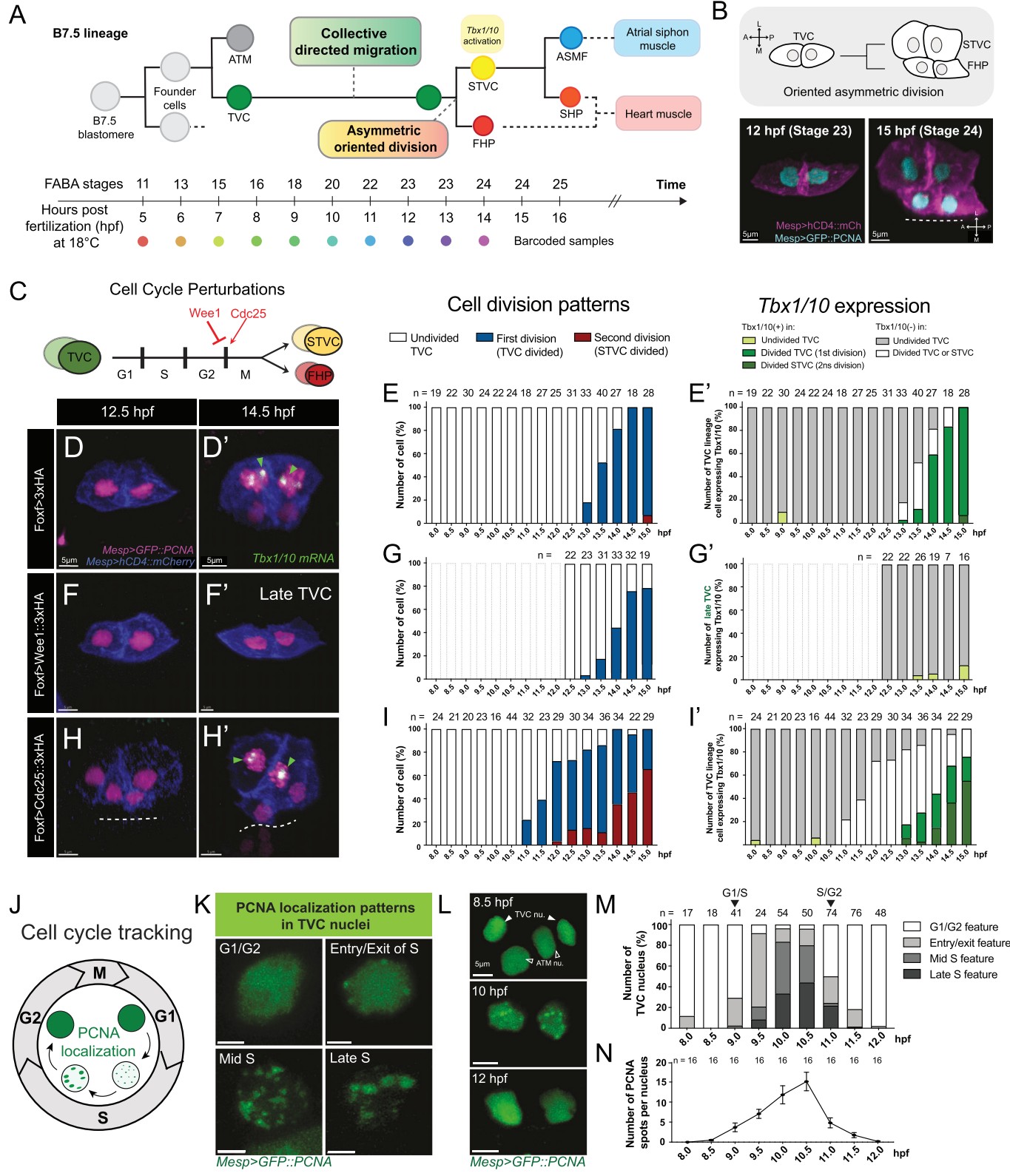

**Figure 1.  Mitosis is necessary but not sufficient for *Tbx1/10* activation in the cardiopharyngeal lineage.**

(A) (Top) The cardiopharyngeal lineage in *Ciona robusta*. The lineage of a B7.5 blastomere is shown to give rise to two founder cells, each producing a TVC and an ATM. (Bottom) Correspondence of FABA stages, post-fertilization developmental time points (18 °C), and color codes of scRNA-seq barcodes. TVC trunk ventral cell, ATM anterior tail muscle, STVC second trunk ventral cell, FHP first heart precursor, ASMF atrial siphon muscle founder cells, SHP second heart precursor. (B) (Top) Schematic diagram of asymmetrically oriented division of TVCs. (Bottom) Confocal images of before (left, 12 hpf) and after (right, 15 hpf) TVC division. Cyan: nuclei (NLS::LacZ); Magenta: cell membranes (hCD4::mCherry). The dashed line represents the embryo midline. M medial, L lateral. Scale bar = 5 µm. (C) Schematic of TVC cell cycle stages and genetic perturbations of mitotic entry. (D–I′) Cell division patterns (E, G, I) and *Tbx1/10* expression (E′, G′, I′) following cell cycle perturbations. Control TVC division (D, E, 3′HA), inhibition of TVC mitotic entry (F, G, Wee1::3′HA), and induction of TVC mitotic entry (H, I, Cdc25::3′HA) conditions are examined from 8 to 15 hpf. Perforated bars in (G, G′) indicate timepoints not analyzed. Magenta: nuclei (GFP::PCNA); blue: cell membranes (hCD4::mCherry); green arrowhead: *Tbx1/10* mRNA, Scale bar = 5 µm. (J) Schematic of variability of PCNA puncta patterns in the TVC nuclei associated with progression through the cell cycle. (K) Confocal images of PCNA puncta distribution in individual TVC nuclei at different stages of the cell cycle. Green: GFP::PCNA. Scale bar = 2.5 µm. (L–N) Determination of the S phase of TVC using PCNA. GFP::PCNA expressed in the B7.5-lineage under the *Mesp* enhancer at 8–12 hpf. Representative confocal images showing the G1, S, and G2 stages of TVC at 8, 10, and 12 hpf (L). Green: GFP::PCNA. Scale bar = 5 mm. Developmental distribution of four PCNA localization patterns (M). Quantification of GFP::PCNA punctæ per nucleus across developmental stages. Error bars show standard error of the mean (SEM). Data represent two biological replicates (N). No blinding was included in the analysis. Source data are available online for this figure.

quantifying the number and size distribution of PCNA spots in time series of fixed embryos (Fig. 1L–N). These analyses indicated that multipotent progenitors enter S-phase at approximately stage 18 (9 hpf at 18 °C) and exit at stage 22 (~11 hpf), albeit with frequent asynchrony between the leader and trailer cells (Appendix Fig. S5). Using this approach to examine cell cycle progression following Cdc25$^{OE}$, we determined that the G1/S transition was shifted only ~30 min compared to controls. As expected, however, the main effect of Cdc25$^{OE}$ was to trigger precocious mitosis in cells that had presumably completed S-phase and entered G2, thus reducing the duration of G2 from ~2 h to less than 30 min (Appendix Fig. S5).

## Single-cell developmental trajectories reveal transcriptome maturation in multipotent progenitors

Among various possible explanations, the lack of precocious *Tbx1/10* expression following forced division suggested that cardiopharyngeal progenitors become competent for mitosis-dependent *Tbx1/10* activation toward the end of interphase, once cells have completed directed collective migration and S-phase, and advanced into G2. Both cell-autonomous and cell-extrinsic factors determine competence and instruct fate decisions, including the transcription factor Hand-r/NoTrlc and FGF-MAPK signaling, which induce *Tbx1/10* activation in the cardiopharyngeal lineage (Razy-Krajka et al, 2018). However, these known regulators are already active in early cardiopharyngeal progenitors and unlikely to limit early *Tbx1/10* activation following precocious cell division.

Focusing on cell-autonomous determinants of cardiopharyngeal competence, we harnessed single-cell RNA-seq to gain insights into transcriptome dynamics as multipotent progenitors progress through the cell cycle, and migrate collectively, until they divide to produce first heart precursors (FHPs) and *Tbx1/10*+ multipotent cardiopharyngeal progenitors (aka second trunk ventral cells, STVCs; Fig. 1A,B). We reasoned that FAC-sorting cardiopharyngeal lineage cells from embryos collected every hour through ten time points encompassing the whole 7-h interphase would leverage the natural variability between individual cells, and allow for the reconstruction of a developmental trajectory providing high-resolution insights into transcriptome dynamics.

We streamlined the experiment, and avoided technical batch effects, by developing a multiplexing approach to collect the entire dataset in one experiment. We created a library of 20 reporter constructs each containing a unique 9-nucleotide barcode in the 3′UTR of the *Mesp > GFP* reporter that labels the B7.5/cardiopharyngeal lineage (Fig. 2A; Appendix Fig. S6). We positioned the barcodes to optimize recovery by RNA sequencing, and used pairs of unique barcoded reporters for each individual sample. To obtain a ten time-points series, we took advantage of *en masse* electroporation of Ciona eggs to generate samples fertilized and transfected with pairs of unique barcoded reporters every hour, and collected all samples 14 h after the first fertilization and electroporation. This approach yielded a complete 5 to 14 h post-fertilization (hpf) time series encompassing stages 11 to 24/25, starting with late gastrula embryos containing *Mesp*+ naive mesodermal progenitors (aka founder cells), prior to the birth of multipotent cardiopharyngeal progenitors, and ending with pre-hatching larvae, which possess first heart precursors and second *Tbx1/10*+ multipotent progenitors after division (Figs. 1A and 2A).

This first experiment yielded 2595 high-quality single cell transcriptomes where 500–70,000 reads detected 428 to 6059 expressed genes in 98% of the cells (Appendix Fig. S6A). We processed single-cell transcriptome data using standard methods incorporated in the Scanpy package (Wolf et al, 2018), in addition to our self-supervised graph-based denoising method DEWAKSS (Tjärnberg et al, 2021), which led us to use 35 principal components for dimensionality reduction and $k = 18$ nearest neighbors for denoising (Appendix Fig. S6B,C).

We specifically amplified and sequenced the barcode-containing regions of expressed transgenes in order to maximize recovery of sample barcodes in individual transcriptomes. The 20 sample barcodes (SBC) were added to the *Ciona robusta* reference genome as artificial genes, and the raw sequencing data were mapped to the genome using 10x Genomics Cell Ranger to generate a UMI count matrix comprising the sample barcodes. We could assign sample barcodes to 50% of the single cell transcriptomes (1261/2517; Appendix Fig. S6E). Remarkably, the barcoded transcriptomes pertained to 6 related clusters separated from the unmarked clusters as illustrated on a projection of the multidimensional dataset in two dimensions using the UMAP algorithm (Fig. 2B; Appendix Fig. S6F,G). Inspection of known markers *Mesp/MSGN1*, *Foxf*, *Gata4/5/6* and *Myod/Myf5* confirmed that the clusters containing most barcoded transcriptomes correspond to B7.5 lineage cells (Fig. 2C; Appendix Fig. S6H,I). The non-barcoded clusters thus reflect "contamination" by cells outside the *Mesp > GFP*-labeled B7.5 lineage during the FACS procedure, including

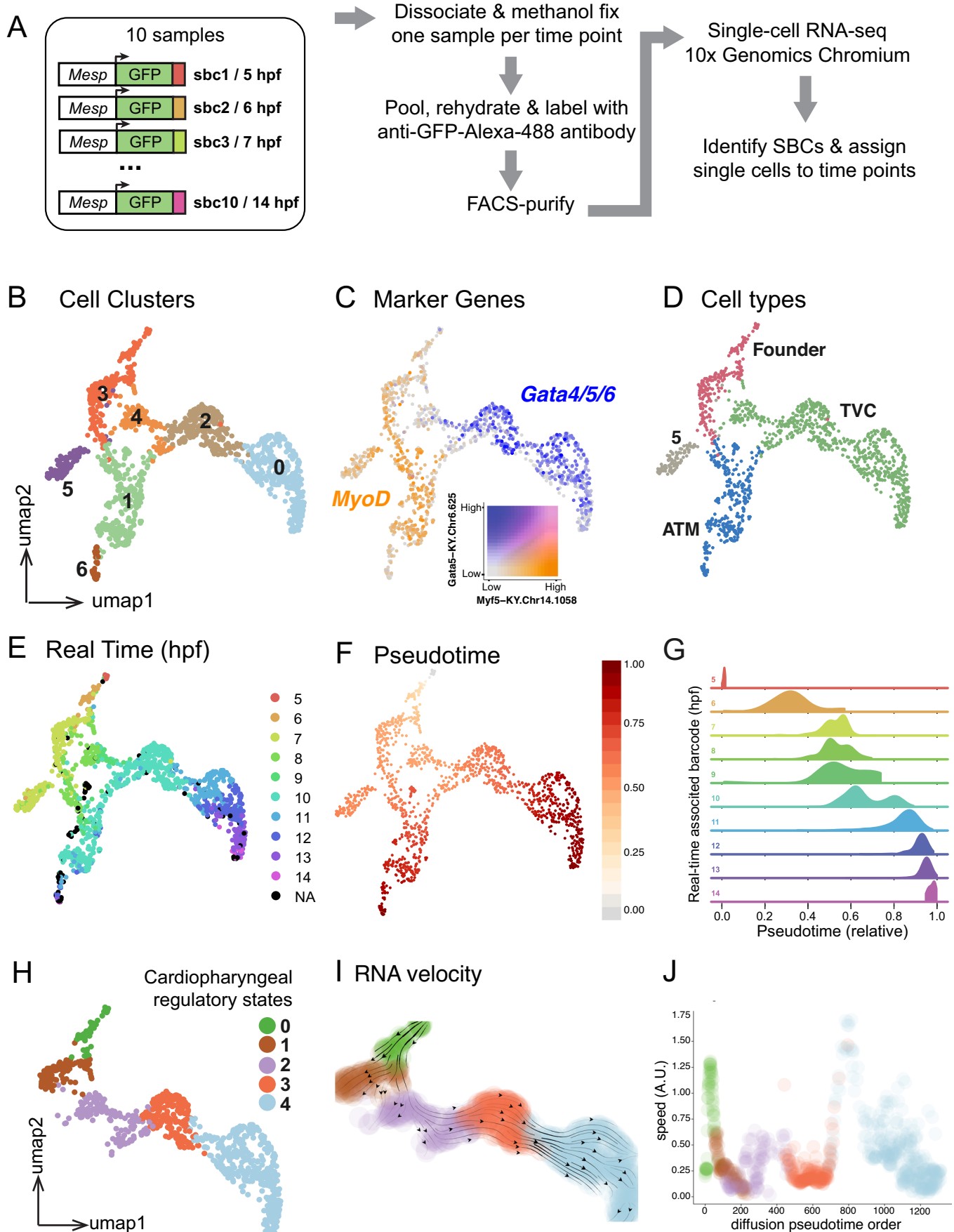

**Figure 2. Single-cell developmental trajectories reveal transcriptome maturation in multipotent progenitors.**

(A) scRNA-seq workflow outlining the lineage-specific barcoding strategy. (B) Denoised Leiden clustering of anterior tail muscles and cardiopharyngeal trajectories. (C) Trajectory-specific expression of the cardiopharyngeal progenitor-specific *Gata4/5/6* transcripts and founder and anterior tail muscle-specific *Myod/Myf5* transcripts. (D) Assignment of cell types to trajectories based on lineage-specific gene expression. (E) Reconstruction of cell trajectories using recovered real-time barcodes. (F) Pseudotime assignment on cell trajectories. (G) Distribution of real-time associated barcodes over pseudotime. (H) Cell states of the cardiopharyngeal progenitor trajectory derived from agglomerative clustering. (I) UMAP representation of the cardiopharyngeal only trajectory colored by cell state and overlayed by RNA velocities calculated from denoised data and diffusion pseudotime, and represented by streamlines. (J) Dot plot of all cells of the cardiopharyngeal trajectory ordered by diffusion pseudotime and colored by cell state. The y axis represents the speed (i.e., the length) of the velocity vector of each cell.

*Foxf+* trunk epidermal cells, for example. This indicated that most B7.5 lineage cells were effectively barcoded, and enriched ~500 times in the sorted sample, since *Mesp > GFP+* cells typically represent <0.1% of whole embryo cell suspensions (Christiaen et al, 2008; Wang et al, 2018). Of note, similar to our previous reports (Christiaen et al, 2008; Wang et al, 2019), we observed a relative depletion of late ATMs (see below), which we attribute to lower "sortability" of these large cells with distinct forward and lateral scatter properties.

With the exception of SBC00 and SBC20, individual barcodes displayed high ( > 0.98) correlation coefficients between cotransfected pairs, and very low ( < 0.05) correlation with other barcodes electroporated independently (Appendix Fig. S6D). This indicated that our transgene-based barcoding and multiplexing strategy allowed us to process pooled samples and demultiplex datasets in silico, assigning a time point of origin to most individual single-cell transcriptomes obtained from B7.5 lineage cells, albeit with a disparate representation of each sample (Appendix Fig. S6E).

Our lineage-specific time series lent itself to developmental trajectory analysis, as two-dimensional UMAP projection clearly distinguished between the anterior tail muscle and cardiopharyngeal/trunk ventral cell trajectories, both stemming from shared origins in the naive *Mesp+* mesoderm (*aka* founder cells; Figs. 1A and 2B–D; Appendix Fig. S6F–I). Pseudotime inference corroborated UMAP projection and was concordant with real time distributions, illustrating that temporal changes in transcriptome composition account for most of the variance (Fig. 2E,G). Reasoning that seemingly continuous gene expression changes reflect transitions between successive regulatory states (Moris et al, 2016), we used a clustering approach to segment individual trajectories, which suggested that cardiopharyngeal progenitor cells transition through 5 predicted regulatory states (Fig. 2H). An orthogonal RNA velocity-based analysis corroborated both directionality of the cardiopharyngeal trajectory (Fig. 2I; Appendix Fig. S7A–C), and the distinct regulatory states, which were separated by increasing "speed" in gene expression changes (Fig. 2J; Appendix Fig. S7D–F).

Gene expression denoising and mapping cells onto lineage-specific pseudotemporal sequences provided a high-resolution view of gene expression dynamics, as illustrated by the sequential activations of key regulators of cardiopharyngeal development *Mesp, Ets1/2, Foxf, Gata4/5/6, Nk4/Nkx2.5,* and *Hand1/2* (Fig. 3A; Appendix Fig. S9A,B). Leveraging this high-resolution view of whole transcriptome dynamics, we identified clusters of genes displaying distinct (pseudo)temporal profiles of expression along the anterior tail muscle and cardiopharyngeal progenitor trajectories, separately (Fig. 3; Appendix Fig. S9C). In the anterior tail muscle trajectory, 8 clusters of genes reflected transition through 3-to-4 regulatory states that seemingly stabilize as early as ~9 hpf in a

state marked by upregulation of cluster 6 genes, including *Myod/Myf5, Smyd1* and *Myl1*, and downregulation of cluster 0 and 2 genes, comprising naive *Mesp+* mesoderm markers and primed cardiopharyngeal genes such as *Rgs21, Ccna, Zfp36L1/2*, which indicated differentiation toward a tail muscle cell type in post-mitotic cells (Appendix Fig. S9B; Dataset EV2). Accordingly, differential gene expression analysis using the scRNA-seq data indicated that cluster 6 genes were more highly expressed in the anterior tail muscle trajectory (Fig. 3C), and >11-fold enriched in anterior tail muscle clusters (Fig. 3D).

In the cardiopharyngeal trajectory, each regulatory state was marked by a combination of relative expression for 8 clusters of genes. For example, states 2 and 3 differ primarily by the relative dynamics of gene clusters 2 and 6, whereby cluster 2 appears to peak in state 3 after an expression onset during state 2, while cluster 6 genes peak in states 1 and 2 and become downregulated in state 3 (Fig. 3B). Notably, gene clusters 0, 1, and 6 appear to be "off" and cluster 2 becomes downregulated in state 4, whereas cluster 4 and 5 peak toward the end of the cardiopharyngeal trajectory. State 4 comprises mostly cells collected from >11 hpf embryo, suggesting that it corresponds to the G2 phase, which sees the emergence of the cellular competence to divide and activate *Tbx1/10* (Fig. 1A). We thus considered regulatory state 4 as the mature state, characterized by low expression of gene clusters 0, 1, and 6, downregulation of cluster 2 and peak expression of clusters 4 and 5.

Importantly, dynamic genes comprised developmental regulators and cell biological effectors in addition to classic "cell cycle genes", such as *Cdc25*, suggesting that the mature state does correspond to a developmentally significant regulatory state (Fig. 3B; Dataset EV2). Finally, both published and new whole mount in situ hybridization assays corroborated the predicted expression dynamics for selected genes (Fig. 3C,F,G; Appendix Fig. S9C). For example, the signaling molecule coding genes *Ptch1* and *Bmper*, were predicted to be activated specifically in the cardiopharyngeal progenitor lineage (Fig. 3F) toward the end of the mature state (Fig. 3G), which we confirmed by fluorescent in situ hybridization (Fig. 3H,I; Appendix Fig. S9D).

Differential gene expression and relative enrichment analyses indicated that gene clusters 2, 4, and 5 were more highly expressed in the cardiopharyngeal progenitors, compared to anterior tail muscles, while gene cluster 0 was downregulated in both trajectories, suggesting an early B7.5 lineage expression. These analyses reveal that successive regulatory states and corresponding gene clusters captured cardiopharyngeal-specific transcriptome dynamics (Fig. 3C,D; Dataset EV2).

We previously reported that multipotent cardiopharyngeal progenitors display the hallmarks of multilineage transcriptional priming, whereby both cardiac and pharyngeal muscle-specific

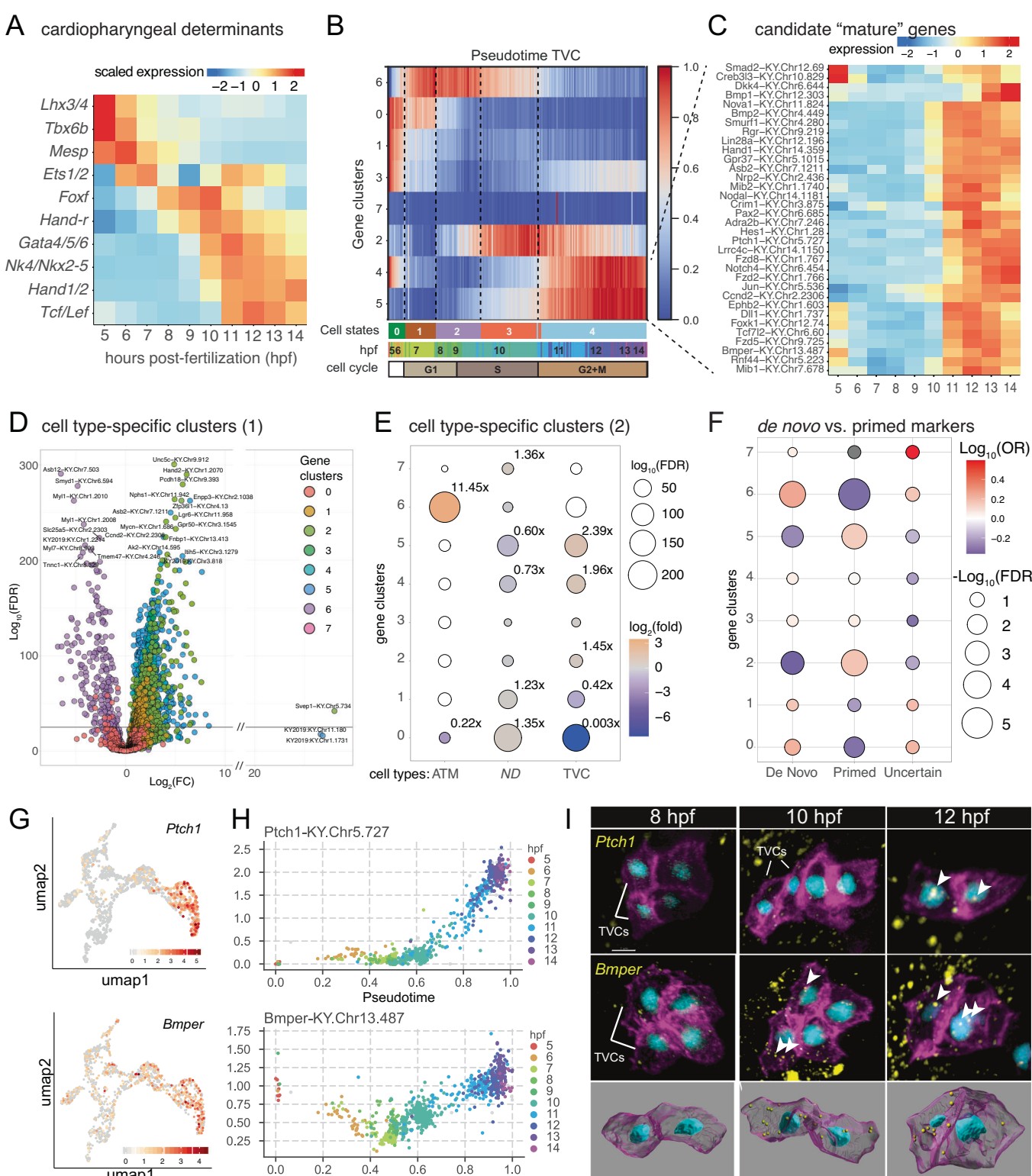

programs are started, albeit incompletely, in multipotent cardio-pharyngeal progenitors (Razy-Krajka et al, 2018, 2014; Wang et al, 2019). Enrichment analysis indicated that multilineage priming, as defined in Wang et al (Wang et al, 2019), starts with the expression of cluster 2 genes, where primed genes are enriched, and is complete when cells reach the mature state and activate genes from cluster 5, which is also enriched in primed genes (Fig. 3F). Note that cluster 6 anterior tail muscle markers are enriched in de novo

◄

**Figure 3.   Gene expression dynamics during cardiopharyngeal progenitor maturation.**

(**A**) Expression dynamics of known cardiopharyngeal determinants. (**B**) Clustered gene expression profiles along the cardiopharyngeal progenitor trajectory. (**C**) Dynamics of cluster 4 and cluster 5 candidate genes selected for targeted CRISPR/Cas9-mediated mutagenesis. (**D**) Volcano plot showing differential gene expression between the cardiopharyngeal (Log2(FC) > 0) and anterior tail muscle (Log2(FC) < 0) with individual genes color-coded as per their clustering along the cardiopharyngeal trajectory. Values calculated from the scRNA-seq experiments described in Fig. 2. (**E**) Fisher's exact tests calculated from the data presented in (**D**) showing the enrichment of cluster 2, 4, and 5 genes among cardiopharyngeal (TVC) markers and of cluster 2 and cluster 6 genes among anterior tail muscle (ATM) markers. (**F**) Fisher's exact tests for the enrichment of primed cardiopharyngeal genes (defined in Wang et al, 2019) among cluster 2 and cluster 5 genes, showing that multilineage priming is completed through transcriptome maturation. OR odds ratio. FDR false discovery rate. (**G–I**) Validation of predicted dynamics of 2 cluster 5 genes, *Ptch1* and *Bmper*. (**G**) Expression of *Ptch1* and *Bmper* in the cardiopharyngeal precursors. (**H**) Dynamics in *Ptch1* and *Bmper* expression levels as a function of pseudotime. Color code corresponds to barcodes assigned to developmental real time. (**I**) Fluorescent in situ hybridization showing activation of *Ptch1* and *Bmper* transcription in the cardiopharyngeal lineage. Nuclei are marked with NLS::LacZ (cyan), membranes are marked with hCD4::mCherry (magenta), and transcripts (yellow) are indicated with arrowheads. Bottom row shows segmented TVCs with transcripts detected as yellow spots. Source data are available online for this figure.

expressed pharyngeal muscle markers because the latter reactivate part of the non-cardiac muscle program expressed in primary tail muscle cells (Wang et al, 2019; Razy-Krajka et al, 2014). Taken together, these analyses indicate that cardiopharyngeal progenitor maturation comprises transitions that correspond to downregulation of naive *Mesp+* mesoderm and primed anterior tail muscle genes, as well as progressive multilineage priming by sequential activation of both cardiac- and pharyngeal muscle-specific markers.

## Generalization to the whole embryo

To expand the above proof of concept, we designed a library of barcoded reporters using the ubiquitously active *Ef1-alpha* driver, and repeated the time series collection between 5 and 14 hpf at 18 °C followed by cell dissociation, pooling, methanol fixation, rehydration and single cell RNA-seq. We omitted the FACS step to profile the entire developmental sequence of the whole embryo in one experiment. We obtained >21,000 single cell transcriptomes altogether expressing >11,000 genes and distributed across 57 clusters (Fig. 4A). We recovered temporal barcodes for 42.6% (8974/21,087) of the cells, which is markedly lower than in our B7.5 lineage-focused experiment, because we did not FACS-purify transfected cells (Fig. 4B; Appendix Fig. S10). Nonetheless, all clusters contained barcoded transcriptomes, allowing us to map them back onto the developmental sequence, with the notable exception of the germline (cluster 54, Fig. 4A), which does not effectively transcribe electroporated transgenes (Ohta and Christiaen, 2024). Using a label propagation approach, we expanded the barcode-derived assignment of real time to 86.6% of the dataset (Fig. 4C). As observed for the cardiopharyngeal trajectory (Fig. 2), pseudotime values were highly correlated with both real and inferred real times (Appendix Fig. S10C), indicating that our multiplexing approach faithfully recovered the transcriptome dynamics underlying developmental progression in the whole embryo. To facilitate exploration of these datasets, we created web-based interfaces using the ShinyApp package, and made them available at http://christiaenlab-sars.com/scRNA-seq-datasets/.

Remarkably, UMAP-based projection of the whole dataset illustrated cell fate diversification from early clusters located centrally toward diverging late cell identities located peripherally on the two-dimensional plot (Fig. 4A–C). Cursory annotation using known markers identified the main tissue types throughout their transition from the gastrula to pre-hatching larval stage (Fig. 4D–F; Appendix Fig. S11). Specifically, the notochord and muscle clusters, including a small number of B7.5/heart lineage cells, separated

from the earliest time points onward, consistent with precocious cell fate specification, as early as the 64-cell stage, and post-mitotic differentiation during embryogenesis. Closer inspection of the notochord cluster suggested that its internal structure followed primarily the temporal sequence, without effectively distinguishing between the primary (A-line) and secondary (b-line) lineages. By contrast, and similar to a previous study (Cao et al, 2019a), the central nervous system clusters exhibited a complex structure that was only partially driven by temporal progression, and blurred the clonal distinction between A- and a-line derivatives. Likewise, the expansive and complex epidermal clusters separated by time points from 6 to 8 hpf, albeit displaying the premise of internal structure, whereas the 9 to 14 hpf cells diversified and clustered orthogonally to temporal progression, principally according to the dorsoventral and anteroposterior (e.g., tail-trunk) patterning, peppered with diverse epidermal sensory neurons. As for the central nervous system, initial clonal distinction between a- and b-line derived epidermal cells did not appear to be a main driver of transcriptome diversity, even at early time points.

By contrast with ectodermal clusters, endomesodermal lineages showed clearer signs of clonally guided developmental progression over time (Fig. 4A–D). For example, posterior B-line trunk endoderm emerged as early at 8 hpf, A-line endoderm and A7.6-line mesoderm (aka trunk lateral cells, TLCs) diverged transcriptionally as early as 6 hpf in gastrula embryos (Fig. 4; Appendix Fig. S11). As for the muscle clusters, B-line mesenchymal lineages were already separated at the onset of the time series, and continued to diversify in progressively canalized manner toward defined late states.

Specifically, the B7.7 lineage (Fig. 4G,H; Appendix Fig. S12) appeared transcriptionally distinct from the B8.5 lineage as early as 6 hpf and further bifurcated into two sublineages, expectedly B8.13 and B8.14, around 10 hpf (mid-tailbud stage). Following an approach like that used for the cardiopharyngeal lineage (Fig. 2), we identified three pre- and two post-bifurcation regulatory states for each branch. These transcriptional states coincided with 10 gene clusters, including late, branch-specific clusters 2 and 6 (Fig. 4J). We further noted that cluster 4 and cluster 1 followed transient expression profiles, consistent with mature and immature states of the multi/bipotent B7.7 progenitors, respectively. Of note, as for the cardiopharyngeal progenitors, *Cdc25* [KY.Chr5.722] was part of the mature B7.7 genes (cluster 4) and transiently expressed shortly before the bifurcation (Appendix Fig. S12).

The B8.5 lineage followed a more complex trajectory profile, resulting in three transient and three distinct terminal

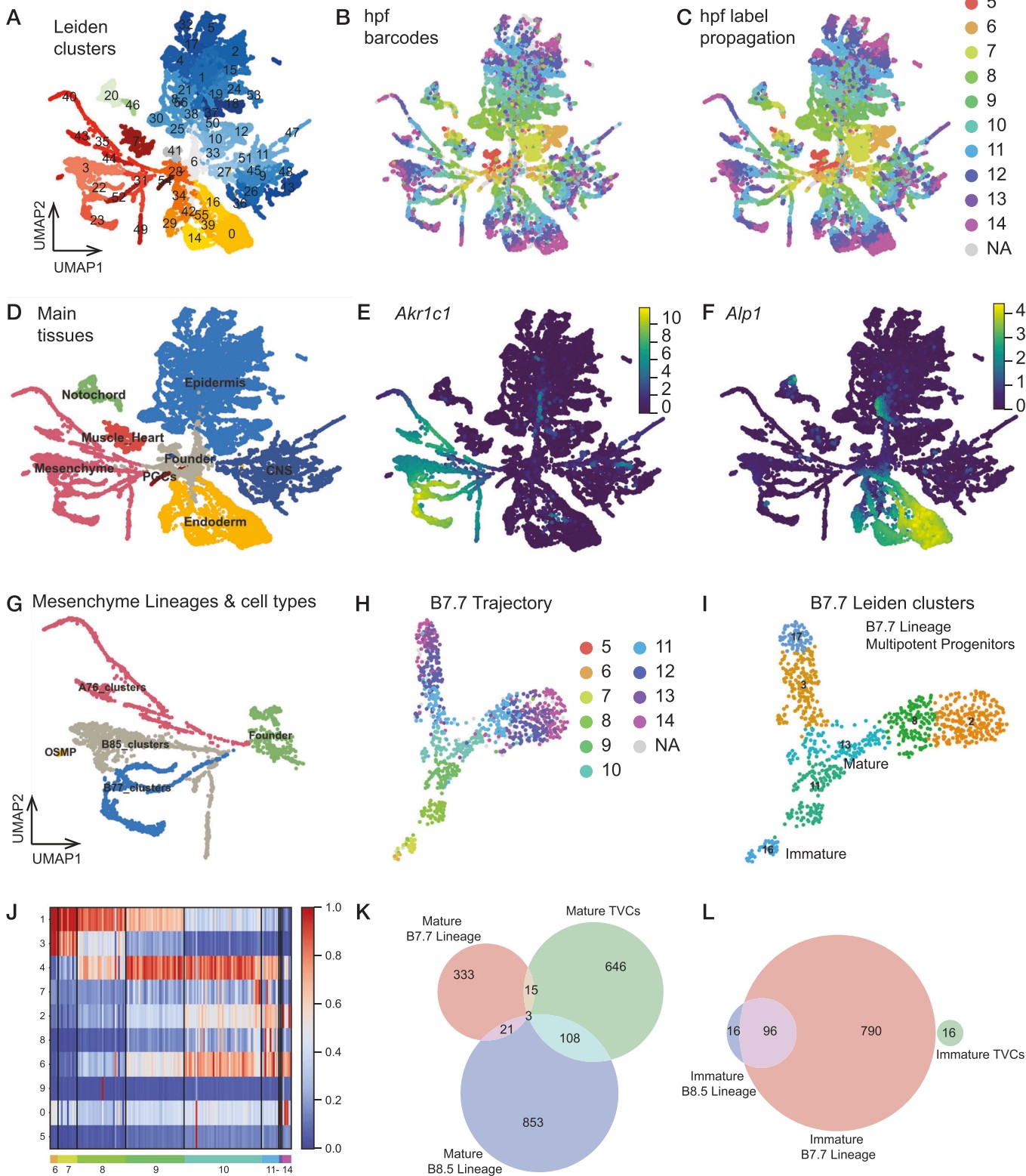

◄

**Figure 4.  Whole embryo single-cell trajectories reveal maturation signatures in canalized endomesodermal lineages.**

(A–D) Uniform manifold approximation and projection (UMAP) of the whole embryo scRNA-seq dataset colored and labeled by Leiden clusters, with shades of blue for epidermal and neuronal clusters, yellow for endodermal clusters, red for mesenchymal, germ line and muscle clusters and green for the notochord (A). The same UMAP is colored by developmental time obtained from the electroporated barcodes (B), or inferred through label transfer from closest neighbors (C). The same UMAP, colored by broad cell types, matching the color scheme detailed in a (D). (E, F) Denoised expression of the pan-mesenchymal marker Akr1c1 (e; "KY.Chr1.2267"), and the pan-endoderm marker Alp1 (f; KY.Chr6.211"). (G) Denoised Leiden clustering of mesenchymal tissues. (H, I) Differentiation trajectory of the B7.7 mesenchymal lineage colored by inferred developmental time (H) and denoised Leiden clustering (I). Note that cluster 16 corresponds to an immature state and cluster 11 to a mature transcriptomic state. (J) Differential expression of genes during differentiation of the B7.7 mesenchymal lineage. Gene clusters 1 and 4 correspond to the immature and mature states, respectively. (K, L) Venn diagram showing the overlap of genes included in the mature (K) and immature gene clusters (L) of the mesenchymal B7.7 (red) and B8.5 (blue) lineages and the TVCs (green).

transcriptional states of B8.5-lineage cells by late tailbud stages (Fig. 4G,I; Appendix Fig. S12). The terminal transcriptional states (regulatory states 0, 15 and 25) were defined by *Irx.c* (*KY.Chr3.1404*), *Mist/Bhlha15* (*KY.Chr3.1309*) and *Spon1/Hmcn1-KY.Chr10.172*, respectively. Of note, B8.5-derived cluster 15 converges toward a transcriptional state that becomes indistinguishable from that of B7.7-lineage cells, including expression of *Mist/Bhlha15*. In total, genes expressed in this lineage could be grouped in 7 clusters, including immature cluster 4 and mature cluster 1 (Appendix Fig. S12).

As both mesenchymal and cardiopharyngeal lineages appear to transition through intermediate regulatory states before fate diversification, we sought to identify a gene expression signature associated with multipotent progenitor maturation in these distinct lineages. By intersecting candidate mature and immature markers of each lineage (Fig. 4K,L), we identified only three candidate mature genes shared between all three lineages, namely (*Chdh-KY.Chr1.882*, *Prmt7-KY.Chr2.2197*, *Trp53i11-KY.Chr6.477*). By contrast, the B7.7 and B8.5 lineages shared 85.7% (97/112) of their candidate immature genes, consistent with their common origin as B-line mesenchyme progenitors. Thus, similar to the cardiopharyngeal progenitor trajectory, multipotent mesenchymal progenitors also appear to progress along canalized trajectories that comprise transcriptional maturation prior to branching points and subsequent fate decisions.

## Mature state-specific determinants are required for heart vs. pharyngeal muscle fate choices

Next, we sought to explore the biological significance of transcriptome maturation in multipotent cardiopharyngeal progenitors. Our initial observations suggest that, although cardiopharyngeal progenitor cells could be forced to divide precociously, they were largely incapable of activating *Tbx1/10* until shortly before its normal timing in control embryos. We reasoned that, besides mitosis, transcriptome maturation might determine the competence of progenitor cells to activate *Tbx1/10* in addition to the known regulators Hand-r and FGF-MAPK signaling (Razy-Krajka et al, 2018).

To test if transcriptome maturation by downregulation of early markers and activation of late genes underpins the acquisition of multilineage competence, we focused on candidate genes activated in mature progenitor cells toward the end of the cardiopharyngeal trajectory, which we hypothesized to be necessary for cardiopharyngeal fate decisions.

While mature clusters 4 and 5 comprised 945 and 859 genes (Dataset EV2), respectively; we cross-referenced these lists with

cardiopharyngeal markers and molecular classes, and selected 37 candidate genes, including those coding for transcription factors (*Hand*, *Hes.a*, *Foxk*, *Tcf/Lef*, *Pax2/5/8.a*, *Jun*, *Smad2/3.b*; Datasets EV3 and EV8), and signaling molecules (*Tolloid/Bmp1*, *Dickkopf/Dkk*, *Delta-like/Dll1*, …), including *trans*-membrane receptors such as *Adra2b* and *Fzd2, 5 and 8*, or the Rho GTPase Activating Protein (RhoGAP) *Depdc1b* (Dataset EV3). We designed 111 single guide RNA (sgRNA)-expressing constructs for CRISPR/Cas9-mediated and B7.5 lineage-specific mutagenesis, using the *Mesp* enhancer to express SpyCas9 (Stolfi et al, 2014; Gandhi et al, 2017) (Fig. 5A). To assay phenotypes, we scored division orientation and asymmetry, and *Tbx1/10* expression at stage 25, after control cardiopharyngeal progenitors had normally divided asymmetrically along the mediolateral axis, and the large lateral progenitors had activated *Tbx1/10* (Fig. 5B–F).

Treating division orientation and asymmetry, and *Tbx1/10* expression as categorical variables, we applied Fisher's exact tests and calculated false discovery rates (FDR) to identify the diverse combinations of selected phenotypes induced by defined perturbations. For example, CRISPR/Cas9 mutagenesis of *Adra2b* significantly inhibited *Tbx1/10* activation without affecting the cell division patterns, while targeting Frizzled-coding genes variably altered cell divisions and/or *Tbx1/10* expression. In total, we identified 25 out of 37 candidate mature genes causing a *Tbx1/10* expression and/or cell division phenotype following CRISPR/Cas9-mediated loss-of-function (Fig. 5B–G). The variable combinations of phenotypes observed hint at the partial independence of the subcellular phenomena, and thus the modular nature of biomolecular networks governing *Tbx1/10* expression and division orientation and asymmetry. On the other hand, with the exception of *Lin28*, all conditions altering division orientation and/or asymmetry also impacted *Tbx1/10* expression. This suggested that, beyond the system's modularity, the geometry of cardiopharyngeal progenitor divisions influences subsequent *Tbx1/10* expression patterns.

Focusing on the DEP domain-containing RhoGAP-encoding *Depdc1b*, fluorescent in situ hybridization validated its expression in maturing progenitor cells, starting approximately at the G1-S transition, and confirming that *Depdc1b* activation precedes oriented and asymmetric divisions in late tailbud embryos (Fig. 3B, see also Fig. 8A). To further characterize the *Depdc1b^CRISPR* division phenotype, we imaged, segmented and quantified defined morphometric parameters in both control, *Tyrosinase^CRISPR*, and *Depdc1b^CRISPR* cardiopharyngeal lineage cells (Fig. 6A–D), as part of systematic high-content image-based CRISPR screen (Failla, Wiechecki et al, in preparation). Specifically, minimum angles between axes formed by cell triplets, which report on the orientation of cell divisions, indicated that the *Depdc1b^CRISPR*

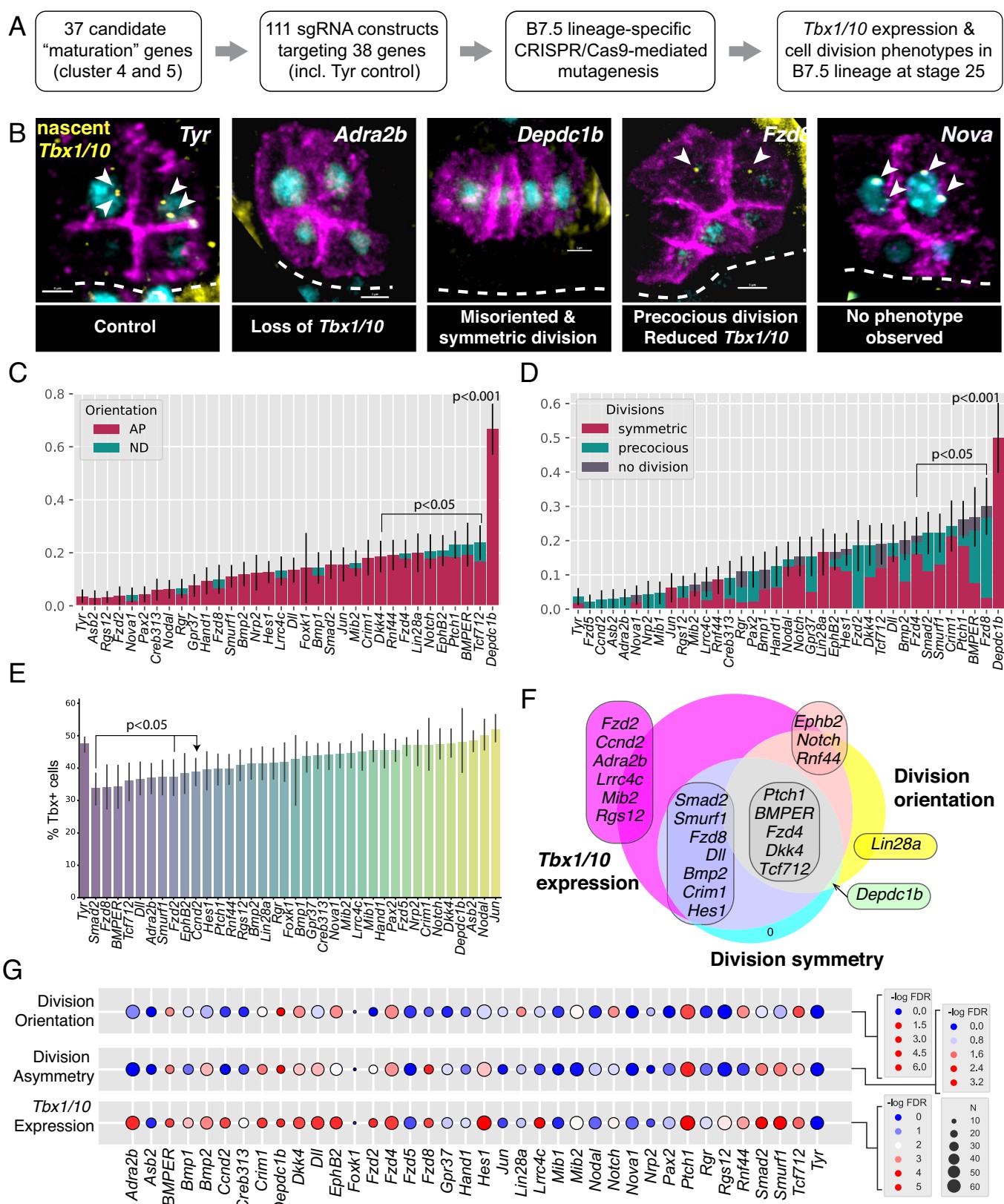

**Figure 5. Maturation state-specific determinants are required for pharyngeal vs heart muscle fate choices.**

(A) Flowchart of the CRISPR screen library generation and embryo collection. (B) Micrographs show examples of phenotypes produced. Membranes are marked with lineage-specific Mesp>hCD4::mCherry (magenta), nuclei with Mesp>LacZ (cyan), and a DIG-labeled intronic RNA probe against *Tbx1/10* transcripts (yellow). Scale bar = 5 μm. (C) Proportions of half-embryos showing the indicated division orientation phenotypes upon targeting the corresponding candidate genes (x axis) by CRISPR/Cas9. Error bars show standard error of proportion. *P* values are calculated using Fisher's exact test with a Bonferroni correction. (D) Proportions of half-embryos showing the indicated division asymmetry phenotypes upon targeting the corresponding candidate genes (x axis) by CRISPR/Cas9. Error bars show standard error of proportion. *P* values are calculated using Fisher's exact test with a Bonferroni correction. (E) Proportions of *Tbx1/10*-positive cells in half-embryos upon targeting the corresponding candidate genes (x axis) by CRISPR/Cas9. Error bars show 95% confidence intervals. *P* values are calculated using Fisher's exact test with a Bonferroni Correction. Data for (C–E) all represent three biological replicates. (F) Summary: Venn diagram showing presenting various combinations of Tbx1/10 expression and division orientation and asymmetry phenotypes. (G). Summary Fisher's exact tests statistics Phenotypes associated with CRISPR targeting of indicated genes. −log$_{10}$ of the false discovery rate (FDR) is shown with a cutoff value of 0.05. Source data are available online for this figure.

perturbation significantly alters division orientation. These data indicated that Depdc1b function is required for both the asymmetry and orientation of cardiopharyngeal progenitor divisions. This role is reminiscent of the function of *LET-99*, the *Depdc1b* homolog in *Cænorhabditis elegans*, which functions to position and orient the mitotic spindle during the first embryonic cleavage (Rose and Kemphues, 1998; Tsou et al, 2002).

Notably, even though Depdc1b controls the orientation of cardiopharyngeal progenitor divisions, its disruption caused gene expression and fate decision phenotypes following abnormal divisions (Fig. 6B–G). Specifically, *Dedpc1b*$^{CRISPR}$ caused a significant fraction of sister cell pairs to align along the anteroposterior axis, by contrast with the medio-lateral alignment typically observed in control embryos (Fig. 6B–D,G). Sister cells misalignment significantly correlated with increased contact with the lateral *Twist > hCD4::mCherry+* mesenchymal lineages (Fig. 6C,E,G), which in turn correlated significantly with *Tbx1/10* expression (Fig. 6C,F,G). Despite the variability and incomplete penetrance of the *Depdc1b*$^{CRISPR}$ phenotype, these analyses thus indicate that Depdc1b controls the orientation of cardiopharyngeal progenitor divisions, which in turn determines differential contact with the lateral mesenchyme and polarized activation of *Tbx1/10* specifically in the lateral second-generation cardiopharyngeal progenitors.

To further explore the role of Depdc1b in cardiopharyngeal progenitor divisions, we re-evaluated the Cdc25$^{OE}$ phenotype, which was characterized by precocious but also misoriented and/or symmetrical divisions, in a manner reminiscent of the *Depdc1b*$^{CRISPR}$ phenotype (Appendix Fig. S2). Since *Depdc1b* expression starts at the G1-S transition and peaks during G2 in the mature state, we reasoned that Cdc25$^{OE}$ may force cells to divide before Depdc1b reaches a functional level, thus impairing proper oriented and asymmetric division. Consistent with this hypothesis, Cdc25$^{OE}$ did not impair the onset of *Depdc1b* expression in early S-phase, but shortened the G2 phase, causing cells to divide with lower *Dedpc1b* expression level, consistent with a contribution to the Cdc25$^{OE}$ cell division orientation and asymmetry phenotypes (Appendix Fig. S13).

Taken together, these results indicate that *Depdc1b* upregulation endows mature cardiopharyngeal progenitors with the ability to divide in an asymmetric and oriented manner, allowing them to produce distinct large *Tbx1/10+* second cardiopharyngeal progenitors, laterally, and small *Tbx1/10-* first heart precursors, medially. We propose that transcriptome maturation of progenitors foster the cellular competence to generate both cardiac and pharyngeal muscle lineages, a hallmark of multipotent cardiopharyngeal progenitor cells.

## Transcriptional regulators of multipotent progenitor maturation

The above results indicated that the transcriptome of multipotent progenitors changes as they migrate and progress through the cell cycle. This molecular maturation fosters the competence to undergo oriented and asymmetric divisions, which condition the production of distinct cardiac and pharyngeal muscle lineages. Considering the importance of multipotent progenitor maturation for subsequent fate decisions, we sought to explore the mechanisms governing transcriptome dynamics during maturation.

We previously described the changes in the immediate environment of multipotent progenitors, which are born laterally, in the vicinity of the mesenchyme and notochord, before migrating ventro-medially in between the epidermis and trunk endoderm (Gline et al, 2015; Bernadskaya et al, 2019). It is likely that distinct surrounding tissues exert variable influence on migrating progenitors. Nevertheless, here we focused on possible cell-autonomous drivers of transcriptome dynamics.

Leveraging our previously published B7.5-lineage-specific chromatin accessibility data (Racioppi et al, 2019), we first sought to identify sequence motifs enriched in accessible regions surrounding genes clustered by pseudotemporal profiles along the cardiopharyngeal trajectory (Fig. 7A,B). Consistent with their roles in tail muscle development, this analysis identified motifs for regulators of tail muscle specification and differentiation, including the myogenic transcription factor Myf5/Myod in cluster 6, which corresponds to genes maintained specifically in the anterior tail muscle cells, as is the case for *Snail*, while *Meis* is expressed in both tail muscles and cardiopharyngeal progenitors, consistent with a general role in myogenesis (Fig. 7B; Appendix Figs. S9C and S14). By contrast, Forkhead/Fox family motifs were depleted in the accessible regions associated with this gene cluster, also consistent with the previously recognized specific role for Fox family factors in cardiopharyngeal fate choices (Beh et al, 2007; Woznica et al, 2012; Racioppi et al, 2019).

Extensive analysis of candidate transcription regulators based on motif enrichment and dynamic gene expression in the cardiopharyngeal lineage were consistent with known regulatory interactions and suggested new ones (Fig. 7 and Appendix Figs. S14–S20; see also Racioppi et al, 2019). Gene clusters 0 and 1 were more highly expressed in B7.5 and founder cells, at the beginning of the trajectory, and included *Mesp* and two of its upstream regulators, *Tbx6-r.b* and *Lhx3/4* (Christiaen et al, 2009). Both *Tbx6-r.b* and *Mesp* were downregulated in the anterior tail muscle and cardiopharyngeal progenitor trajectories, unlike *Lhx3/4*, which

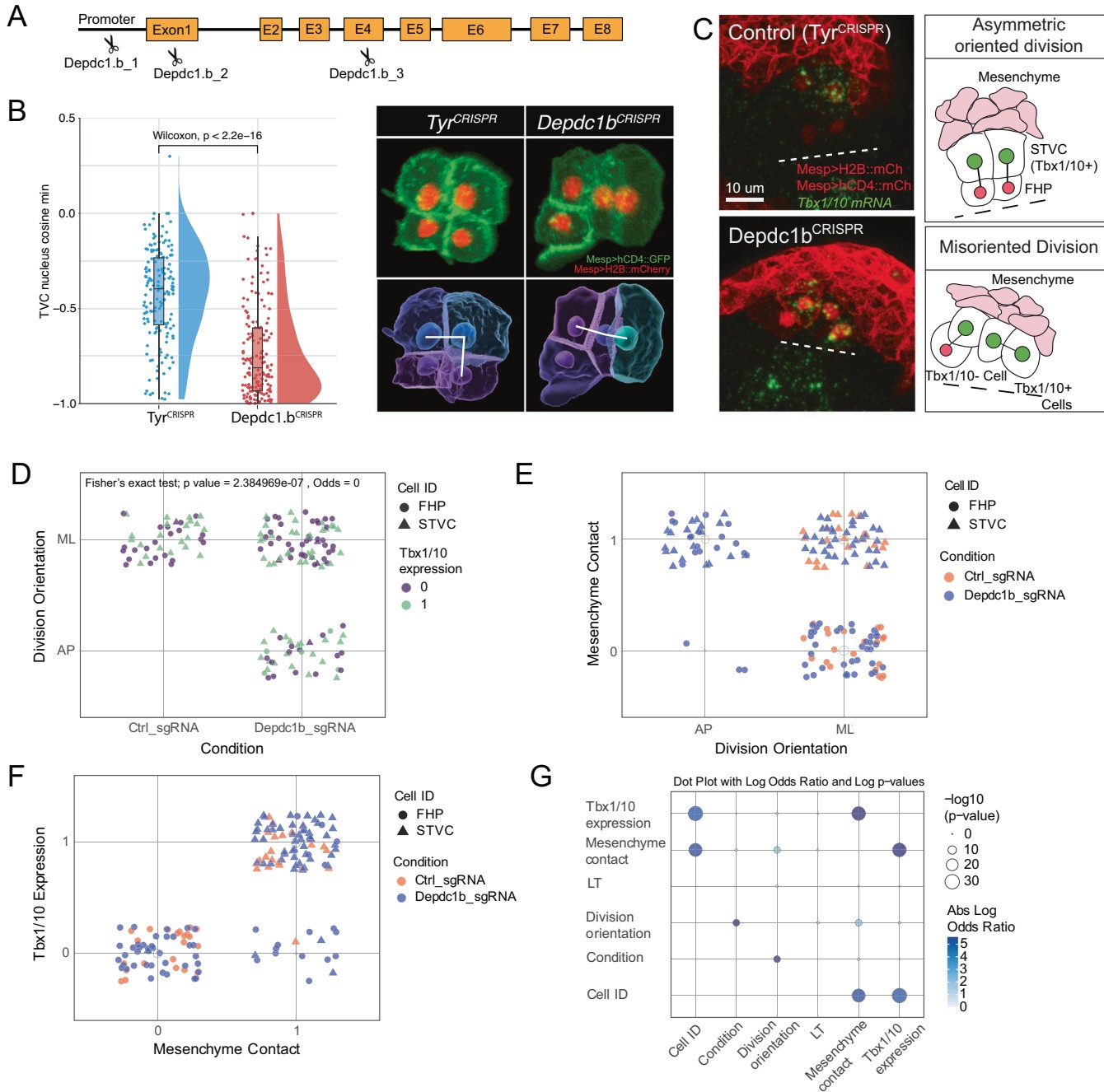

**Figure 6. Depdc1b^CRISPR cellular phenotypes affect the pattern of *Tbx1/10* expression.**

(A) Simplified map of the *Depdc1b* genomic locus with indicated guide RNA targets. (B) Range of phenotypes produced by CRISPR/Cas9-mediated disruption of the *Depdc1b* locus. The graph shows the change in division orientation as the minimum cosine of the angle between all possible pairs of axes defined by cell triplets. Micrographs show rendered volumes of cells color coded by relative volume and division orientation indicated by white lines. The data represent five and seven biological replicates for *Depdc1b* and *Tyr* conditions, respectively. (C) Micrographs showing FISH revealing the *Tbx1/10* transcripts (green) in the cardiopharyngeal lineage cells that contact the mesenchyme, which is marked with tissue-specific *Twist > hCD4::mCherry*. Cardiopharyngeal lineage cell membranes and nuclei are marked with *Mesp > hCD4::mCherry* and *Mesp > H2B::mCherry*, respectively. Dashed lines mark the embryonic midline. Diagrams to the right highlight the contact between *Tbx1/10* expressing cells and the mesenchyme. No blinding was included in the analysis. (D–F) Jitter plots showing correlations between indicated parameters. Each point is an individual half embryo scored as indicated. (D) CRISPR targets and division orientation. (E) Division orientation and mesenchymal contact, and (F) *Tbx1/10* expression. (G) Log odds ratios and significant associations of characteristics and phenotypes between control and *Depdc1b^CRISPR* conditions. The conditions significantly affect the division orientation. The orientation significantly affects the contact with the mesenchyme, which significantly affects *Tbx1/10* expression. Hence the inferred chain of causality: Depdc1b -> division orientation -> polarized contact with the mesenchyme -> asymmetric *Tbx1/10* expression, which is mediated by polarized MAPK signaling (see Razy-Krajka et al, 2018). Source data are available online for this figure.

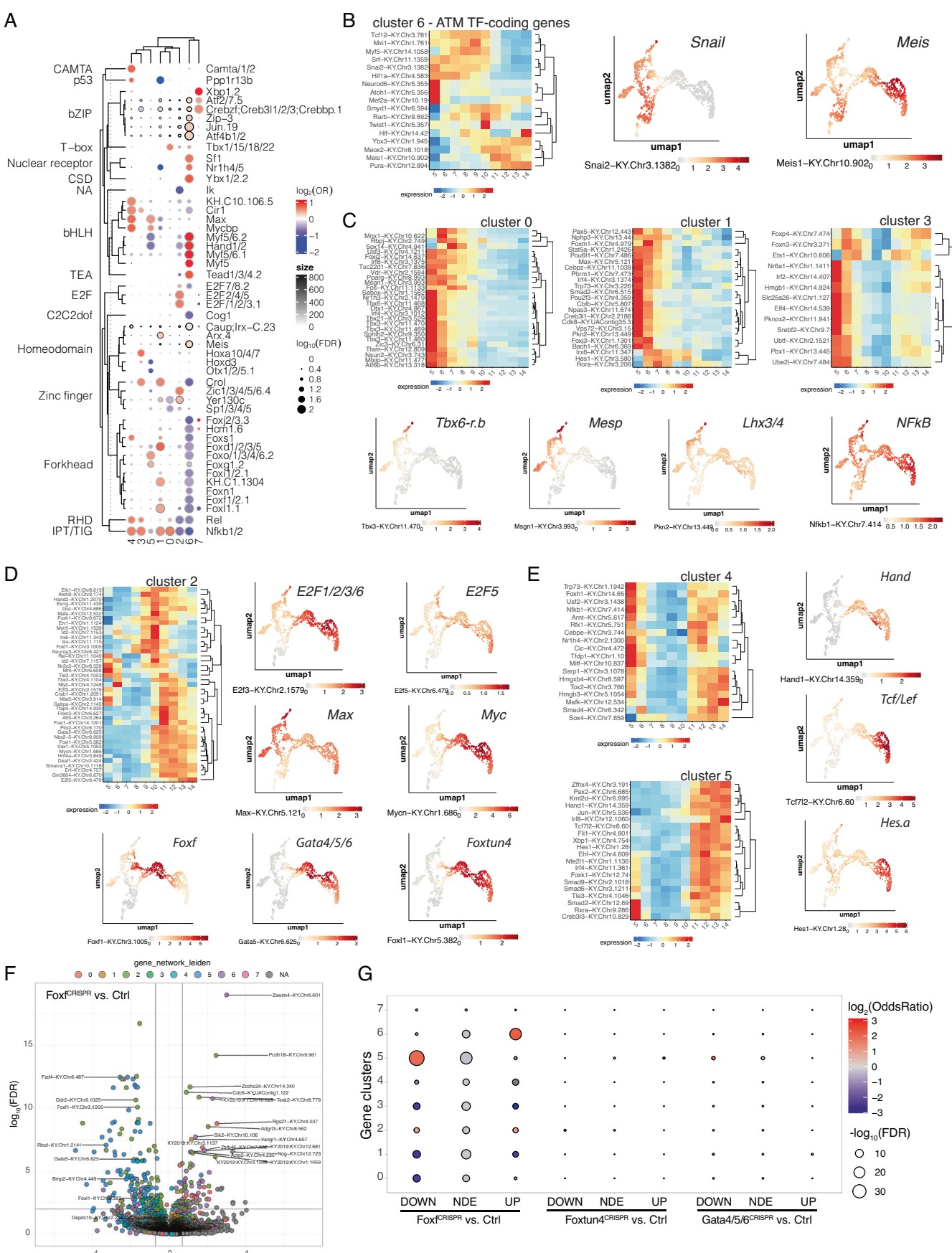

**Figure 7. Candidate transcription factors involved in the regulation of maturation genes.**

(A) Cluster-specific transcription factor motif enrichment. Dot color gives the $\log_2$ odds ratio of the occurrence of a motif in a motif set to the occurrence of a motif in the background. Dot size gives the p-value of the hypergeometric test. Outline color gives occurrences of a motif in the motif set. (B–E) Detailed scaled expression of transcription factor-coding genes in indicated gene clusters. Heatmaps show temporal expression in the cardiopharyngeal trajectory, while FeaturePlots for selected examples show both the cardiopharyngeal and anterior tail muscle trajectories (see Fig. 2S for cell types annotations). (F) Volcano plot showing differentially expressed genes between control (Ctrl) and Foxf^CRISPR conditions. $\log_2$ fold change is shown on the x axis, calculated from the CRISPR x scRNA-seq experiment. Color code represents gene clusters as shown in Fig. 3B. (G) Relative enrichments (log odds ratios and false discovery rates from hypergeometric tests) between gene clusters (defined in Fig. 3B) and differentially expressed genes in indicated CRISPR/Cas9 conditions.

was maintained at lower levels in the cardiopharyngeal trajectory, consistent with its proposed role in activating cardiopharyngeal progenitor-specific markers (Pickett et al, 2024). Accordingly, homeodomain binding motifs were enriched in accessible regions near cluster 1 and cluster 3 genes, while T-box binding motifs were enriched near cluster 0 and 6 genes, consistent with known early roles of Tbx6 transcription factors in both cardiopharyngeal lineage and tail muscle specification (Kugler et al, 2010; Yu et al, 2019; Yagi et al, 2005).

Inspecting "unscaled" expression data for transcription factor-coding gene allowed us to explore the range of expression levels (Appendix Figs. S14–S20), pointing at the more highly expressed factors with corresponding motif enrichment in early B7.5 lineage cells in this (Fig. 7A), and our previous analysis (Racioppi et al, 2019). In addition to T-box and Mrf/MyoD/Myf5 motifs, our analyses revealed enrichments of homeodomain, Zinc finger, forkhead and helix-loop-helix binding motifs near genes in early clusters 0, 1, and 3 (Fig. 7A). Focusing on the top expressed transcription factor-coding genes (groups A and B in Fig. 7B,C; Appendix Figs. S14–S20; Dataset EV4) from clusters 0, 1 and 3 identified 21 candidate transcription factors among the corresponding bHLH, Forkehad, Homeobox and T-box families, including the known determinants Tbx6-r.b and Mesp, as well as under-studied candidate cardiopharyngeal regulators such as Foxn2/3, Otx or Hes.b (Dataset EV4).

Intriguingly, NFkB motifs were enriched near both "early" clusters 0 and 1 and "cycling" clusters 3 and 4, whose expression is the lowest in the middle of the trajectory (9 and 10 hpf; Fig. 7A,C,E), and depleted near both cluster 6 anterior tail muscle genes and clusters 2 and 5 cardiopharyngeal progenitor genes.

Conversely, motifs for (a) Zic-family transcription factor(s) were enriched near cluster 2 genes, which peak during S-phase in the middle of the cardiopharyngeal progenitor trajectory (Fig. 3B). This is consistent with a possible role for Zic-r.g/ZicL, which is expressed early as part of cluster 0 (Appendix Fig. S14) and plays (an) important role(s) in myogenic mesoderm development in Ciona (Imai et al, 2002). This illustrates that, as expected, there are likely delays between transcription factor expression and activity, as potentially reflected in motif enrichment analysis.

Cluster 2 is the most transcription factor-rich cluster, with 48 genes distributed in 5 temporal subclusters, and 5 tentative "levels" of expression as defined by clusters of unscaled data (Appendix Fig. S16). This cluster comprises genes encoding some of the iconic conserved cardiac transcription factors, including Gata4/5/6, Nk4/Nkx2-5 and the key determinant Foxf (Appendix Fig. S16; Dataset EV4). This and our previous work (Racioppi et al, 2019), including motif enrichment analysis, emphasized the importance of Forkhead and GATA-family transcription factors in

cardiopharyngeal development (Beh et al, 2007; Ragkousi et al, 2011, 2009; Ragkousi and Davidson, 2010; Christiaen et al, 2008). While Gata4/5/6 stands out as the only relevant GATA family factor expressed and functional in cardiopharyngeal progenitors, several Forkhead family members are also expressed at high levels besides Foxf (e.g., Foxh.a, Foxn1/4.b and Foxtun4). Likewise, Other Ets family members potentially complement Ets1/2/Ets.b in regulating gene expression during cardiopharyngeal fate specification.

Finally, several of the late-expressed transcription factor-coding genes found in clusters 4 and 5, including Hand, Lef/Tcf and Hes.a (Fig. 7C; Appendix Figs. S18 and S19) had variable effects on cardiopharyngeal progenitors cell division and Tbx1/10 expression (Fig. 5), indicating roles in transcriptome maturation albeit not clearly detected in motif enrichment analyses (Fig. 7A). Taken together, while these observations warrant future functional analysis of a broader range of candidate transcription regulators in the cardiopharyngeal lineage, we surmise that transcription regulators found in cluster 2 play (a) pivotal role(s) in orchestrating the transitions between early specification and maturation in multipotent cardiopharyngeal progenitors.

Previous work identified cluster 2 transcription factor-coding Foxf as one of the earliest developmental control genes activated in cardiopharyngeal progenitors, in response to FGF-MAPK signaling (Beh et al, 2007; Christiaen et al, 2008). Foxf controls subsequent activation of the conserved cardiac transcription factors coding genes Gata4/5/6, Nk4/Nkx2.5 and Hand (Beh et al, 2007; Christiaen et al, 2008; Woznica et al, 2012), at least in part by promoting chromatin accessibility at cardiopharyngeal enhancers, and subsequent transcriptional activation (Racioppi et al, 2019). Transcription factors of the GATA4/5/6 family also play essential and conserved regulatory roles in cardiac vs. pharyngeal lineage specification. To assay the roles of candidate TFs during maturation of cardiopharyngeal progenitors, we performed CRISPR/Cas9-mediated mutagenesis of Foxf and Gata4/5/6 along with a forkhead family transcription factor Foxtun4, selected among other candidate TFs dynamically expressed in the cardiopharyngeal trajectory (Fig. 7D). We profiled the transcriptomes of FACS-purified B7.5 lineage cells at stage 23/24 (~12 hpf at 18 °C), when control cells reach the mature state, using transgenic barcode-mediated multiplexed scRNA-seq. We pooled four barcoded samples corresponding to Foxf, Gata4/5/6, Foxtun4-specific and control CRISPR/Cas9 reagents into a single library for scRNA-seq, followed by in silico demultiplexing.

Building on highly correlated barcode pairs (Appendix Fig. S21A), we confidently assigned 112, 126, 92 and 193 single cell transcriptomes to either Foxf, Gata4/5/6, Foxtun4 loss-of-function or control conditions, respectively (Appendix Fig. S21B).

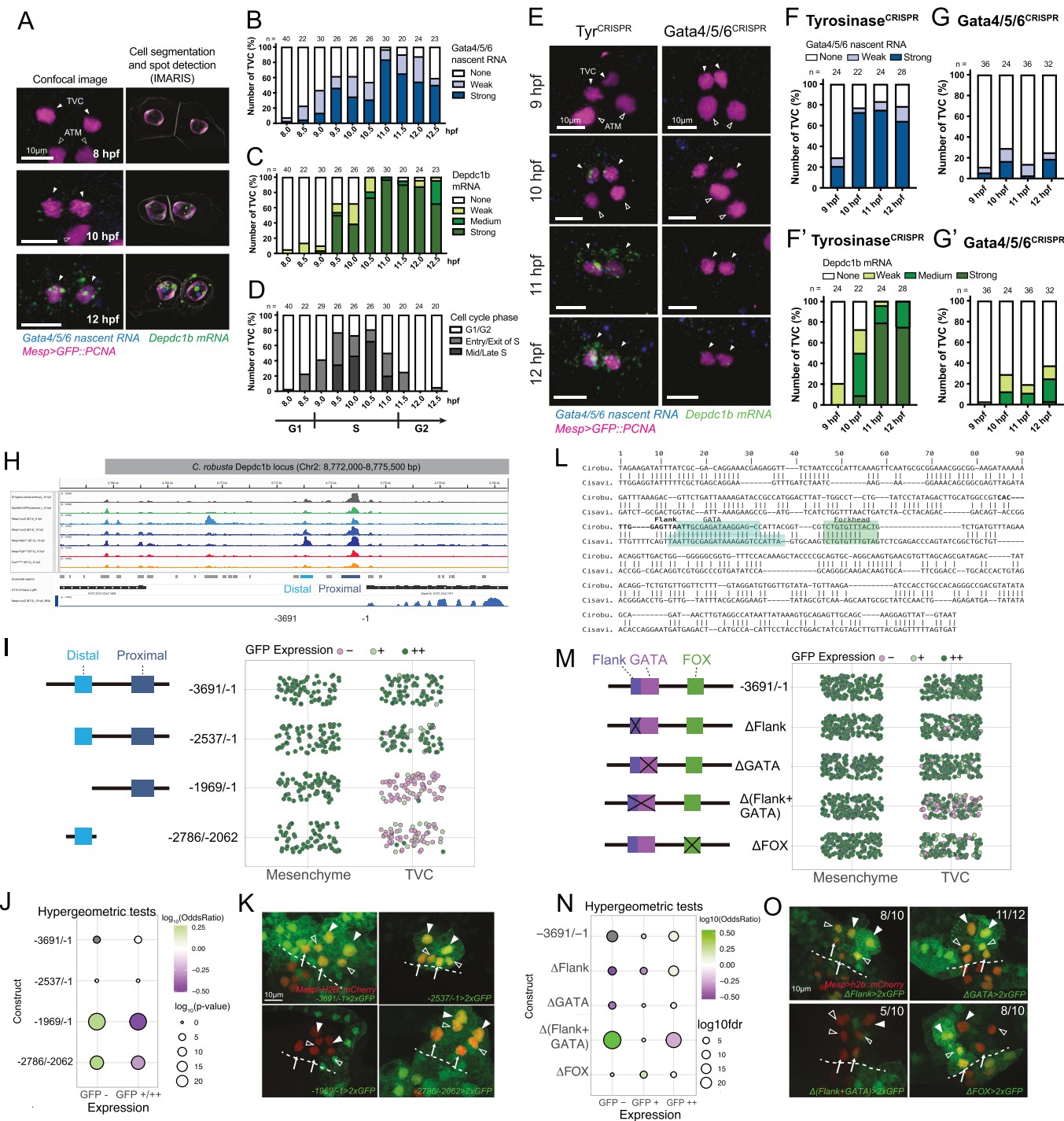

Among those, clustering, UMAP projection and marker gene expression identified 223 cardiopharyngeal progenitors and 326 anterior tail muscle, indicating that reporter-driven barcodes are effective to identify B7.5-lineage cells with >95% certainty at early stages (Appendix Fig. S21C–E). In silico demultiplexing of pooled scRNA-seq then allowed us to evaluate the cardiopharyngeal progenitor-specific effects of individual perturbations independently of cell sorting (Appendix Fig. S21F,G). *Foxf*, *Gata4/5/6*, and *Foxl1* were among the most significantly downregulated genes in

their corresponding CRISPR/Cas9 conditions (Fig. 7E; Appendix Fig. S21A,B), indicating effective perturbations. Both *Gata4/5/6* and *Foxtun4* were also downregulated in *Foxf*[CRISPR] cells, consistent with previous observations and the established role for Foxf in driving chromatin accessibility and gene activation in the cardiopharyngeal lineage (Fig. 7E; Appendix Fig. S22). Applying permissive statistical cutoffs (FDR < 0.05 and |Log$_2$(FC)| > 0.5), we identified 302 candidate differentially expressed genes in *Foxf*[CRISPR] compared to control cells, including 192 down- and 110 upregulated transcripts.

Figure 8. Gata4/5/6 regulates *Depdc1b* expression.

(A–C) Developmental dynamics of Gata4/5/6 gene expression. a. Representative confocal images showing double FISH detection of Gata4/5/6 and Depdc1b expression. Segmented cells, nuclei and transcripts shown on the right. Magenta: nuclei (GFP::PCNA); blue: Gata4/5/6 nascent RNA; green: Depdc1b mRNA. Scale bar = 5 μm; arrowhead: TVC; open arrowhead: ATM. (B, C) Semi-quantification of Gata4/5/6 (B) and Depdc1b (C) gene expression spanning TVC migration but prior to TVC division. (D) Cell cycle phases as determined by the GFP::PCNA reporter. (E) Confocal images of double FISH revealing *Gata4/5/6* and *Depdc1b* mRNAs in Tyr$^{CRISPR}$ control and Gata4/5/6$^{CRISPR}$ embryos. Scale bar = 10 μm Arrowhead: TVC; Open arrowhead: ATM. (F–F', G–G'). Semi-quantification of *Gata4/5/6* (F–F') and *Depdc1b* (G–G') expression in indicated conditions. (H) Accessibility of the *Depdc1b* locus during cardiopharyngeal development showing distal and proximal regulatory accessible regions upstream of the *Depdc1b* start site. Data from Racioppi et al, 2019. (I) Systematic deletion of distal and proximal regulatory regions upstream of a 2xGFP reporter. The diagram shows control and deletion constructs. The dot plot on the right shows the level of GFP expression detected in either the mesenchyme or the TVCs upon disruption of regulatory regions, separately or in combination. Jitter is added to the graph for ease of visualizing expression changes. (J) Hypergeometric tests of GFP expression based on regulatory region perturbation. Color scale indicates log$_{10}$ odds ratio, and size indicates log$_{10}$ P value. (K) Micrographs of GFP expression as driven by constructs containing both proximal and distal regulatory regions or lacking one or both regulatory regions. (L) Regions of the *Depdc1b* regulatory region showing conservation of the Flank region, and putative GATA and FOX-binding sites, between *Ciona robusta* and *Ciona savignyi*. (M) Constructs containing the Flank, GATA, and FOX binding sites driving 2xGFP expression and subsequent analysis of binding site requirements for the tissue specific expression of GFP. The dot plot on the right shows proportions of GFP+ detected in either the Mesenchyme or the TVCs upon disruption of regulatory regions, separately or in combination. Jitter was added to the graph for ease of visualizing expression changes. (N) Hypergeometric tests of GFP expression based on regulatory region perturbation. Color scale indicates log$_{10}$ odds ratio, and size indicates log$_{10}$ P value. (O) Micrographs of GFP expression as driven by constructs containing both proximal and distal regulatory regions or lacking one or both regulatory regions. For both (K, O), arrows – First Heart Precursors (FHPs), open arrowheads – Second Heart Precursors (SHPs), solid arrowheads Atrial Siphon Muscle Founder cells (ASMF). Source data are available online for this figure.

By contrast, *Gata4/5/6*$^{CRISPR}$ and *Foxtun4*$^{CRISPR}$ yielded more modest numbers, with 9 and 1 gene significantly differentially expressed, including *Gata4/5/6* and *Foxtun4* in their respective conditions (Appendix Fig. S22; Datasets EV5–7). Despite a modest genome-wide impact, the top candidate *Gata4/5/6* target was *Depdc1b*, the mature-state marker required for oriented asymmetric division (Appendix Fig. S22, see also Fig. 8 below).

Leveraging its more widespread impact on the cardiopharyngeal transcriptome (and chromatin accessibility (Racioppi et al, 2019)), we focused on the *Foxf*$^{CRISPR}$ condition to evaluate its role(s) in transcriptome maturation. First, we verified that the multiplexed scRNA-seq approach correlated with previous profiling of the same *Foxf*$^{CRISPR}$ perturbation by bulk RNA-seq (Racioppi et al, 2019) (Pearson's $\rho = 0.91$, $P < 2.2 \times 10^{-16}$; Appendix Fig. S22C). Next, we used the *Foxf*$^{CRISPR}$ vs. control volcano plot to visualize the individual responses of genes grouped by pseudotemporal clusters along the cardiopharyngeal progenitor trajectory (Fig. 7F). This suggested that genes in cluster 5, which comprises mature-state-specific markers (Fig. 7F,G), are mostly downregulated, while the intermediate gene cluster (cluster 2; Fig. 7F,G) splits between up- and downregulated genes. On the other hand, cluster 6, which comprises anterior tail muscle markers downregulated in cardiopharyngeal progenitors, comprised genes that were upregulated. Hypergeometric tests showed that clusters 2 and 5 were significantly over-represented among downregulated genes, while clusters 2 and 6 were over-represented among upregulated genes (Fig. 7F,G). Taken together, these observations indicate that the early cardiopharyngeal determinant Foxf, promotes multipotent progenitor maturation both by activating intermediate and mature-state markers and by downregulating other intermediate markers as well as the alternative anterior tail muscle program (see "Discussion").

Regulatory cascades and feed-forward circuits are classic and widespread gene regulatory network motifs accounting for the timed deployment of lineage-specific transcriptional programs in development (Davidson, 2010). The above and previous analyses placed Foxf atop the cardiopharyngeal regulatory hierarchy in Ciona, including *Gata4/5/6* as one of its notable target genes (Beh et al, 2007; Christiaen et al, 2008; Woznica et al, 2012; Racioppi et al, 2019) (Fig. 7F).

Intriguingly, the mature gene *Depdc1b*, which is necessary for proper oriented and asymmetric division of cardiopharyngeal progenitors (Figs. 5 and 6), was also the top downregulated gene in *Gata4/5/6*$^{CRISPR}$ cells, prompting us to further explore the regulatory relationship between the two genes. First, double FISH assays using embryos expressing the S-phase marker *Mesp > PCNA::GFP* indicated that *Gata4/5/6* expression precedes that of *Depdc1*, which started at the G1-S transition (Fig. 8A–D). Double FISH assays following CRISPR/Cas9-mediated loss of *Gata4/5/6* function corroborated the scRNA-seq data, whereby both *Gata4/5/6* and *Depdc1b* were strongly downregulated in *Gata4/5/6*$^{CRISPR}$ embryos (Fig. 8E–G'). These data indicated that the function of Gata4/5/6 is necessary for *Depdc1b* expression, which starts at the G1-S transition.

Leveraging our previous cardiopharyngeal lineage-specific chromatin accessibility data (Racioppi et al, 2019), we identified 2 main accessible regions located ~2.5 and 0.5 kbp upstream of the *Depdc1b* start site (Fig. 8H). Notably, the region located ~2.5 kb upstream displayed the typical ATAC-seq pattern of Foxf-dependent cardiopharyngeal progenitor-specific accessibility: the signal was higher in B7.5 lineage cells than whole embryos or mesenchyme lineage cells at 10 hpf, it increased between 6 and 10 hpf, and it was downregulated in conditions that inhibit cardiopharyngeal progenitor induction or Foxf function. To test the importance of these elements for *Depdc1b* regulation, we built a reporter construct by cloning a 3961 bp genomic DNA fragment upstream of a 2xGFP reporter, which we electroporated into fertilized eggs alongside a *Mesp > H2B::mCherry* lineage marker and assayed GFP expression at stage 25, shortly after the division of cardiopharyngeal progenitors (Fig. 8I, J). The original −3961/−1 construct, and its trimmed −2537/−1 version, showed similar GFP expression in the cardiopharyngeal lineage with $100 \pm 0\%$ ($n = 70$; ± S.E.) and $92.8 \pm 13.1\%$ ($n = 69$; ± S.E.) of the embryos, respectively (Hypergeometric test $P = 0.028$; Fig. 8J,K). By contrast, deletion of either the distal or proximal elements significantly reduced the proportions of embryos with detectable expression in the cardiopharyngeal lineage to $16.9 \pm 4.4\%$ ($n = 71$; ± S.E.; $P = 2.6 \times 10^{-21}$) and $35.9 \pm 6.0\%$ ($n = 64$; ± S.E.; $P = 1.3. \times 10^{-12}$), respectively (Fig. 8I–K). These results indicate that both distal and proximal elements are necessary for cardiopharyngeal expression of the *Depdc1b* reporter construct.

Next, we focused on the distal element, which showed cardiopharyngeal-specific and Foxf-dependent accessibility. This element proved to be highly conserved between *Ciona robusta* and a related species *Ciona savignyi*, with short blocks of identical sequences that typically correspond to putative transcription factor binding sites (Fig. 8L). Joint motif search using both sequences and the Cis-BP database and software (Lambert et al, 2019) identified several high-scoring putative binding sites for such cardiopharyngeal regulators as ETS, bHLH, Homeodomain, FOX, and GATA family factors (Appendix Fig. S23A). Focusing on the latter, we introduced microdeletions of conserved sequences corresponding to putative GATA and FOX binding sites, and a conserved region flanking the GATA motif in the original −3961/−1 construct, which contain a conserved TTAATT sequence matching homeodomain binding motifs, and assayed reporter gene expression (Fig. 8L–O). Of these, the joint "Flank + GATA" deletion caused a drop of cardiopharyngeal expression in electroporated embryos to 59.0 ± 4.2% ($n = 139$; ±S.E.) from 96.6 ± 1.5% ($n = 146$; ±S.E.) and 95.3 ± 1.5% ($n = 148$; ± S.E.) with constructs lacked either the "Flank" or GATA motif, respectively (Fig. 8L–O). Taken together, these data suggest that Gata4/5/6 activate *Depdc1b* expression in cardiopharyngeal progenitors, at least in part by directly binding to the evolutionarily conserved "Flank+GATA" motif located in a lineage-specifically accessible distal upstream element.

### Cell cycle progression promotes *Depdc1b* expression

Finally, several of the above observations suggested that cell cycle progression through interphase promotes progenitor maturation. Specifically, the mature state emerges during the late G2 phase, putative binding motifs for established cell cycle transcription factors, E2F and Myc, were enriched in accessible regions near dynamically expressed cardiopharyngeal lineage markers (clusters 2 and 5; Fig. 7A), the corresponding transcription factor-coding genes are dynamically expressed in cardiopharyngeal progenitor (Fig. 7D), and the *Depdc1b* proximal promoter region contains several GC-rich putative E2F binding sequences (Lambert et al, 2019) (Appendix Fig. S23B,C). Marked dynamics in transcriptome composition also coincide with cell cycle phase transitions, especially G1-S, which correlates with the onset of both collective migration and *Depdc1b* expression (Figs. 1–3). We inhibited the G1-S transition by overexpressing Cdkn1b (Cdkn1b$^{OE}$) in the cardiopharyngeal progenitors using the minimal *Foxf* enhancer as previously described (Razy-Krajka et al, 2018) (Fig. 9A). We assayed the Cdkn1b$^{OE}$ phenotypes using the S-phase reporter Mesp>GFP::PCNA and double FISH to detect *Gata4/5/6* and *Depdc1b* transcripts (Fig. 9B). As expected, Cdkn1b$^{OE}$ significantly reduced the proportions of cardiopharyngeal progenitors entering S-phase during the 9 to 12 hpf window observed in controls (Fig. 9C,D). This perturbation did not alter the onset of *Gata4/5/6* expression, which remained detectable as early as 9 hpf, even though it did lower its highest level (Fig. 9B,C',D'). On the other hand, Cdkn1b$^{OE}$ markedly inhibited *Depdc1b* activation, as well as its accumulation toward the end of interphase (Fig. 9B,C",D"). Together, these observations provide direct experimental support to the notion that progression through G1-S promotes transcriptome maturation in cardiopharyngeal progenitors as exemplified by the impact of Cdkn1b$^{OE}$ on the activation of *Depdc1b*, a mature state marker required for proper oriented and asymmetric division of multipotent progenitors.

## Discussion

As development proceeds, cell identities are canalized through progressive fate restriction of multipotent progenitors. Focusing on multipotent cardiopharyngeal progenitors in the tunicate Ciona, we explored the acquisition of multilineage competence and the coupling of progressive fate specification with cell cycle progression. Combining defined perturbations of cell cycle progression with gene expression analysis, we first provided evidence that mitosis is required but not sufficient for *Tbx1/10* activation, an essential step in the heart vs. pharyngeal muscle fate choice (Razy-Krajka et al, 2018; Wang et al, 2019, 2013; Song et al, 2022). Using sample multiplexing and scRNA-seq to profile a time-series of FAC-sorted cardiopharyngeal progenitors, we uncovered transcriptome dynamics characterized by transitions through regulatory states during interphase, which we refer to as maturation. We identified gene clusters representing defined (pseudo)temporal profiles in the maturation trajectory. The S-phase marker PCNA::GFP revealed a correspondence between the G1-S and S-G2 transitions and the maturation of multipotent progenitors, which reach a mature state characterized by the downregulation of early and intermediate markers and peak expression of late/mature genes in late G2. Using loss-of-function by CRISPR/Cas9-mediated mutagenesis, we showed that most mature genes are necessary for proper oriented and asymmetric division of multipotent progenitors, and subsequent polarized activation of *Tbx1/10*. Focusing on the RhoGAP-coding gene *Depdc1b*, which is activated at the G1-S transition and peaks in G2, we uncovered a role in the orientation and asymmetry of cell division, which impacts the ability of multipotent progenitors to generate two distinct, *Tbx1/10*-positive and negative, daughter cells along the medio-lateral axis. These results support the notion that multipotent cardiopharyngeal progenitors need to mature to acquire the competence to divide in an oriented and asymmetric fashion, and produce both a large lateral *Tbx1/10+* second cardiopharyngeal progenitor and a small median first heart precursor. Finally, we present evidence that transcriptome maturation relies, at least, on classic gene regulatory network motifs such as forward regulatory cascades, as well as coupling with cell cycle progression, especially the G1-S transition (Fig. 10).

Classic concepts in developmental biology, dating from homo- and heterotopic and -chronic grafting experiments by gifted embryologists, suggested that progenitor cells transition through specification and commitment to restricted identities, while the notion of maturation has been considered extensively for differentiated cells, such as cardiomyocytes, which are contractile but not fully functional until they reach a mature state. Here, we adopted a general definition of the concept of maturation (Alvarez-Dominguez and Melton, 2022), considering a biological entity that persists through time and progressively acquires its full functionality through underlying changes. From that standpoint, since multipotent cardiopharyngeal progenitors transition through successive regulatory states as they progress through interphase, and ultimately acquire the competence to divide in an oriented and asymmetric manner and produce distinct first heart vs. *Tbx1/10+* second craniopharyngeal progenitors, we argue their transcriptome maturation fosters multilineage competence.

To generalize the notion of multipotent progenitor maturation, we leveraged our transgene-based barcoding strategy for

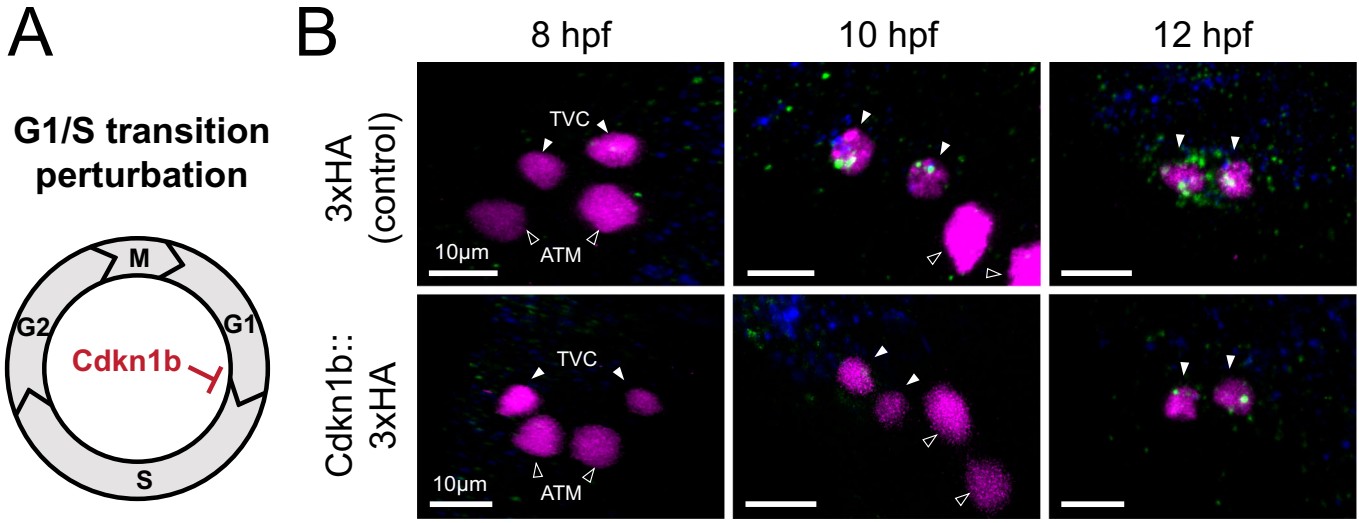

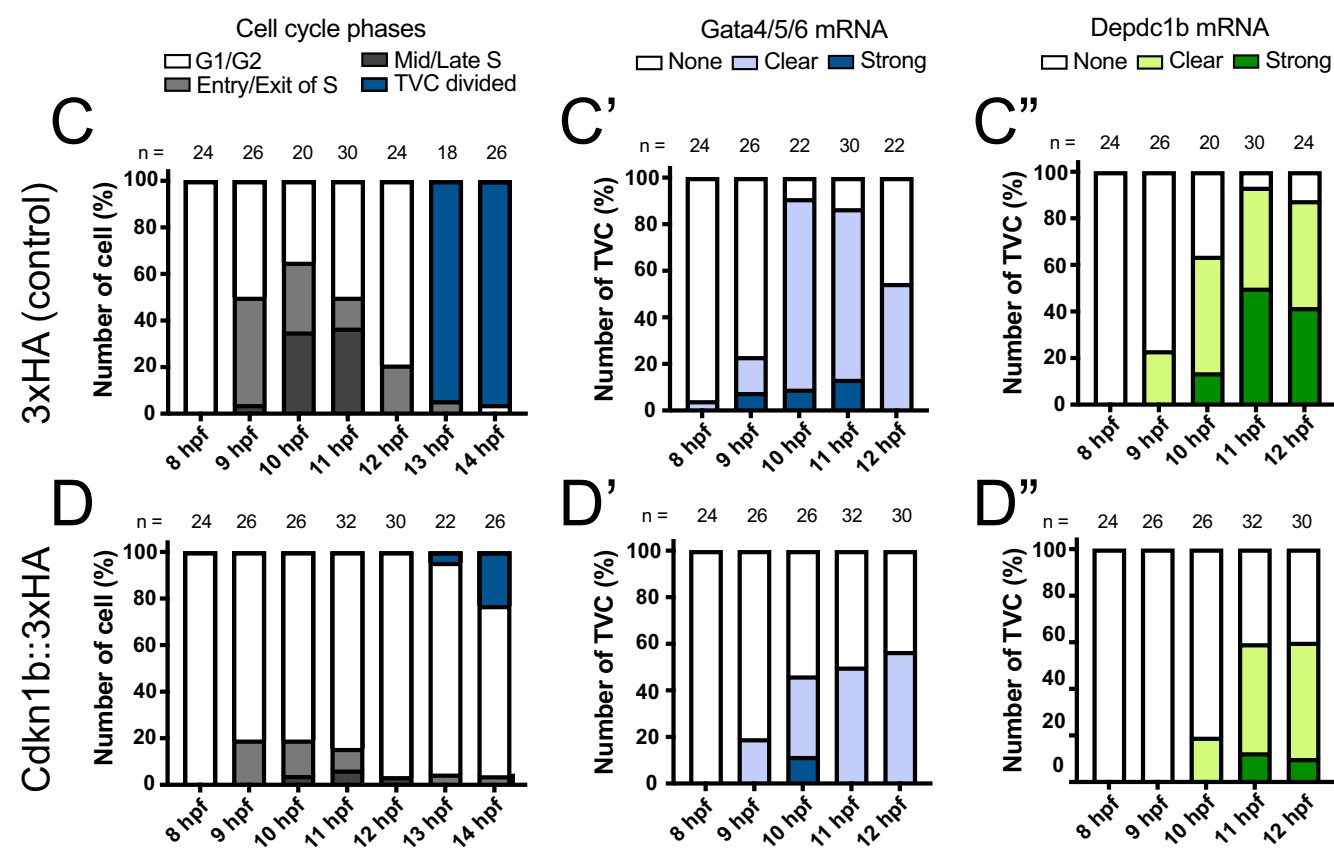

**Figure 9.  Cell cycle progression drives sub-circuits that promote multipotent progenitor maturation.**

(A) Schematic of cell cycle stages and genetic perturbation of G1-S transition. (B) Representative confocal images of Gata4/5/6 and Depdc1b expression at 8, 10, and 12 hpf in control and Cdkn1b::3xHAOE embryos. Magenta: nuclei (GFP::PCNA); blue: Gata4/5/6 mRNA; green: Depdc1b mRNA. Scale bar = 10 µm; arrowhead: TVC; open arrowhead: ATM. (C, D) Dynamic expression of *Gata4/5/6*, *Depdc1b* expression, and S phase progression in control and Cdkn1b-overexpressing cardiopharyngeal progenitors. No blinding was included in the analysis. Source data are available online for this figure.

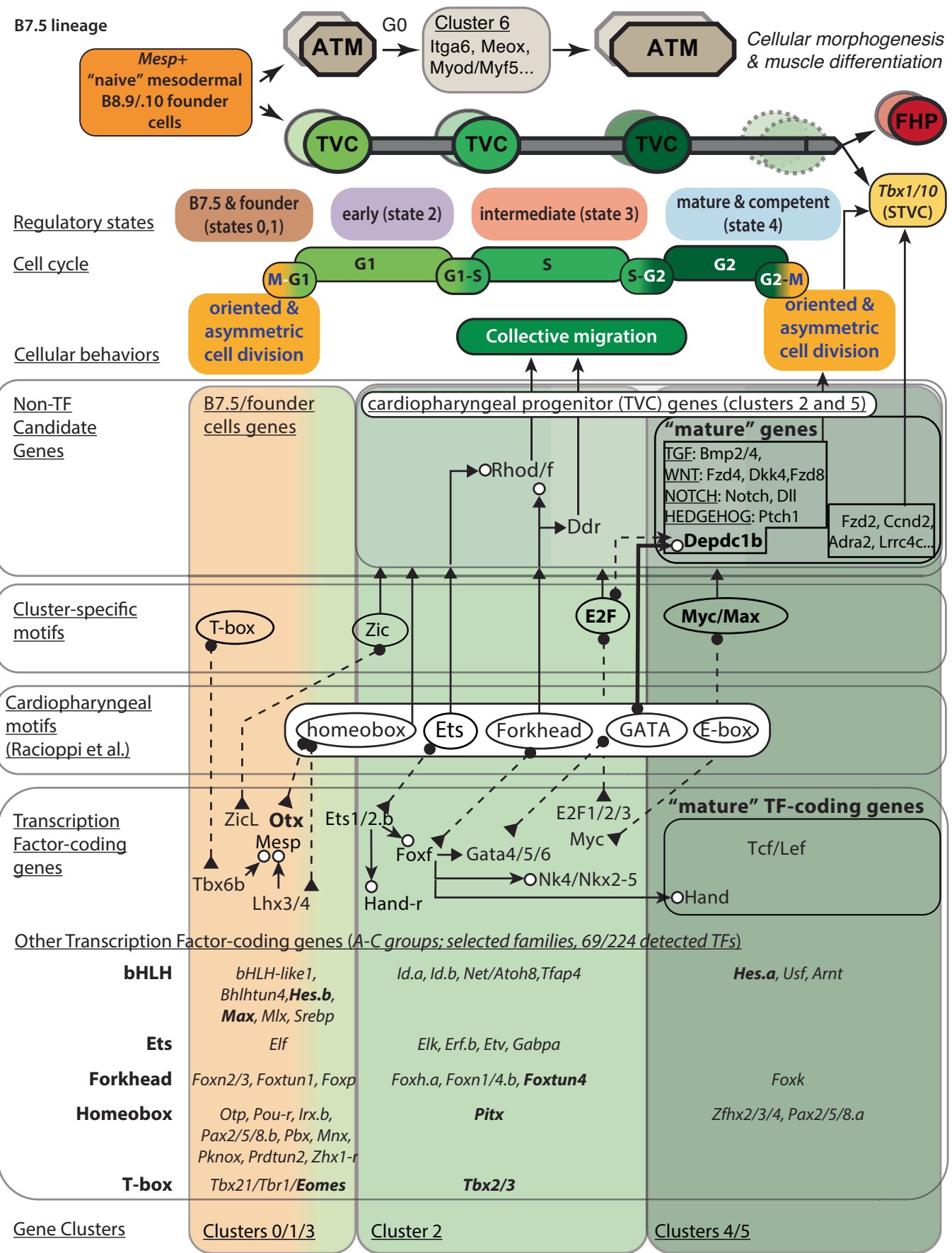

**Figure 10. Provisional summary model.**

The interplay between cardiopharyngeal regulatory cascades and cell cycle progression promotes transitions between regulatory states marked by the relative expression of 5 gene clusters. This progression is referred to as maturation toward a regulatory state that is competent for subsequent asymmetric and oriented division and polarized activation of *Tbx1/10*, specifically in the second generation of multipotent cardiopharyngeal progenitors (a.k.a. second trunk ventral cells, STVC). Cardiopharyngeal transcription factors are indicated as regulating lineage-specific markers from clusters 2, 4, and 5, including candidate mature genes, such as *Depdc1b*, which impact cell division orientation, asymmetry and/or *Tbx1/10* activation. The expression of E2F and Myc transcription factors and enrichment of their cognate binding sequences near gen clusters 2 and 5 are consistent with a direct role for cell cycle progression in transcriptome maturation. Transcription regulators and cell migration effector genes identified in previous studies are indicated. Candidate regulatory motifs identified in this study and in Racioppi et al, 2019(Racioppi et al, 2019) are indicated. Selected candidate transcription factor-coding genes belonging to molecular families consistent with enriched motifs are indicated, illustrating the need for future extensive functional assays. ATM anterior tail muscle, TVC trunk ventral cell, STVC second TVC, FHP first heart precursor.

multiplexed scRNA-seq, and profiled whole embryos from gastrula to pre-hatching larval stages in one experiment. With ~22,000 single-cell transcriptomes, we recovered the same main lineages and identities as previous studies (Cao et al, 2019a; Winkley et al, 2021), but with reduced batch effects. Our barcoded time-stamps unequivocally identified developmental trajectories that followed temporal sequences. For instance, the transcriptional signatures of early lineages appeared overridden by patterning signatures in the ectoderm-derived epidermis and nervous system. By contrast, endomesodermal lineage displayed more conspicuous signs of clonally guided developmental progression, due to the persistence of transcriptional signatures of their clonal origins. Through subsetting and reclustering, we characterized specific endomesodermal trajectories, which we interpreted as multipotent progenitors maturing before splitting into distinct identities, and identified corresponding candidate mature gene clusters.

Extending beyond ascidian embryogenesis, whole animal profiling of developmental trajectories during vertebrate embryogenesis has repeatedly uncovered similar patterns of multipotent progenitors seemingly maturing along linear trajectories before producing distinct cell identities, presumably following cell divisions (Qiu et al, 2022a; Farrell et al, 2018; Briggs et al, 2018). As we observed by comparing distinct trajectories within the ascidian embryo, gene expression signatures of maturing multipotent progenitors tend to be lineage and/or fate-specific. For instance, multilineage transcriptional priming is a hallmark of multipotent progenitor transcriptomes, and we observed that primed heart vs. pharyngeal muscle-specific genes are enriched in mature gene clusters. However, multilineage transcriptional priming started prior to the mature state as shown by another enrichment of primed genes in the intermediate gene cluster (cluster 2 in Fig. 2), indicating that priming is completed through maturation, consistent with the notion that maturation corresponds the acquisition of full functionality. This indicates that multilineage transcriptional priming is one of the molecular system's features that is acquired at least in part through transcriptome maturation.

Consistent with marked lineage-specific components to transcriptome maturation, classic gene regulatory network motifs contribute to progenitor maturation. Specifically, the extensive roles of key cardiopharyngeal determinants Foxf and Gata4/5/6 corroborate previous studies, as does the observed sequence of gene activation for primed cardiac determinants *Gata4/5/6*, *Nk4/Nkx2.5*, and *Hand1/2*, which also depends on chromatin priming through enhancer opening by Foxf (Racioppi et al, 2019). There is extensive evidence for such "regulatory competence", whereby chromatin and transcriptional mechanisms determine multilineage competence, in

a variety of developmental systems (Jain and Epstein, 2018; Zaret et al, 2008). Here, we showed that Gata4/5/6 also regulates the timed expression of the "mature gene" *Depdc1b*, which encodes a Rho GAP necessary for proper asymmetric and oriented division. Careful analysis of the CRISPR/Cas9-induced phenotypes suggested that Depdc1b controls primarily division orientation and asymmetry, which in turn impacts contact with the surrounding mesenchyme, and the pattern of *Tbx1/10* expression. This interpretation is consistent with the known roles of the Depdc1b homolog in *C. elegans*, LET-99, which controls the position and orientation of the mitotic spindle during the first embryonic division (Rose and Kemphues, 1998; Tsou et al, 2002). Taken together, these results indicate that transcriptome maturation also endows multipotent progenitors with "behavioral competence", i.e., the cytoskeletal/cellular ability to divide in a manner compatible with the emergence of two distinct cell identities marked by differential *Tbx1/10* expression. Future work will uncover the role(s) of the lateral mesenchyme as a cardiopharyngeal niche (Christiaen lab., unpublished observations).

Our results indicate that cell cycle progression influences cardiopharyngeal fate specification in at least two distinguishable ways: *Tbx1/10* activation shortly follows mitosis, which is necessary, but not sufficient as multipotent progenitors themselves need to progress through G1, S and G2 to reach a mature state competent to express *Tbx1/10*. As reported above, the competence for polarized *Tbx1/10* activation requires oriented asymmetric division, but certain mature genes, such as the GPCR-coding genes *Adra2b* and *Lrrc4c* appeared to impact *Tbx1/10* without altering the division pattern. Given the short estimated delay between mitosis and *Tbx1/10* activation ( < 30 min), we anticipate gene activation to occur in G1. This is reminiscent of a CDK-independent and nuclear role for late G1 cyclin Ds in activating *TBX1* in human pluripotent stem cells (Pauklin et al, 2016), and of the general propensity of the G1 phase to promote fate choices, for example, during stem cell specification (Dalton, 2013; Pauklin and Vallier, 2013). On the other hand, a high-throughput RNAi screen identified G2 phase regulators as promoters of the undifferentiated state in mammalian stem cells (Gonzales et al, 2015). Here, we propose that the transcriptome changes that unfold in S and G2 promote maturation of the multipotent progenitors, and their competence to produce distinct cardiopharyngeal cell identities. From this standpoint, we propose a multitier coupling between cell cycle progression and progressive fate specification that unites the two views, since maturation completed the multilineage priming of progenitors, and argue that progression through G2 and maturation poise the cells to choose a cardiopharyngeal identity in subsequent divisions while preserving multipotency, and the competence to choose either fate. Finally, the tight coupling between cell cycle progression and fateful

gene expression changes observed in ascidian lineages might represent an extreme case, in keeping with their highly stereotyped embryogenesis (Guignard et al, 2020; Lemaire, 2009). Work in *Drosophila* has also begun to uncover the relationship between mesodermal fate decisions, cell cycle progression, and collective migration (Sun et al, 2024). By contrast, the more relaxed coupling observed in vertebrates, together with signs of progenitor maturation along developmental trajectories, suggests that—just like genetic determinism manifests at the tissue and organ scales—the acquisition of multilineage competence occurs through "lineage maturation" as multipotent progenitors change through rounds of cell divisions.

## Methods

### Reagents and tools table

| Reagent/resource | Reference or source | Identifier or catalog number |
|---|---|---|
| **Experimental models** | | |
| *Ciona robusta* | M-Rep | |
| **Recombinant DNA** | | |
| Mesp>GFP | Davidson et al, 2005 | |
| CRISPR Library (111 plasmids, seq available as part of supplemental tables) | This study | Dataset EV3 |
| 2mU6>sgRNA(F + E) | Stolfi et al, 2014 | |
| Mesp>nls::Cas9 | Stolfi et al, 2014 | |
| Mesp>H2B::mCherry | | |
| Mesp>hCD4::EGFP | Gline et al, 2015 | |
| Mesp>GFP::PCNA | This study | |
| Mesp>GFP + barcodes library | This study | |
| Mesp>hCD4::mCherry | Gline et al, 2015 | |
| Mesp>nls::LacZ | Davidson et al, 2005 | |
| Twist>hCD4::mCherry | Gline et al, 2015 | |
| Depdc1b GFP reporter expression constructs (Fig. 8) | This study | |
| Foxf(TVC)>hCD4::mCherry | Gline et al, 2015 | |
| Foxf(TVC) > GFP::PCNA | This study | |
| Foxf(TVC)>Cdc25 | This study | |
| **Antibodies** | | |
| Alexa-488-conjugated Rat anti-GFP antibody | BioLegend | FM264G |
| anti-dpERK Mouse monoclonal antibody | Sigma-Aldrich | M9692 |
| Anti-Digoxigenin-POD Fab fragment | Roche | 11093274910 |
| Anti-GFP chicken antibody | Aves Lab | GFP-1010 |
| Anti-mCherry rabbit polyclonal antibody | BioVision | 5993-100 |

| Reagent/resource | Reference or source | Identifier or catalog number |
|---|---|---|
| Anti-β-Galactosidase mAb | Promega | Z3781 |
| Donkey-anti-chicken-555 | Invitrogen | A78949 |
| Goat-anti-rabbit-405 | Invitrogen | A-31556 |
| **Oligonucleotides and other sequence-based reagents** | | |
| CRISPR guide RNAs | This study | Dataset EV3 |
| Primer: U6-F-910 | This study | *caattgccccaagctctcttc* |
| Primer: seqU6-F | This study | *ggatcgcgcgagccc* |
| **Chemicals, enzymes, and other reagents** | | |
| Bsal | New England Biolabs | R3733S |
| Instant-Stickyend Ligase | New England Biolabs | M0370S |
| *Cirobu.Tbx1/10* cDNA antisense RNA probe | Wang et al, 2013 | |
| *Cirobu.Tbx1/10* intronic antisense RNA probe | Wang et al, 2013 | |
| *Cirobu.Ptch1* antisense RNA probe | This study | |
| *Cirobu.Bmper* antisense RNA probe | This study | |
| *Cirobu.Depdc1b* antisense RNA probe | This study | |
| *Cirobu.Gata4/5/6* intronic antisense RNA probes | This study | |
| **Software** | | |
| TCS SP8 X Leica | https://www.leica-microsystems.com/products/confocal-microscopes/p/leica-tcs-sp8-x/downloads/ | |
| Bitplane Imaris | https://imaris.oxinst.com/ | |
| R | R Core Team (2023) R: A Language and Environment for Statistical Computing. R Foundation for Statistical Computing, Vienna, Austria. URL https://www.R-project.org/. | |
| Rstudio | Posit team (2025) RStudio: Integrated Development Environment for R. Posit Software, PBC, Boston, MA. URL http://www.posit.co/. | |
| Python | Python Software Foundation (2023) Python (Version 3.x). Available at https://www.python.org/. | |
| **Other** | | |
| NucleoBond Xtra Midi kit | Macherey-Nagel | 740410.50 |
| 10X Genomics Chromium Universal 3′ gene expression | https://www.10xgenomics.com/products/universal-three-prime-gene-expression | |

## Animals

Animal care and experiments in this study were conducted in accordance with the Animal Welfare Guidelines of the NIH and the Laboratory Animal Guidelines of China. Adult *Ciona robusta* were purchased from M-Rep, USA or collected from coastal waters in Rongcheng, China, maintained in artificial seawater with constant lighting, and used for experiments within one week of arrival.

## Sample barcoding (SBC) plasmids

To generate reporter transcripts with unique tags detectable by the 10X Chromium Single Cell 3' Gene Expression Profiling System, 9 bp-short tags were inserted into *Mesp > GFP* construct 276 bp after the stop code of the EGFP coding region, between *Kpn*I and *Xcm*I restriction sites. The position of the tags on the 3'UTR of the transcripts was optimized based on previous RNA-seq data to ensure effective detection by RNA sequencing.

## Sample multiplexing

Samples from different developmental stages or CRISPR knock-out conditions were electroporated with a pair of sample barcoding (SBC) constructs as a unique label for multiplexing. Detection of SBC pairs (as designed) is the basis for determining the origin of each cell. To minimize variability between samples, we collected eggs for all samples into a large pool obtained from multiple animals, split the pool and proceeded to fertilize, dechorionate, and electroporate the paired SBCs individually at one-hour intervals according to the experimental design. In this study, ten batches of embryos of 5–14 hpf were individually dissociated, and cell suspensions were methanol-fixed and pooled for storage at −80 °C. *Cirobu.Mesp* enhancer-driven SBC reporters were used for barcoding and FACS enrichment of cardiopharyngeal lineage cells, while the universally expressing *Cirobu.Eef1a* enhancer-driven SBC reporters were used to multiplex whole embryo time series samples. The sample was then rehydrated, immunolabeled with a fluorescent anti-GFP antibody for FAC-sorting of B7.5 lineage cells prior to loading onto a 10X Genomics Chromium controller.

## Single-cell suspension and fixation

For both TVC trajectory and whole embryo datasets, single cell suspensions were prepared as described (Chen et al, 2018). Embryos and larvae at the desired developmental stages and experimental conditions were collected in 5 mL borosilicate glass tubes (Fisher Scientific, catalog no. 14-961-26) and washed with 2 mL Calcium- and Magnesium-free artificial seawater (CMF-ASW: 449 mM NaCl, 33 mM Na$_2$SO$_4$, 9 mM KCl, 2.15 mM NaHCO$_3$, 10 mM Tris-Cl at pH 8.2, 2.5 mM EGTA). Embryos and larvae were dissociated in 2 mL of 0.2% trypsin (w/v, Sigma, T-4799) CMF-ASW by pipetting with glass Pasteur pipettes. Dissociation was stopped by adding 2 mL of filtered ice-cold 0.05% BSA CMF-ASW. Dissociated cells were passed through a 40 μm cell strainer and collected in 5 mL round-bottomed polystyrene tubes (Corning Life Sciences, ref. 352235). Cell suspensions were transferred to 2.0 mL LoBind Tube (Eppendorf, Cat. No.022431102) and collected by centrifugation at 800× *g* for 3 min at 4 °C, followed by two washes with ice-cold 0.05% BSA CMF-ASW. After dissociation, cell suspensions were again concentrated into a pellet at 800 g for 3 min at 4 °C, and most of the supernatant was

removed, leaving over 100 μL. The pellet was then thoroughly resuspended by gentle pipetting. Prechilled 900 μL −20 °C methanol was added dropwise to the cell suspension while gently tapping the tube to avoid aggregation. When all 900 μL of methanol was added, the tubes were thoroughly mixed by inversion three times and placed on ice for 30 min (inversions of 10 min each). Cell suspensions can be stored at −80 °C for weeks without a significant change in RNA integrity.

## Cell rehydration and immunostaining for FACS

Fixed cell suspensions were removed from the −80 °C freezer and allowed to settle on ice for 10 min. Cells were harvested by centrifugation at 1500× *g* for 5 min at 4 °C to minimize cell loss, followed by two washes with ice-cold 3X SSC high BSA rehydration cocktail (SSC: 3×, BSA, 0.25%, DTT, 40 mM). The cell suspensions were then concentrated to a pellet at 800× *g* for 3 min at 4 °C. After removing 300 μL of supernatant, the cell pellets were carefully and gently resuspended in the remaining 200 μl of 3× SSC high BSA rehydration cocktail. Alexa-488-conjugated GFP antibody (BioLegend, FM264G) was added to the cell suspensions at 5 μL to 200 μL dilution, and then incubated at 4 °C with gentle agitation for 2 h. After incubation, cells were retrieved by centrifugation at 800 relative centrifugal force (rcf) for 5 min at 4 °C and then resuspended with 500 μL of ice-cold 3× SSC high BSA rehydration cocktail, followed by gentle agitation for 10 min. The cells were washed three times with 3× SSC low BSA rehydration cocktail (SSC: 3×, BSA, 0.05%, DTT, 40 mM supplemented with 0.2 U/μL RNAse inhibitor) and used for Alexa 488 positive sorting within 1 h.

## Whole embryo barcoding

Hashtag barcoding antibody-oligos (HTOs) were conjugated to the mouse anti-dpERK monoclonal antibody (Sigma-Aldrich, M9692) using the CITE-seq hyper antibody-oligo conjugation protocol (https://cite-seq.com/), and the antibody was incubated with cell suspensions in the whole embryo study using the same immunostaining procedure as for the Alexa-488-conjugated GFP antibody. This addition did not alter the quality of the scRNA-seq data; however, the information obtained from the mouse anti-dpERK monoclonal antibody is not discussed in this study.

## 10X Single Cell 3' Gene Expression library preparation and sequencing

Rehydrated cell suspensions from whole embryos or FACS-purified B7.5 lineage were collected at 800 relative centrifugal force (rcf) for 5 min at 4 °C. The supernatant was removed as much as possible before adding 33.4 μL RT mix + 46.6 μL water = 80 μL to the bottom of the tube. Gently pipette ~30 times to ensure cells are thoroughly resuspended. The 10X Single Cell 3' Gene Expression Library was prepared according to the 10X protocol except that 1 μL 0.2 μM of SBC additive primer (5'-ttgccgctatttctctgggtacc-3') was added to the 10X cDNA amplification mix to generate a separate SBC library in parallel with the gene expression library. The 3' gene expression library was purified using the standard 10X protocol, while the SBC library was purified using two rounds of 2× SPRI from the supernatant of the 0.6× SPRI 3' gene expression library purification step.

## Single-cell sequencing data preprocessing

Twenty SBC barcodes, including the flanking sequences, were added as pseudogenes to the *Ciona robusta* reference genome (http://ghost.zool.kyoto-u.ac.jp/download_ht.html and Dataset EV1). Raw sequencing data were mapped to the reference genome using 10X Genomics' Cell Ranger (Version 3.0.1) pipeline to generate UMI count matrix for the downstream analysis. An SBC count matrix was generated individually using Bowtie.

## RNA velocity

In order to run RNA velocities, fastq data were aligned to the KY genome (Satou et al, 2019) of *Ciona robusta* using cell-ranger version 9.1.0 [10x Genomics]. The aligned reads were then converted to a loom file containing spliced and unspliced reads using velocyto (La Manno et al, 2018) (version 0.17. Dynamo (Qiu et al, 2022b) [version 1.4.2] was used to generate RNA velocities using the deterministic mode based on spliced/unspliced counts. As in the case for some other datasets, the predicted flow of RNA velocities was partially inverted on the trajectory, which we suspect this to be at least partially due to the fact that Ciona has a compact genome with many single-exon genes. Only 6% of the reads were classified as unspliced.

Therefore, we used the option to calculate RNA velocities directly from the denoised gene expression matrix and the diffusion pseudotime calculated previously. Finally, we made use of the vector field calculation tools of Dynamo to extract the speed of the RNA velocity calculated using either approach.

## Whole embryo scRNA-seq data analysis

### Imputation of developmental time

We imputed the developmental time of cells lacking a barcode-derived developmental age by using an approach inspired by ancestry voting. The dataset was split into cells with and without timestamps based on electroporated barcodes. The twenty nearest neighbors for each cell without a timestamp were identified. The inferred developmental time of a time stamp was set to the median of the timestamps of its k-nearest neighbors. Only in cases where the standard deviation of the developmental time of the k-nearest neighbors was below 1 did we assign an inferred time stamp to a cell. Using all cells with a timestamp as a ground truth and performing this label transfer approach for each cell with timestamp individually, we estimate that we infer the correct transcriptional age in 80.4% of the cases and are within 1 h of development in 95.9% of cells. Thus, only 4.1% of the cells might have gotten a severely wrong inferred timepoint, in addition to the 20.3% of cells that did not get any time label assigned. For the final developmental time, we first considered if a cell had a barcode-derived time stamp, in case of differences between the inferred and barcode-derived developmental time, the barcode-derived developmental time was retained.

### Mesenchymal trajectories

To obtain developmental trajectories of individual endodermal and mesenchymal lineages, we subclustered the whole mesenchyme and endoderm independently and grouped the resulting higher-resolution Leiden clusters by expression of marker genes and

continuity of the UMAP projection. We constructed a UMAP for each isolated branch and calculated a branch-specific pseudotime using Scanpy, setting a random cell with the lowest barcode-derived developmental time stamp as root cell.

### Gene clustering
Gene clusters were calculated by transposing the gene count matrix, filtering out genes that were expressed in less than three cells in the derived matrix, and performing dimensional reduction and Leiden clustering on the transposed matrix, similar to the standard procedure in the identification of cell clusters. Mature gene clusters were defined as clusters of genes that were upregulated shortly before and around the time of lineage bifurcation, while immature gene clusters were defined as the first gene cluster that was upregulated in the founder cell population.

### Marker gene identification
A Wilcoxon rank sum test was used to identify marker genes for different cell states. To enrich for significant marker genes, we filtered the putative markers for an average $\log_2$(FC) of 0.25 and an adjusted p-value < 0.1.

## sgRNA design and library construction

### CRISPR library construction
All sgRNAs were cloned into the *2mU6>sgRNA(F + E)* scaffold as per Stolfi et al (Stolfi et al, 2014) *2mU6>sgRNA(F + E)* plasmids were digested with BsaI-HF (NEB R3535, discontinued) in rCutsmart buffer for optimal positivity rates. Oligos ordered from Sigma at 50 mM concentration in water, were annealed by combining 10 μL of forward and reverse oligos and incubating in a thermocycler with the following protocol: (1) 95 °C for 30 s (melt), (2) 72 °C for 2 min, (3) 37 °C for 2 min, (4) 25 °C for 2 min. Annealed oligos were diluted 1:50 in TE buffer, and ligated into an opened vector using Instant Sticky-end Ligase Mix (NEB M0370S) with 0.5 μL vector, 2.5 μL diluted oligo, 3 μL ligase mix. 3 μL of ligation mix were transformed into 15 μL Stellar Competent cells (Takara 636763) and plated on AMP LB selection plates followed by overnight incubation at 37 °C. Four colonies per plate were selected for colony PCR using the U6-F-910 forward primer (5'-caattgccccaagctctcttc-3') and the guide RNA-specific sgRNA-R oligo (same as the R for annealing diluted to 10 μM). Positive colonies were grown overnight in 4 mL LB amp, miniprepped, and sent for sequencing using the seqU6-F primer (5'-ggatcgcgcgagccc-3'). Sequence results were checked to assure no mutations were present in the guide RNA sequence and correct clones were frozen down in Hogness Freezing Medium (Bioworld 30629174-1) and stored at −70 °C.

## CRISPR screen data collection and analysis

For each gene in the screen, the following electroporation conditions were used: Mesp>NLS::LacZ (nuclear marker) - 20 μg, Mesp>hCD4::mCherry (cell membrane marker) - 20 μg, Mesp>NLS::-Cas9::NLS - 25 μg, 2mU6>sgRNA.1.2.3 - 80 μg. The total DNA amount used for electroporation did not exceed 150 μg. Embryos were incubated at 18 °C for 15 h post fertilization (hpf). At 15 hpf, embryos (approx. FABA stage 25) were fixed in 4% paraformaldehyde and used for fluorescent in situ hybridization as described (Bernadskaya et al,

2019) using an intronic *Tbx1/10* digoxigenin-labeled RNA probe (Wang et al, 2013). Three biological replicates were performed for each condition. 3D stacks were collected on a Leica SP8 Confocal microscope equipped with a white light laser and using a ×63 oil immersion lens. No blinding was included in the subsequent analysis. Images were analyzed manually using Imaris Bitplane software for the presence of 2, 1, or absence of *Tbx1/10* nascent transcripts per cell, qualitative cell division orientation along the anterior-posterior and medio-lateral axes of the embryo (AP or ML), and for asymmetric cell division as qualitatively determined by cell size post-division (A, S, or P for precocious divisions). In total >1500 embryo halves were analyzed. Graphs and statistical analysis performed in Python using Matplotlib and Seaborn pipelines. Fisher's exact test was used to detect significant changes in the three categories with an FDR with a cutoff of 0.05 to determine the rate of Type I errors.

## Depdc enhancers characterization

*Depdc1b* enhancers were predicted based on the TVC-specific chromatin accessibility identified by Racioppi et al, (Racioppi et al, 2019). Three versions of the enhancers (3691/−1, −2537/−1 and −1969/−1) including the native promoter of the *Depdc1b-KY.Chr2.2230* gene were amplified using the following primers (5′-tccggcgcgccGTATTGGGGAACCCGGGATAAAAATATTGAAACGTC-3′ and 5′-accgcggccgcTCTTATGTGTTTGATAAACCTGTTAAACGAAAAGC-3′ for -3691/-1, 5′-ttaggcgcgccAACAATATTGTCCTGGTTTGACTTTC-3′ and 5′-accgcggccgcTCTTATGTGTTTTGATAAACCTGTTAAACGAAAAGC-3′ for -2537/-1, 5′-ggcggcgcgccTGCTACTTATAGGAAGAAATATATAGTTTTTG-3′ and 5′-accgcggccgcTCTTATGTGTTTGATAAACCTGTTAAACGAAAAGC-3′ for -1969/-1) with the addition of AscI and NotI restriction sites. The enhancers were subcloned into the AscI-NotI restriction site of the 2xGFP vector. The enhancer (-2786/-2062) was amplified using the following primers (5′-tccggcgcgccCATTATTACGACTCTAAAAGAGCGCTGATAATTGTTAAACGC-3′ and 5′-ggatctagaTCTGCTATATACGTCGGCCCTGTGGCAGGTGGATTCTTAACAT-3′) cloned into the AscI-XbaI restriction site of bpfog: :2xGFP vector. Using the −3691/−1 > 2xGFP reporter as a template, ΔFlank, ΔGata4/5/6, Δ(Flank+Gata4/5/6), and ΔFoxa1/2/3 were amplified using the Mut Express II Fast Mutagenesis Kit V2 (Vazyme, Cat. No. C214-01) with the following primers (5′-TTGCATGGCCGTGCGAGATAAGGAGCCATTACGG-3′ and 5′-CTCGCACGGCCATGCAAGTCTATAGGATACAGAGGCC-3′ for ΔFlank, 5′-TTGGAGTTAATTAGCCATTACGGTCGTCTGTGTTTACTG-3′ and 5′-GTAATGGCTAATTAACTCCAAGTGACGGCCATGCAA-3′ for ΔGata4/5/6, 5′- TTGCATGGCCGTAGCCATTACGGTCGTCTGTGTTTACTG-3′ and 5′-GTAATGGCTACGGCCATGCAAGTCTATAGGATACAGAGGCC-3′ for Δ(Flank+Gata4/5/6), 5′-TTACGGTCGTCTCTGATGTTTAGAAACAGGTTGACTGGGGG-3′ and 5′-AAACATCAGAGACGACCGTAATGGCTCCTTATCTCGCAATTAACTC-3′ for ΔFoxa1/2/3) and cloned into the AscI–NotI restriction site of the 2xGFP vector.

## Motif enrichment analysis

### Motif selection

Motifs were selected from Cis-BP (Lambert et al, 2019), ANISEED (Dardaillon et al, 2020), and HOMER (Heinz et al, 2010). The motif database is available at https://github.com/kewiechecki/

CrobustaTFs. For each unique transcription factor, the PWM with the lowest Shannon entropy was selected. Motifs were searched in the *C. robusta* accessome (Racioppi et al, 2019) using motifmatchr with default parameters.

### Motif accessibility

ATAC-seq reads were mapped to each motif site using bedtools. Differential accessibility was calculated using DESeq2. Motifs were considered differentially accessible if the likelihood ratio test returned a false discovery rate <0.1.

DA motif sets were defined as follows:

Motif Set log2 fold change cutoff

Open in FoxF: FoxF_KO vs. control > 1

Closed in FoxF: FoxF KO vs. control < −1

TVC accessible mesp_dnFGFR vs. control < −1 or mesp_Mek-Mut vs. control > 1

ATM accessible mesp_dnFGFR vs. control > 1 or mesp_Mek-Mut vs. control < −1

ASM accessible handr_dnFGFR vs. control < −1 or handr_-MekMut vs. control > 1

Heart accessible handr_dnFGFR vs. control > 1 or handr_Mek-Mut vs. control < −1.

### Motif annotation

Motif sites were assigned to a gene if they overlapped the gene body, promoter, or distal region of a gene. The 5' UTR, coding regions, introns, and 3' UTR were considered the gene body. −1107 to +107 bp from the transcription start site (TSS) was considered the promoter. −10 kbp from the TSS or +10 kbp from the transcription termination site (TTS) were considered distal elements.

### Enrichment

All enrichments were calculated using a hypergeometric test for the occurrence of a motif in the target motif set vs. occurrence in the background. We tested for motif enrichment in both sets of differentially accessible peaks in each condition, all peaks annotated to a gene cluster, and TVC accessible peaks annotated to a gene cluster. We also tested for enrichment of peaks annotated to specific feature types. We ran separate tests using all accessible motifs as a background and using TVC-accessible peaks as a background.

## Transcription factor-coding gene analysis

Transcription factor-coding genes IDs and corresponding names were downloaded from the Ghost database (Satou et al, 2005) http://ghost.zool.kyoto-u.ac.jp/HT_TF_KY21.html and integrated with our dataset using our KY21-KY-KH dictionary (Dataset EV1). The pivot table function in Excel was used to parse gene lists by cluster and subclusters. The heatmap and GeneExpr functions of our online http://christiaenlab-sars.com/scRNA-seq-datasets/ were used to generate heatmaps and FeaturePlots shown in Fig. 7 and Appendix Figs. S14–S20.

## Data availability

TVC trajectories data and CRISPR data collected at 12 hpf are available under the GEO accession number GSE274445. Whole-embryo data is available under the GEO accession number GSE275373.

The source data of this paper are collected in the following database record: biostudies:S-SCDT-10_1038-S44318-025-00613-y.

## Peer review information

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

## Acknowledgements

We are grateful to Drs. Alberto Stolfi and Basile Gravez for preliminary work on *Depdc1^CRISPR*, and cell cycle progression and coupling with gene expression, respectively. We are grateful to NYU Gencore facility for support with the FACS and scRNA-seq experiments. This work was supported by NIH awards R01 HL108643 and R01 HD096770 to LC, and by core funding to the S16 (Christiaen) group at the Michael Sars Centre, from the University of Bergen and from the Research Council of Norway. PZ and WW were supported by a grant from the National Natural Science Foundation of China (Grant No. 32270870, to WW).

## Author contributions

**Yelena Y Bernadskaya**: Data curation; Formal analysis; Supervision; Investigation; Visualization; Writing—original draft; Project administration. **Ariel Kuan**: Investigation. **Andreas Tjärnberg**: Formal analysis. **Jonas Brandenburg**: Data curation; Formal analysis. **Ping Zhang**: Investigation. **Keira Wiechecki**: Data curation; Formal analysis. **Nicole Kaplan**: Investigation. **Margaux Failla**: Investigation. **Maria Bikou**: Investigation. **Olivier Madilian**: Investigation. **Noah Bruderer**: Resources; Data curation; Software; Visualization. **Wei Wang**: Conceptualization; Data curation; Formal analysis; Supervision; Validation; Investigation; Visualization; Methodology; Project administration. **Lionel Christiaen**: Conceptualization; Resources; Data curation; Formal analysis; Supervision; Funding acquisition; Validation; Investigation; Visualization; Writing—original draft; Project administration; Writing—review and editing.

Source data underlying figure panels in this paper may have individual authorship assigned. Where available, figure panel/source data authorship is listed in the following database record: biostudies:S-SCDT-10_1038-S44318-025-00613-y.

## Funding

## Disclosure and competing interests statement

The authors declare no competing interests.

