## [Peer Review File · The EMBO Journal]

Transcriptome maturation confers multilineage competence to cardiopharyngeal progenitors

Yelena Bernadskaya, Ariel Kuan, Andreas Tjärnberg, Jonas Brandenburg, Ping Zhang, Keira Wiechecki, Nicole Kaplan, Margaux Failla, Maria Bikou, Olivier Madilian, Noah Bruderer, Wei Wang, and Lionel Christiaen

Corresponding authors: Lionel Christiaen (lionel.christiaen@uib.no) , Wei Wang (ww8898@ouc.edu.cn)

Review Timeline:

Submission Date:	27th Sep 24
Editorial Decision:	18th Nov 24
Revision Received:	29th May 25
Editorial Decision:	16th Jul 25
Revision Received:	27th Sep 25
Accepted:	10th Oct 25

Editor: Ieva Gailite

Transaction Report:

Dear Lionel,

Thank you for submitting your manuscript for consideration by the EMBO Journal. I sincerely apologise for the unusually prolonged assessment process due to delays in reviewer report submission, We have now received comments from three reviewers, which are included below for your information.

As you will see from the reports, all reviewers appreciate the study and find that it is a significant contribution to the research field. However, reviewers #1 and #2 with expertise in developmental biology and single cell transcriptomics analyses indicate that, in addition to toning down of some statements, further insights into the performed single cell RNAseq analysis would be needed; reviewer #2 in particular suggests applying RNA velocity and SCENIC+ analyses on the dataset.

Based on the overall interest and positive assessments by these experts, I invite you to revise the manuscript along the lines indicated in the referee comments. I would be happy to discuss the revision in more detail via email or phone/videoconferencing should that be helpful. I should also add that it is The EMBO Journal policy to allow only a single major round of revision and that it is therefore important to resolve the main concerns at this stage.

We generally allow three months as standard revision time, which can be extended to six months in the case of major revisions. Should you foresee a problem in meeting this deadline, please let us know in advance to discuss an extension. As a matter of policy, competing manuscripts published during this period will not negatively impact on our assessment of the conceptual advance presented by your study. However, please contact me as soon as possible upon publication of any related work to discuss the appropriate course of action.

When preparing your letter of response to the referees' comments, please bear in mind that this will form part of the Review Process File and will therefore be available online to the community. For more details on our Transparent Editorial Process, please visit our website: <https://www.embopress.org/page/journal/14602075/authorguide#transparentprocess>. Please also see the attached instructions for further guidelines on preparation of the revised manuscript.

Please feel free to contact me if you have any further questions regarding the revision. Thank you for the opportunity to consider your work for publication. I look forward to your revision.

With best wishes,

Ieva

- a point-by-point response to the referees' comments, with a detailed description of the changes made (as a word file).

- a word file of the manuscript text.
 - individual production quality figure files (one file per figure)
 - a complete author checklist, which you can download from our author guidelines (<https://www.embopress.org/page/journal/14602075/authorguide>).
 - Expanded View files (replacing Supplementary Information)
- Please see out instructions to authors
<https://www.embopress.org/page/journal/14602075/authorguide#expandedview>
- a Reagents and Tools Table as part of the Methods section, which can be downloaded from our author guidelines (<https://www.embopress.org/page/journal/14602075/authorguide#structuredmethods>)

We realize that it is difficult to revise to a specific deadline. In the interest of protecting the conceptual advance provided by the work, we recommend a revision within 3 months (16th Feb 2025). Please discuss the revision progress ahead of this time with the editor if you require more time to complete the revisions.

Referee #1 (ascidian biology, single cell analyses):

This manuscript from Lionel Christaen's lab uses single cell transcriptomics and a large-scale CRISPR screen to identify new regulators of Trunk Ventral Cells (TVCs) in the *Ciona* cardiopharyngeal lineage. The focus is on understanding how the TVCs gain competence over time to undergo their fate split into STVCs and First Heart Precursors (FHPs). They show that passage through mitosis is necessary but not sufficient for this competence, and identify candidate genes that become expressed in the TVCs late in the previous interphase. They test several of these in a tissue-specific CRISPR screen and identify several potential regulators, including a DEPDC1b ortholog where gene disruption causes major effects on asymmetric division in the TVC lineage. The single cell RNAseq analyses and CRISPR screen are very impressive and provide a newly detailed and comprehensive look at what has become an important model for systems-level analyses of cell fate decisions. It is, in general, a rigorous manuscript on a topic of broad interest. The conceptual advances are well matched to the journal. That said, there are several non-trivial issues that need to be addressed.

1. The manuscript makes grandiose claims to have discovered a new process of 'transcriptome maturation'. It is widely understood that transcriptional profiles change over time and that these changes are needed for cells to undergo later cell fate decisions. There is no novelty to that idea and it should not be claimed as a breakthrough. The novelty here is that they gained an unusually comprehensive view of what these changes actually are for a specific cell fate decision and how they are linked to the cell cycle. The grandiose language in the abstract, conclusion and elsewhere about 'transcriptome maturation' as a newly described concept needs to be greatly toned down.
2. The details of the time series scRNAseq experiments are not clear. How many adults were used to provide sperm and eggs? Was there a separate fertilization and barcode electroporation for each timepoint? Were gametes from different animals used for each timepoint? The methods section at lines 656-666 is not at all clear. The sentence beginning 'The best practice to avoid technical variation...' needs to either cite sources or provide some argument as to why these are the best practices. It still isn't clear from the methods text, however, how these experiments are either following or deviating from those best practices.
3. In figure 2E, it looks like the sample barcodes from the oldest few timepoints are only represented on the TVC branch and not the ATM branch. I know the focus of this ms is on the TVCs and not the ATMs, but potential imbalances here raise concerns about the entire approach. The barcode-less cells labeled in black also seen potentially non-random in how they are distributed. These issues should be addressed, at least in part by quantifying the distribution of timepoint barcodes across the two branches (like 2G but split out by ATM vsTVC branch), and by showing plots equivalent to Figure 2L for some markers of the ATM lineage.
4. Line 194: "Inspection of known markers *Mesp*/*MSGN1*, *Foxf*, *Gata4/5/6* and *Myod*/*Myf5* confirmed that the clusters containing most barcoded transcriptomes correspond to B7.5 lineage cells (Figures 2C, 2-S1G-I). The point being made here is supported by the supplemental figure and not the main text figure panel. That said, the inclusion/exclusion of the non-B7.5 lineage cells seems to make a major change in the shape of the umap- in the supplemental figure the barcoded cells are all B7.5 lineage but they no longer show the nice fork into ATM and TVC branches seen in Fig 2C. Should we be worried about this? umap

projections can be unpredictable and mysterious but this is a bit unsettling...

5. The whole-embryo single cell data in Figure 3 is very interesting but it is tangential to the thrust of this manuscript and not analyzed in any detail. I recommend this be removed and saved for another publication.

6. One of the major strengths of the manuscript is that it integrates relatively tight experimental timepoints with a detailed pseudotime analysis. These generally match up quite well, but the language at lines 219-222 seems to imply that pseudotime reconstruction might be making diverging patterns of gene expression somewhat smoother than they really are. There are certainly hints of this comparing 2I and 2J. This should be addressed more directly. Can you distinguish between pseudotime reconstruction smoothing out more step-like temporal patterns vs binned timepoints artifactually imposing them? Does this matter with respect to the hypothesis that maturation involves a progression through a series of distinct gene expression states?

7. The manuscript is extremely vague as to how genes were selected for the CRISPR screen. How many genes were actually in clusters 4/5? What were the selection criteria?

8. The extensive quantitative phenotyping of the CRISPR screen is a real strength of the manuscript, but too much is lost when it is just boiled down to a dot plot of p-values in Fig 4B. The plots in Fig 4-S1A-C should really be included in the main figure. In particular, these make it clear that DEPDC1B is a real outlier in terms of effect size. That said, I'm having a hard time reconciling the information presented across the main fig and the supplemental fig. The color scale in 4B is confusing- does red indicate $\text{fdr} < 0.05$ for all three assays? why is the color map scaled differently for the different assays? At first glance, 4B makes it look like more of the perturbations have statistically significant phenotypes than 4-S1... This should be clarified...

9. The discussion of feedforward networks is very confusingly presented. The heart of it is simply that FoxF is needed to induce GATA expression and both FoxF and GATA are needed to directly activate DepDC1b. The data are convincing but this needs to be stated clearly and directly.

10. The text needs quite a bit of editing- there are lots of typos, some of the supplemental figure legends are messed up, and the writing overall needs greater clarity.

Referee #2 (single cell transcriptomics, developmental biology):

Bernadskaya, Kuan, and Tjarnberg et al. present a comprehensive study on Ciona development at single-cell resolution, augmented by CRISPR perturbation experiments. The authors also tackle fundamental questions regarding the relationship between the cell cycle, cell fate decisions, and progenitor maturation prior to fate transitions.

While the article presents significant contributions and sets ambitious goals, there remains considerable room for improvement in the analyses.

Major Issues:

1. Line 230: It is challenging to determine whether a state is stable or not based solely on gene expression patterns. The mixed expression programs of adjacent states could result from noise. The authors could strengthen this point by applying RNA velocity to validate these observations.

2. The authors reference the concept of canalization multiple times but only vaguely define it at the end, in the discussion section. They should define it at its first occurrence and highlight the canalized trajectories on the UMAP with arrows, rather than relying solely on Venn diagrams of genes. Moreover, quantifying canalization would significantly enhance the analysis.

3. The cell annotations for Figure 3 are of lower quality than those in previous Ciona atlases, such as Cao et al. (2019, Nature) and Winkley et al. (2021, BMC Biology). The authors should provide more precise and fine-grained labels on a comprehensive UMAP, in addition to the broad lineages shown in Figure 3D. Figures 1A and 1B are not particularly useful and should be moved to the supplementary materials. The authors should keep using one single set of cell annotation labels through out the article.

4. The authors should explain the need for denoising in Figure 2 and provide an unprocessed version of the data in the supplementary materials for comparison.

5. Line 406: The motif analysis is interesting, but it fails to correlate the motif enrichment in the regulatory elements with the expression levels of transcription factors. The authors should consider using the SCENIC+ software to derive gene regulatory networks, which would help focus on expressed transcription factors and their motifs.

6. In the Foxf, Gata, and Foxl1 loss-of-function studies with single-cell readouts, were there any observed shifts in cell states? It is difficult to discern this from Figure 5-S1G. Conducting enrichment analyses for each cell state would provide more insight for each perturbation.

7. One of the key questions the authors did not address is the underlying gene regulatory network topology. While SCENIC+ analysis (as mentioned in Major point 5) could partially answer this, the authors should also include a diagram summarizing their

models. This would help readers better navigate the results and understand the network's architecture.

8. In the discussion section, the authors summarize that most mature genes are necessary for proper oriented and asymmetric division of multipotent progenitors. This is a surprising result. The authors should explain why they believe there is a lack of redundancy or suggest alternative hypotheses to explain this observation.

Minor Points:

1. The numbering of author affiliations seems incomplete, with numbers 4 and 5 missing.
2. In line 59, the authors should consider citing Zhou et al. (2019), "Single-Cell Analysis Reveals Regulatory Gene Expression Dynamics Leading to Lineage Commitment in Early T Cell Development."
3. "FACS" is misspelled as "FAC" in three instances and should be corrected.
4. The term "pseudogenes" in line 188 is used incorrectly. It conflicts with the established concept of pseudogenes. Consider using "spike-ins" or "artificial genes" instead.
5. The use of dashes around "for example" in line 197 is confusing. Please revise for clarity.
6. The term "label transfer" is incorrectly applied here, as it usually refers to a computational approach for reference-based annotation. The authors might consider using "propagation" instead.
7. Figure 4B needs better grouping of the genes by phenotypes to provide clearer insights into regulatory modules.
8. Line 355: It seems that Figure 5A is mistakenly labeled-it should be Figure 6A.
9. Figure 5A: The clusters should be ordered temporally to reflect developmental progression.
10. Figure 5B: Gene expressions should be plotted in pseudotime rather than aggregated developmental time to give a clearer temporal perspective.
11. Line 1823: The link to the pre-print has been duplicated. The correct link should direct readers to the published version rather than the bioRxiv pre-print.

Referee #3 (ascidian development and evolution):

This is a highly significant study that provides novel insights into the intricate relationship between cell cycle progression and cell fate determination. By leveraging the unique features of the *Ciona* embryo model and employing advanced single-cell RNA sequencing techniques that includes transgene based cell barcoding, the authors have made substantial contributions to the field of developmental and stem cell/pluripotent biology.

Key strengths of the study:

- **Innovative Experimental Design:** The authors' innovative approach of combining single-cell RNA sequencing with transgene-based cell barcoding allows for a high-resolution analysis of cell fate decisions and gene expression dynamics.
- **Mechanistic Insights:** The study provides mechanistic insights into how cell cycle progression, gene expression, and cell division orientation influence cell fate specification.
- **Broad Implications:** The findings have broad implications for understanding fundamental developmental processes and may have potential applications in regenerative medicine.

Specific Highlights:

- The authors elegantly demonstrate how the cell cycle drives transcriptome maturation, leading to the acquisition of cellular competence.
- The study elucidates the crucial role of cell division orientation in determining cell fate.
- The authors identify both classical genetic regulatory networks and cell cycle-driven mechanisms that contribute to cell fate decisions.

Overall, this study is a significant advancement in the field of developmental biology. It provides a comprehensive understanding of the molecular and cellular processes that govern cell fate determination in cardiopharyngeal progenitors. The findings have the potential to inspire future research and contribute to the development of novel therapeutic strategies.

The MS is well-written and can be published as is. A minor suggestion would be to include, in the Discussion section, findings on hematopoietic stem cells, such as those presented in Passegue et al. 2005, which explored the global analysis of proliferation and cell cycle gene expression in the regulation of hematopoietic stem and progenitor cell fates.

Referee #1 (ascidian biology, single cell analyses):

This manuscript from Lionel Christiaen's lab uses single cell transcriptomics and a large-scale CRISPR screen to identify new regulators of Trunk Ventral Cells (TVCs) in the *Ciona* cardiopharyngeal lineage. The focus is on understanding how the TVCs gain competence over time to undergo their fate split into STVCs and First Heart Precursors (FHPs). They show that passage through mitosis is necessary but not sufficient for this competence, and identify candidate genes that become expressed in the TVCs late in the previous interphase. They test several of these in a tissue-specific CRISPR screen and identify several potential regulators, including a DEPDC1b ortholog where gene disruption causes major effects on asymmetric division in the TVC lineage. The single cell RNAseq analyses and CRISPR screen are very impressive and provide a newly detailed and comprehensive look at what has become an important model for systems-level analyses of cell fate decisions. It is, in general, a rigorous manuscript on a topic of broad interest. The conceptual advances are well matched to the journal. That said, there are several non-trivial issues that need to be addressed.

REPLY. We thank the reviewer for a generally positive assessment.

Technically we argue that competence is acquired toward the end of G2, and that it is the whole interphase that is required, while mitosis is required for *Tbx1/10* activation once the progenitor cells have acquired the necessary competence.

1.1. "grandiose"

The manuscript makes grandiose claims to have discovered a new process of 'transcriptome maturation'. It is widely understood that transcriptional profiles change over time and that these changes are needed for cells to undergo later cell fate decisions. There is no novelty to that idea and it should not be claimed as a breakthrough. The novelty here is that they gained an unusually comprehensive view of what these changes actually are for a specific cell fate decision and how they are linked to the cell cycle. The grandiose language in the abstract, conclusion and elsewhere about 'transcriptome maturation' as a newly described concept needs to be greatly toned down.

REPLY. First, we would like to respectfully point out that we start the abstract by stating that "*Defined changes in biomolecular composition underlie the progressive loss of potency and acquisition of lineage-specific characteristics.*", which explicitly acknowledges the reviewer's first point that: "*It is widely understood that transcriptional profiles change over time and that these changes are needed for cells to undergo later cell fate decisions.*".

We appreciate the reviewer's acknowledgement that the novelty resides in the "*unusually comprehensive view of what these changes actually are for a specific cell fate decision and how they are linked to the cell cycle*", which is precisely what the term "maturation" refers to.

Indeed, in LL. 559-563 of the discussion we stated: "*the notion of maturation has been considered extensively for differentiated cells, such as cardiomyocytes, which are contractile but not fully functional until they reach a mature state. Here, we adopted a general definition of the concept of maturation, considering a biological entity that persists through time and progressively acquires its full functionality through underlying changes.*"

The notion that this happens within one interphase (i.e. the same cell persists through time) is essential to the concept of "maturation", and thus the cell cycle context in which this happens is central to our findings and statements.

We would like to argue that our statements do not qualify as "grandiose". We do not use terms such as *novel*, *new* or *unprecedented*, or other exaggerated emphasis, when referring to what we call maturation. We kept our words very *matter-of-fact* and indicated that this is the term we use to refer to the changes in transcriptome, within one interphase, that lead to the acquisition of multilineage competence, and defined what we mean by "maturation" again in the discussion as indicated above

Finally, we are mindful of the possibility that we may have missed some key references and would be grateful and willing to incorporate them if the reviewer can point us to papers showing that transcriptome-wide changing within the same multipotent progenitor cell, prior to division, confers the equivalent of multilineage competence.

MODIFICATIONS. We rephrased the part of the abstract that reads "*We termed*" into "*We refer to as*", and amended the text as indicated in the version with tracked changes to avoid exaggerations.

1.2. scRNA-seq details

The details of the time series scRNAseq experiments are not clear. How many adults were used to provide sperm and eggs? Was there a separate fertilization and barcode electroporation for each timepoint? Were gametes from different animals used for each timepoint? The methods section at lines 656-666 is **not at all clear**. The sentence beginning 'The best practice to avoid technical variation...' needs to either cite sources or provide some argument as to why these are the best practices. It still isn't clear from the methods text, however, how these experiments are either following or deviating from those best practices.

REPLY. We thank the reviewer for this worthwhile request for clarification.

Regarding "*The details of the time series scRNAseq experiments are not clear*", we wish to point out that the details extend in the methods section on p.18-19, LL. 649-721 of the original manuscript.

Regarding "best practices", we wrote "*The best practice to avoid technical variation between samples is to prepare the fertilized eggs for all samples in a large pool and electroporate the paired SBCs individually at one-hour intervals according to the experimental design.*"

This is not how the time series samples were prepared in Cao et al., Nature, 2019, which has a lot of batch effects. We did not emphasize this point in our manuscript, and wish to avoid counterproductive and unnecessary controversy.

MODIFICATION: We amended the methods section to clarify this point.

"To avoid technical variation between samples we prepared fertilized eggs for all samples in a large pool of dechorionated eggs, obtained from multiple animals, and electroporate the paired SBCs individually at one-hour intervals according to the experimental design."

1.3. TVCvsATM

In figure 2E, it looks like the sample barcodes from the oldest few timepoints are only represented on the TVC branch and not the ATM branch. I know the focus of this ms is on the TVCs and not the ATMs, but potential imbalances here raise concerns about the entire approach. The barcode-less cells labeled in black also seem potentially non-random in how they are distributed. These issues should be addressed, at least in part by quantifying the distribution of timepoint barcodes across the two branches (like 2G but split out by ATM vs TVC branch), and by showing plots equivalent to Figure 2L for some markers of the ATM lineage.

REPLY. This is an astute observation by the reviewer, which we addressed in previous papers (e.g. Christiaen et al., Science, 2008) but not explicitly here. We know that at later stages, such as the 12-14 hpf stages represented here, anterior tail muscle cells do not sort well by FACS, presumably because of their larger size and optical properties compared to TVCs, which positions them differentially on the forward and lateral scatter plot used to define the sorting gate. We added comments to clarify this point.

MODIFICATIONS.

Plot number of cells per cluster, explain and cite papers.

Explain non-labelled cell (epidermis)

1.4. Markers, UMAP

Line 194: "Inspection of known markers Mesp/MSGN1, Foxf, Gata4/5/6 and Myod/Myf5 confirmed that the clusters containing most barcoded transcriptomes correspond to B7.5 lineage

cells (Figures 2C, 2-S1G-I). The point being made here is supported by the supplemental figure and not the main text figure panel.

REPLY. The reviewer is correct that main figure shows only the barcoded B7.5 lineage cells, most of them having a time tag as shown in Fig 2E, while panel 2C shows both Myod/Myf5 and Gata4/5/6 (ATM and TVC markers, respectively). We appreciate the reviewer's acknowledgement that "*the point being made here is supported by the supplemental figure*".

MODIFICATION. None

That said, the inclusion/exclusion of the non-B7.5 lineage cells seems to make a major change in the shape of the umap- in the supplemental figure the barcoded cells are all B7.5 lineage but they no longer show the nice fork into ATM and TVC branches seen in Fig 2C. Should we be worried about this? umap projections can be unpredictable and mysterious but this is a bit unsettling...

REPLY. We wish to point out that the supplemental figure DOES show the equivalent fork on the UMAP without "major change", which we edited to clarify this point. Moreover, UMAPs are more reproducible from one run to the next than tSNE plots. There are many criticisms (e.g. Lior Pachter's and Raphael Irizarry's takes on UMAPs), but they served a purpose to illustrate the transcriptional divergence between TVCs and ATMs, which is well described elsewhere as well as thoroughly captured in our dataset. So, in our view there is "*nothing to worry about*".

MODIFICATIONS.

Add arrows pointing at the TVCs vs. ATMs branches in the supplemental figure, and explain in the legends

1.5. Whole embryo

The whole-embryo single cell data in Figure 3 is very interesting but it is tangential to the thrust of this manuscript and **not analyzed in any detail**. I recommend this be removed and saved for another publication.

REPLY. We thank the reviewer for this positive outlook and agree that it departs from the main thread on the cardiopharyngeal progenitors and that another publication would do it more justice. In and of itself, the novelty would be impacted by its similarity with Cao et al. Nature, 2019, which also covers Ciona embryogenesis with similar temporal resolution, 4 to 5 times more cells and more extensive annotation.

We nonetheless included this section to extend both the use of our barcoding strategy and the concept of progenitor maturation, as well as to provide this dataset as a citable resource that benefits from reduced technical batch effects and DEWAKSS-based denoising.

Moreover, we wish to argue that the main and supplemental figures presented do not fit the characterization as "*not analyzed in any detail*", inasmuch as (1) we annotated groups of clusters by main tissue types, (2) with "sub-setted" and reclustered endomesodermal clusters to (3) infer trajectories and (4) annotate lineages; we (5) used this analyses to focus on branching trajectories and evaluate the concept of multipotent progenitor maturation in other contexts than the cardiopharyngeal lineage and (6) compared candidate "mature genes" across lineages.

MODIFICATIONS. There is no specific modification that we can implement to address this comment, besides removing the section, which is an editorial decision that does not impact the validity of the results and analyses presented.

We updated the server for the ShinyApp, which makes the dataset easy to navigate and mine for the community.

1.6. Pseudotime

One of the major strengths of the manuscript is that it integrates relatively tight experimental timepoints with a detailed pseudotime analysis. These generally match up quite well, but the language at lines 219-222 seems to imply that pseudotime reconstruction might be making diverging patterns of gene expression somewhat smoother than they really are. There are certainly

hints of this comparing 2I and 2J. This should be addressed more directly. Can you distinguish between pseudotime reconstruction smoothing out more step-like temporal patterns vs binned timepoints artifactually imposing them? Does this matter with respect to the hypothesis that maturation involves a progression through a series of distinct gene expression states?

REPLY. The reviewer refers to the following passage: "*We reasoned that seemingly continuous gene expression changes in fact reflect transitions between successive regulatory states. We used a clustering approach to segment individual trajectories into significantly different states (Figure 2H,I, 2-S2B)*", but we do not quite see how this "*seems to imply that pseudotime reconstruction might be making diverging patterns of gene expression somewhat smoother than they really are*".

2I and 2J indeed show gene expression in two different ways: one is aggregated by cluster and z-score normalized over pseudotime (2I), while the other is a gene by gene, and collection time point by collection time point view, both of which show an upregulation of cluster 4 and 5 candidate genes starting at approx. 10 hpf.

This is further illustrated and corroborated by fluorescent *in situ* hybridization data in Fig. 2K-M for *Ptch1* and *Bmper*.

To the reviewer's point, some of these genes are upregulated already in S-phase (e.g. *Depdc1b*), while other apparently in G2 (e.g. *Ptch1*), so there are differences between individual genes in terms of gene expression dynamics along the pseudotemporal trajectory, and these changes appear more or less gradual (smooth).

Of note, this does not depend on pseudotime ordering, which is done at the level of whole cell transcriptomes. In other words, the transcriptome, which is a cell-level property that integrates all individual gene expression values, is what drives the pseudotime ordering.

The point of our comment is to highlight the change of scale between gene-level and cell-level properties, and hint at the emerging property of a state transition, whether individual genes perfectly recapitulate it or not.

MODIFICATIONS.

We change the text to read as "*We reasoned that despite potential disparate gene dynamics, individual gene expression changes may collectively reflect transitions between successive regulatory states.*"

More substantially, we added an RNA velocity-based analysis, which appears to largely corroborated our proposed regulatory states and transitions.

1.7. Candidate genes

The manuscript is **extremely vague** as to how genes were selected for the CRISPR screen. How many genes were actually in clusters 4/5? What were the selection criteria?

REPLY.

We would like to point out that our text reads "*We selected 37 candidate genes, including those coding for such signaling molecules as the trans-membrane receptors *Adra2b* and *Fzd8*, or the Rho GTPase Activating Protein (RhoGAP) *Depdc1b*.*" and that the complete list of genes in each cluster is provided as indicated in the Supplemental Table S2 (L. 242).

MODIFICATIONS.

We rephrased the above sentence into "While "mature" clusters 4 and 5 comprised 945 and 859 genes (Table S2), respectively; we cross-referenced with cardiopharyngeal markers and molecular classes, and selected 37 candidate genes, focusing on those coding for signaling molecules such as the trans-membrane receptors *Adra2b* and *Fzd8*, or the Rho GTPase Activating Protein (RhoGAP) *Depdc1b*."

1.8. CRISPR phenotypes

The extensive quantitative phenotyping of the CRISPR screen is a real strength of the manuscript, but too much is lost when it is just boiled down to a dot plot of p-values in Fig 4B. The plots in Fig

4-S1A-C should really be included in the main figure. In particular, these make it clear that DEPDC1B is a real outlier in terms of effect size. That said, I'm having a hard time reconciling the information presented across the main fig and the supplemental fig. The color scale in 4B is confusing- does red indicate $fdr < 0.05$ for all three assays? why is the color map scaled differently for the different assays? At first glance, 4B makes it look like more of the perturbations have statistically significant phenotypes than 4-S1... This should be clarified...

REPLY. We are grateful to the reviewer for this generous assessment, as we think that our CRISPR "screen" remains limited and qualitative. By presented the summary statistics in the main figure, we aimed to simplify and streamline a Figure that also has more data in the second two thirds (Fig. 4C-I), which provide a more quantitative analysis of the $Depdc1^{CRISPR}$ phenotype and its impact on the division pattern and *Tbx1/10* expression.

The reviewer is correct that *Depdc1b* is an outlier in our dataset

MODIFICATIONS.

- Consolidated the main and supplemental data into one main figure (now figure 8)

1.9. Feed-forward

The discussion of feedforward networks is very confusingly presented. The heart of it is simply that FoxF is needed to induce GATA expression and both FoxF and GATA are needed to directly activate DepDC1b. The data are convincing but this needs to be stated clearly and directly.

REPLY. The reviewer is correct that the feedforward circuit is most clearly demonstrated for Foxf-Gata4/5/6 and Depdc1b interactions. However, we also report on $Foxf^{CRISPR}$ and $Gata4/5/6^{CRISPR}$ perturbations profiled jointly by scRNA-seq and deconvolved *in silico* using our sample barcoding method. Combined with our cardiopharyngeal trajectory analysis and published ATAC-seq and motif enrichment analysis, this suggested further regulatory cascades and feed-forward logics, which we did not detail indeed. This comment echoes reviewer 2's request to use SCENIC+ and expand our analysis of the GRN.

MODIFICATIONS.

We expanded substantially our description of the transcription factor coding genes and their dynamic expression in the dataset, as well as correspondence with enriched motifs in this and our previous study (Racioppi et al., eLife, 2019).

1.10. Editing

The text needs quite a bit of editing- there are lots of typos, some of the supplemental figure legends are messed up, and the writing overall needs greater clarity.

REPLY. Thank you.

MODIFICATIONS. Too many to list here. See version with tracked changes

Referee #2 (single cell transcriptomics, developmental biology):

Bernadskaya, Kuan, and Tjarnberg et al. present a comprehensive study on Ciona development at single-cell resolution, augmented by CRISPR perturbation experiments. The authors also tackle fundamental questions regarding the relationship between the cell cycle, cell fate decisions, and progenitor maturation prior to fate transitions.

While the article presents significant contributions and sets ambitious goals, there remains considerable room for improvement in the analyses.

Major Issues:

2.1. states

Line 230: It is challenging to determine whether a state is stable or not based solely on gene expression patterns. The mixed expression programs of adjacent states could result from noise. The authors could strengthen this point by applying RNA velocity to validate these observations.

REPLY. This is a good point by the reviewer, considering that we indeed made that point based on the observation that gene expression changes during this transition, which follows a pseudotime ordering of the single cell transcriptomes with a high correlation with real time. We defined the states based on clustering along this time-informed trajectory, the states do comprehend the gene expression dynamics and not static time points.

Unfortunately, only about 3% of the reads in our dataset could be used for RNA velocity, and while we attempted it initially, we did not pursue this route. Instead, our self-supervised denoising/imputation methods allowed us to build smooth gene expression trajectories (e.g. Fig. 2L), which we used to calculate "pseudovelocity" as the derivative of the imputed expression level over pseudotime.

MODIFICATIONS.

We added an RNA velocity based analysis

2.2. Canalization

The authors reference the concept of canalization multiple times but only vaguely define it at the end, in the discussion section. They should define it at its first occurrence and highlight the canalized trajectories on the UMAP with arrows, rather than relying solely on Venn diagrams of genes. Moreover, quantifying canalization would significantly enhance the analysis.

REPLY. This is a good point by the reviewer, since we used a slightly different definition than the more commonly used one. Indeed, we refer to the observations that transcriptome changes in the endomesodermal lineages follows temporal progression, and presumably early lineage separation, by contrast with neurectodermal lineages, especially the epidermis, where secondary patterning events seem to override the transcriptional signatures of lineages.

In the literature, the term "canalization" has been used to refer to the robustness of biological systems to external perturbations, which is not how we use it here. We use a more "Waddingtonian" version where cells enter fate restricted path and changes over time.

We're not quite sure how to quantify canalization, and would appreciate pointers to publication and established methods where this has been achieved.

MODIFICATIONS.

We avoided using the term "canalization" in the way we intended and instead discuss clonally-guided developmental progression.

2.3. Whole embryo

The cell annotations for Figure 3 are of lower quality than those in previous Ciona atlases, such as Cao et al. (2019, Nature) and Winkley et al. (2021, BMC Biology). The authors should provide more precise and fine-grained labels on a comprehensive UMAP, in addition to the broad lineages shown in Figure 3D. Figures 1A and 1B are not particularly useful and should be moved to the supplementary materials. The authors should keep using one single set of cell annotation labels through out the article.

REPLY. The reviewer is correct that we did not intend to present and use our whole embryo scRNA-seq dataset as a comprehensive atlas, in part due to the existence of other such atlases for Ciona, and in order to focus on extensions of (1) the transgene-based sample barcoding approach and (2) the concept of multipotent progenitor maturation, for which we focused on endomesodermal lineages.

We also wish to address the possibility of mining the dataset further in future publications, and would argue that a much more exhaustive annotation, while warranted, extend beyond the scope of this study.

We nonetheless provide a ShinyApp allowing users to mine the datasets in an interactive fashion. We argue that such atlas project is poised to be a live one that evolves and improves, and this is how with intend to keep augmenting the interpretability of our dataset.

MODIFICATIONS.

We updated the hosting solution for our ShinyApps, in an attempt to make them more stable and easier to use.

2.4. Denoising

The authors should explain the need for denoising in Figure 2 and provide an unprocessed version of the data in the supplementary materials for comparison.

REPLY. The key advantage of using denoised data is for cell-level analysis of trajectories and states, and cluster-level analysis of expression profiles along trajectories. Basically, the method helps focus on the gene-level signal that drive these higher level patterns.

MODIFICATIONS.

We added the raw/undenoised version for some of the analyses e.g. in Figure 2-S3

2.5. Motifs, SCENIC+

Line 406: The motif analysis is interesting, but it fails to correlate the motif enrichment in the regulatory elements with the expression levels of transcription factors. The authors should consider using the SCENIC+ software to derive gene regulatory networks, which would help focus on expressed transcription factors and their motifs.

REPLY. The reviewer is correct that gene expression for transcription factors does not always reflect their activity (e.g. Tjärnberg et al., Genome Biol, 2024), and that motif enrichment in accessible regions could reflect other inputs than those considered during the cardiopharyngeal progenitor migration. For example, certain enhancers are open but not active until later during cardiopharyngeal fate decisions (e.g. Racioppi et al., eLife, 2019).

With regards to the use of SCENIC+, we do appreciate the suggestion and share the reviewer's interest in deploying computational methods to further our understanding of the gene regulatory network governing TVC maturation.

We believe, however, that this would extend well beyond the scope of this study, especially when considering the necessary follow-up validation experiments for the computation predictions that would emerge.

Finally, the Results section of the SCENIC+ paper reads :"*SCENIC+ takes as input either paired or unpaired scRNA-seq and scATAC-seq data*"; unfortunately, we do not have scATAC-seq data at the corresponding resolution for the cardiopharyngeal progenitors, but only bulk ATAC-seq data obtained from FACS-purified cells at one time point in Racioppi et al., eLife 2019.

MODIFICATIONS.

We provided a more extensive description of the dynamic gene expression of transcription factor coding genes and correspondence with enriched motifs in this and our previous Racioppi et al, 2019 paper.

We will have to reserve the SCENIC+ and related approaches for scATAC-seq and GRN modeling to future work.

2.6. CRISPR x scRNA-seq

In the Foxf, Gata, and Foxl1 loss-of-function studies with single-cell readouts, were there any observed shifts in cell states? It is difficult to discern this from Figure 5-S1G. Conducting enrichment analyses for each cell state would provide more insight for each perturbation.

REPLY. This is good question from the reviewer, and something that we observed with Foxf^{CRISPR}, as shown in Fig. 5S1C and G, where clusters 4 and 7 appear enriched Foxf^{CRISPR} cells (which we can quantify better), but not with Gata4^{CRISPR} or Foxtun4^{CRISPR} (misnamed Foxl1 in our previous version). We do not think we can formally conclude on a negative result for the latter two, but ought to clarify the wider effects of Foxf^{CRISPR}, which corroborates several previous studies (e.g. Fig. 7-S9). Specifically, the observed enrichment of cluster 5 genes (Fig. 7), which are upregulated late during TVC maturation, and converse enrichment of clusters 2 and 6 genes, which are normally down-regulated by the later time point, suggest that at least part of the Foxf^{CRISPR} phenotype can be interpreted as a failure to mature.

MODIFICATIONS.

We highlighted the importance of the "intermediate" state, which includes key transcription factors like Foxf and Gata4/5/6, for the transition to a mature state.

2.7. Underlying GRN

One of the key questions the authors did not address is the underlying gene regulatory network topology. While SCENIC+ analysis (as mentioned in Major point 5) could partially answer this, the authors should also include a diagram summarizing their models. This would help readers better navigate the results and understand the network's architecture.

REPLY. As above, we appreciate the reviewer's interest in the cardiopharyngeal GRN, which has been a long-standing interest of our, but also a challenging one to formally model using mathematical and computational methods.

MODIFICATIONS.

We provided more extensive analyses of transcription factor coding genes and an updated and extended summary model, albeit still qualitative.

2.8. Discussion - phenotypes

In the discussion section, the authors summarize that most mature genes are necessary for proper oriented and asymmetric division of multipotent progenitors. This is a surprising result. The authors should explain why they believe there is a lack of redundancy or suggest alternative hypotheses to explain this observation.

REPLY. This is another good question from the reviewer.

One element to take into consideration, especially compared to vertebrate models, is the fact that the Ciona genome did not undergo whole genome duplication and numerous genes are thus present in a single copy.

This surprising lack of redundancy might also hint at the extreme determinism observed in Ciona, including for the migration and division patterns of cardiopharyngeal progenitors, which may thus rely on very tight molecular control. But that is very speculative and we ought to keep the discussion focused.

In another study that we are finalizing (Failla, Wiechecki et al., in preparation), we performed high-content morphometric analysis of cellular phenotypes obtained with some of the same CRISPR perturbations and started building a provisional model of the modular subcellular network controlling these cardiopharyngeal behaviors. We find several of these "mature genes" in shared

modules controlling specific aspects of the oriented and asymmetric division. This is going to be one of our longstanding quests for understanding.

MODIFICATIONS.

Add a note to that effect in the discussion, and refer generally to the paper in preparation.

Minor points

1. The numbering of author affiliations seems incomplete, with numbers 4 and 5 missing.
2. In line 59, the authors should consider citing Zhou et al. (2019), "Single-Cell Analysis Reveals Regulatory Gene Expression Dynamics Leading to Lineage Commitment in Early T Cell Development."
3. "FACS" is misspelled as "FAC" in three instances and should be corrected.
4. The term "pseudogenes" in line 188 is used incorrectly. It conflicts with the established concept of pseudogenes. Consider using "spike-ins" or "artificial genes" instead.
5. The use of dashes around "for example" in line 197 is confusing. Please revise for clarity.
6. The term "label transfer" is incorrectly applied here, as it usually refers to a computational approach for reference-based annotation. The authors might consider using "propagation" instead.
7. Figure 4B needs better grouping of the genes by phenotypes to provide clearer insights into regulatory modules.
8. Line 355: It seems that Figure 5A is mistakenly labeled-it should be Figure 6A.
9. Figure 5A: The clusters should be ordered temporally to reflect developmental progression.
10. Figure 5B: Gene expressions should be plotted in pseudotime rather than aggregated developmental time to give a clearer temporal perspective.
11. Line 1823: The link to the pre-print has been duplicated. The correct link should direct readers to the published version rather than the bioRxiv pre-print.

Referee #3 (ascidian development and evolution)

This is a highly significant study that provides novel insights into the intricate relationship between cell cycle progression and cell fate determination. By leveraging the unique features of the *Ciona* embryo model and employing advanced single-cell RNA sequencing techniques that includes transgene based cell barcoding, the authors have made substantial contributions to the field of developmental and stem cell/pluripotent biology.

Key strengths of the study:

- **Innovative Experimental Design:** The authors' innovative approach of combining single-cell RNA sequencing with transgene-based cell barcoding allows for a high-resolution analysis of cell fate decisions and gene expression dynamics.
- **Mechanistic Insights:** The study provides mechanistic insights into how cell cycle progression, gene expression, and cell division orientation influence cell fate specification.
- **Broad Implications:** The findings have broad implications for understanding fundamental developmental processes and may have potential applications in regenerative medicine.

Specific Highlights:

- The authors elegantly demonstrate how the cell cycle drives transcriptome maturation, leading to the acquisition of cellular competence.
- The study elucidates the crucial role of cell division orientation in determining cell fate.
- The authors identify both classical genetic regulatory networks and cell cycle-driven mechanisms that contribute to cell fate decisions.

Overall, this study is a significant advancement in the field of developmental biology. It provides a comprehensive understanding of the molecular and cellular processes that govern cell fate determination in cardiopharyngeal progenitors. The findings have the potential to inspire future research and contribute to the development of novel therapeutic strategies.

The MS is well-written and can be published as is. A minor suggestion would be to include, in the Discussion section, findings on hematopoietic stem cells, such as those presented in Passegue et al. 2005, which explored the global analysis of proliferation and cell cycle gene expression in the regulation of hematopoietic stem and progenitor cell fates.

REPLY. We thank the reviewer for this positive and generous assessment. We agree that relevant references abound in the hematopoietic literature.

MODIFICATION.

Update the citations accordingly.

Dear Lionel,

Thank you for submitting a revised version of your manuscript. It has now been seen by two of the original reviewers, and I have copied their comments below.

As you can see, while both reviewers appreciate the added revisions, they also find that some of their initial requests were not clarified satisfactorily. Please address these remaining points by both reviewers in the final minor revision, which would mainly require textual clarifications. From the editorial side, removal of whole embryo data (reviewer #1, point 1.5) is not required.

Additionally, there are a few editorial points that need addressing before I can extend official acceptance of the manuscript:

1. Please submit up to five keywords.
2. Please check that the funding information is correct and identical both in the manuscript and our online system. Currently, the University of Bergen, the Research Council of Norway and the National Natural Science Foundation of China (Grant No. 32270870) are missing in our online system.
3. Please submit a complete author checklist, which you can download from our author guidelines (<https://www.embopress.org/pb-assets/embo-site/EMBO%20Press%20Author%20Checklist-1642513524327.xlsx>). Please insert information in the checklist that is also reflected in the manuscript. The completed author checklist will also be part of the Review Process File.
4. Please remove the figures from the manuscript text file and upload the main figures as individual production quality figure files in the .eps, .tif, or .jpg format (one file per figure).
5. The supplemental figures should be removed from the manuscript text and compiled in a PDF file labelled "Appendix". The appendix file will need a short table of contents with page numbers. Please correct the nomenclature to "Appendix Figure S1" etc.
6. All Materials and Methods need to be described in the main text using our 'Structured Methods' format. According to this format, the Methods section includes a Reagents and Tools Table (listing key reagents, experimental models, software and relevant equipment and including their sources and relevant identifiers) followed by a Methods and Protocols section describing the methods, ideally using a step-by-step protocol format. The aim is to facilitate adoption of the methodologies across labs. Please download and fill our Reagents and Tools Table template (.docx), which you can find in our author guidelines: <https://www.embopress.org/page/journal/14602075/authorguide#structuredmethods>
When submitting your revised manuscript, please do not include the Reagents and Tools Table in the Methods section of the manuscript but upload it as a separate file choosing the file type "Reagent Table".
An example of a Method paper with Structured Methods can be found here: <https://www.embopress.org/doi/10.15252/msb.20178071>.
7. CRedit has replaced the traditional author contributions section because it offers a systematic, machine-readable author contributions format that allows for more effective research assessment. Please remove the Authors Contributions from the manuscript and use the free text boxes beneath each contributing author's name in our online submission system to add specific details on the author's contribution. More information is available in our guide to authors.
8. Please add a "Disclosure and competing interests statement" section after "Acknowledgements". Further info: <https://www.embopress.org/page/journal/14602075/authorguide#conflictsofinterest>.
9. Please update references according to The EMBO Journal style - it should be alphabetical. Where there are more than 10 authors on a paper, the first 10 should be listed, followed by 'et al.' Please see further information here: <https://www.embopress.org/page/journal/14602075/authorguide#referencesformat>
10. Please rename the supplementary tables into Dataset EV1 - EV7. Please remove the legends from the manuscript text and add to each dataset file in a separate tab/worksheet.
11. Please rename "Materials and Methods" into "Methods".
12. Several figure panels are not mentioned in the manuscript text: Fig 3I, Fig 4E,F,G, Fig 5G, Fig 6A, Fig 7G, Fig 8E,F,G. Table S1 is also not mentioned in the text. There is a callout for Table S8, but this file is missing.
13. Figure panels 8N-Q are mentioned in the manuscript, but there are no panels P and Q.
14. Please check that the figure panels are mentioned in the manuscript text in a sequential order. Currently, Fig 8 is called out before Fig 7F.
15. In the Data Availability section, please add resolvable links to the datasets. More information about the format of this section can be found here: <https://www.embopress.org/page/journal/14602075/authorguide#dataavailability>.
16. Please upload source data files as one zip folder per figure.
17. Our data editors have flagged the following issues in figure legends that need correcting:
 - Please provide the exact p values in the legend of figure 5C-E; 6B.
 - Please indicate the statistical test used for data analysis in the legends of figures 5C-E.
 - Please note that the box plots need to be defined in terms of minima, maxima, centre, bounds of box and whiskers, and percentile in the legend of figure 6B.
 - Please provide information on the number and nature of replicates in the legends of figures 1N, 3D, 5C-E; 6B; 7F.
 - Please define the error bars in the legends of figures 1N; 5C-E; 6B.

- Please note that scale bar and its definition are missing for figure 5B.

18. Papers published in The EMBO Journal are accompanied online by a 'Synopsis' to enhance discoverability of the manuscript. It consists of A) a short (1-2 sentences) summary of the findings and their significance, B) 3-4 bullet points highlighting key results and C) a synopsis image that is 550x300-600 pixels large (width x height, jpeg or png format). You can either show a model or key data in the synopsis image. Please note that the image size is rather small and that text needs to be readable at the final size.

With best wishes,

Ieva

We realize that it is difficult to revise to a specific deadline. In the interest of protecting the conceptual advance provided by the work, we recommend a revision within 3 months (14th Oct 2025). Please discuss the revision progress ahead of this time with the editor if you require more time to complete the revisions.

Referee #1:

Comments in response to the rebuttal

1.1 'grandiose'. I feel like the authors and I are maybe talking past one another on this issue. My point is just that it is not the *concept* of transcriptome maturation that is novel here. The novelty comes from how they've elegantly identified the actual details of one specific example of transcriptome maturation. I'm generally happy with the changes here. I'd recommend removing the quotation marks when discussing transcriptome maturation, regulatory competence, and behavioral competence, to help make it clear that you are defining your terms without the implication that these are newly minted concepts

Another modest but helpful change here would be to refine the discussion at lines 641-645 of the revised MS. It may not be the authors' intent, but to me this sentence seems to argue that developmental biologists have only addressed the concept of 'maturation' in the context of terminal differentiation and not earlier cell fate decisions. It's probably true that the word 'maturation' has mostly been used in the context of terminal differentiation, but the Melton review cited takes a very flexible view of maturation and it is certainly the case that developmental biologists routinely study transcriptional changes over time that are important for early branches in the Waddingtonian landscape.

1.2 'scRNAseq details'

I promise you I did read the methods section as well on the original submission... I'm still confused though by the details here.

The current methods text reads 'To avoid technical variation between samples, we prepared fertilized eggs for all samples in a large pool of dechorionated eggs, obtained from multiple animals, and electroporate the paired SBCs

individually at one-hour intervals according to the experimental design.'

This doesn't make sense as written, because if you did indeed prepare a single large pool of fertilized and dechorionated eggs, then you wouldn't have been able to do the separate fertilizations needed to have all the timepoints ready for the same harvest time. I'm assuming you killed a bunch of animals at the start of the experiment and stored the pooled sperm and pooled eggs separately, then fertilized aliquots at intervals? That seems like a smart way to do it but it is not clear from how it is described in either the original or revised submissions... Did you dechorionate the whole pool of unfertilized eggs and then fertilize them in batches afterwards or did you do separate dechorionations for each fertilization? How were the gametes stored? Are there any concerns about potential biases from having unfertilized eggs sit around ex vivo for many hours? This just needs a bit more detail and clarity.

1.3. The issue of some cell types being more sortable and recoverable than others in scRNAseq experiments is problematic and often swept under the table in scRNAseq papers. The manuscript is much improved here with the clear discussion of ATMs being less sortable than TVCs. I'm still a bit suspicious that there could be something more going on given how there seems to be such a big difference in ATM sortability within a single cell cycle, but I'm happy with how it is treated within the scope of this manuscript.

1.4 I'm satisfied with the changes made.

1.5 I continue to feel that this is a very long paper and that the whole embryo experiments aren't critical to its key findings. It would be an obvious thing to remove if there is an editorial consensus that the paper should be shorter. It's interesting data though and I have no problem with it staying in as long as the journal is happy with the overall length and complexity.

1.6 I'm happy with the changes here. I still think there are some interesting opportunities to use the relatively high temporal resolution of this time course dataset to test some of the strengths/weaknesses of pseudotime inference more broadly, but that would be outside the scope of this MS.

1.7 The revised manuscript makes it clear that these are just a subset of the candidate genes and that they were selected manually based on having interesting predicted molecular functions. I'm satisfied with the changes.

1.8 This figure is much improved and far more interpretable with the key findings moved into the main text figure from the supplements. I'm generally happy with it. I remain concerned about Fig 5G, where my previous concerns have not been addressed in the rebuttal. From a data-viz perspective, I find it confusing to use a diverging color map for p-value data, at least not without having it centered on a threshold of interest like $-\log_{10}(0.05)$. It's also confusing to have different scales for the three assays. Beyond that, I'm still having a hard time reconciling the $-\log_{10}$ (FDR) p-values shown in 5G with the p-values indicated with brackets in 5C-E and the Venn diagram in F. Why do so many more genes seem to have strong statistical support for a defect in Tbx1/10 expression in 5G vs 5E? Are these different statistical tests? This section would also benefit from a more detailed treatment in the methods of how the images were analyzed, how the orientation, asymmetry and expression data were reduced to categorical phenotypes, and which statistics were used where.

1.9 No concerns as currently presented.

1.10 Much improved, though I note a few small things lower down.

I'll also note a few things that resonated with me in response to Reviewer 2's comments:

2.1 I like the velocity analysis though I'd suggest that the text should be more direct about how these are pseudovelocities and not calculated in the conventional way. The question of how many sequential regulatory states exist in this lineage and whether they are 'stable' or not is an interesting one. The text should still perhaps be a bit more circumspect in places about how ad hoc clusters of transcriptional profiles may not map perfectly to the underlying regulatory states. Fig2J for example arguably suggests that there are 3 distinct states, not 5.

2.4 I'll defer to Reviewer 2 but my sense is that a bit more could be done to explain and justify the denoising.

2.8 I think this an important point from reviewer 2 that it is indeed quite surprising to have so many hits in the CRISPR screen. I'd note that A) the CRISPR screen didn't randomly sample mature genes but instead focused on TFs and signal transduction molecules, and B) apart from DEPDC1, most of the phenotypes seem to be quite subtle. These details are more clear in the revised manuscript (assuming the stats etc in 5G get properly explained...)

small concerns:

line 79 typo: 'a providing...'

line 390 'Remarkably' seems like the wrong word here given that you could easily expect altered division orientation to cause the

observed cell fate defects. I'd recommend 'Notably' as an alternative.

line 549 S-phase marker? label?

line 599 I recommend making it clear (I think...) that you are talking about cell cycle phase transitions and not other sorts of phase transitions

Figure 2-S1 I believe the figure legends are misordered.

Referee #2:

The manuscript has improved a lot and will be impactful. A small number of issues still remain.

Major comments:

1. The availability of the authors' datasets via ShinyApp is appreciated. However, the current analysis of cell clusters remains rudimentary and lacks a thorough comparison with previously published Ciona embryo atlases. Given the existence of established references, it is essential to align cluster annotations with prior studies to ensure consistent interpretation across datasets. The authors can perform label transfer or similar comparative analyses. This would facilitate clearer integration of their findings within the broader context of Ciona cell atlas efforts, help avoid redundancy and highlight the novelty.
2. A key issue that remains unaddressed is the identification of transcription factors (TFs) corresponding to the enriched motifs. Because multiple TFs can recognize the same motif-and not all are necessarily expressed-it is important to evaluate both motif enrichment and TF expression levels. While SCENIC+ requires paired ATAC-seq data, the original SCENIC framework or scMTNI can be applied using transcriptomic data alone. Alternatively, the authors should plot the expression levels of the transcription factors and highlight the motif-expression concordant ones.

Minor comments:

1. Citation 46,49,51 and others have been officially published (not just biorxived) and please update it.
2. "FACS" is misspelled as "FAC" in three instances and should be corrected!! (such as Line 160)
3. The term "pseudogenes" in line 808 is used incorrectly. It conflicts with the established concept of pseudogenes. Consider using "spike-ins" or "artificial genes" instead.

Referee #1:

Comments in response to the rebuttal

1.1 'grandiose'.

I feel like the authors and I are maybe talking past one another on this issue. My point is just that it is not the *concept* of transcriptome maturation that is novel here. The novelty comes from how they've elegantly identified the actual details of one specific example of transcriptome maturation. I'm generally happy with the changes here. I'd recommend removing the quotation marks when discussing transcriptome maturation, regulatory competence, and behavioral competence, to help make it clear that you are defining your terms without the implication that these are newly minted concepts

Another modest but helpful change here would be to refine the discussion at lines 641-645 of the revised MS. It may not be the authors' intent, but to me this sentence seems to argue that developmental biologists have only addressed the concept of 'maturation' in the context of terminal differentiation and not earlier cell fate decisions. It's probably true that the word 'maturation' has mostly been used in the context of terminal differentiation, but the Melton review cited takes a very flexible view of maturation and it is certainly the case that developmental biologists routinely study transcriptional changes over time that are important for early branches in the Waddingtonian landscape.

We agree with the reviewer and we acknowledge that "developmental biologists routinely study transcriptional changes over time that are important for early branches in the Waddingtonian landscape", through our citations of the very first paragraph of the introduction with "Changes in the biomolecular composition of cells underlie these developmental transitions, and differential transcriptional activity governing transcriptome dynamics is an established driver of fateful molecular transitions during development (Levine & Tjian, 2003; Levine & Davidson, 2005).", and in the discussion with "Extending beyond ascidian embryogenesis, whole animal profiling of developmental trajectories during vertebrate embryogenesis has repeatedly uncovered similar patterns of multipotent progenitors seemingly "maturing" along linear trajectories before producing distinct cell identities, presumably following cell divisions (Qiu *et al*, 2022a; Farrell *et al*, 2018; Briggs *et al*, 2018). "

Of note, the branching in the Waddingtonian landscape is not part of the maturation, as it occurs after cell division, but the main point of our paper is that the ability to branch is what is acquired through multipotent progenitor maturation.

We specifically and explicitly apply the term maturation here because the developmental transcriptome changes happen WITHIN ONE interphase. It is the same cell, and the term maturation requires the entity to remain the same over time. So we are focusing on transcriptome-wide changes that occur in the same cell as it migrates and goes through interphase, but do not consider the gene expression changes that occur after cell division to be part of the maturation.

We are pleased to see that the reviewer acknowledges that "It's probably true that the word 'maturation' has mostly been used in the context of terminal differentiation". Regarding the cited review by Alvarez-Dominguez and Melton, their first figure places "maturation" as occurring AFTER commitment to a specific cell identity, and they quite clearly make it part of the differentiation program. Our main point is that such process as maturation, in "Classic concepts in developmental biology, dating from homo- and heterotopic and -chronic grafting experiments by gifted embryologists, suggested that progenitor cells transition through specification and commitment to restricted identities, while the notion of maturation has been considered extensively for differentiated cells, such as cardiomyocytes, which are contractile but not fully functional until they reach a mature state". Here, we adopted a general definition of the concept of maturation, considering a biological entity that persists through time and progressively acquires its full functionality through underlying changes. From that standpoint, since multipotent cardiopharyngeal

progenitors transition through successive regulatory states as they progress through interphase, and ultimately acquire the competence to divide in an oriented and asymmetric manner and produce distinct first heart vs. *Tbx1/10+* second craniopharyngeal progenitors, we argue their transcriptome maturation fosters multilineage competence."

We would like to respectfully point out that we address this as a discussion point, which is meant to cast a broader light on the topic, without making extraordinary claims about novelty or appropriation. Specifically, the sentence " Here, we adopted a general definition of the concept of maturation[insert Melton citation], considering a biological entity that persists through time and progressively acquires its full functionality through underlying changes." is entirely consistent with "general definition" being already adopted in other papers like the Melton one, which we thus cite there now. We'd be happy to include more references to similar maturation of multipotent progenitors within one interphase.

changes:

changes L 625 to which we refer to as maturation;

We removed quotation marks when referring to maturation/maturing/mature

1.2 'scRNAseq details'

I promise you I did read the methods section as well on the original submission... I'm still confused though by the details here.

The current methods text reads 'To avoid technical variation between samples, we prepared fertilized eggs for all samples in a large pool of dechorionated eggs, obtained from multiple animals, and electroporate the paired SBCs individually at one-hour intervals according to the experimental design.'

This doesn't make sense as written, because if you did indeed prepare a single large pool of fertilized and dechorionated eggs, then you wouldn't have been able to do the separate fertilizations needed to have all the timepoints ready for the same harvest time. I'm assuming you killed a bunch of animals at the start of the experiment and stored the pooled sperm and pooled eggs separately, then fertilized aliquots at intervals? That seems like a smart way to do it but it is not clear from how it is described in either the original or revised submissions... Did you dechorionate the whole pool of unfertilized eggs and then fertilize them in batches afterwards or did you do separate dechorionations for each fertilization? How were the gametes stored? Are there any concerns about potential biases from having unfertilized eggs sit around *ex vivo* for many hours? This just needs a bit more detail and clarity.

We thank the reviewer for catching this and clarified the relevant section into " To minimize variation between samples, we collected eggs for all samples into a large pool obtained from multiple animals, split the pool and proceeded to fertilize, dechorionate and electroporate the paired SBCs individually at one-hour intervals according to the experimental design."

1.3. sortability

The issue of some cell types being more sortable and recoverable than others in scRNAseq experiments is problematic and often swept under the table in scRNAseq papers. The manuscript is much improved here with the clear discussion of ATMs being less sortable than TVCs. I'm still a bit suspicious that there could be something more going on given how there

seems to be such a big difference in ATM sortability within a single cell cycle, but I'm happy with how it is treated within the scope of this manuscript.

A: Thank you

1.4

I'm satisfied with the changes made.

A: Thank you

1.5 Whole embryo dataset

I continue to feel that this is a very long paper and that the whole embryo experiments aren't critical to its key findings. It would be an obvious thing to remove if there is an editorial consensus that the paper should be shorter. It's interesting data though and I have no problem with it staying in as long as the journal is happy with the overall length and complexity.

A: Thank you

1.6 RNA velocity

I'm happy with the changes here. I still think there are some interesting opportunities to use the relatively high temporal resolution of this time course dataset to test some of the strengths/weaknesses of pseudotime inference more broadly, but that would be outside the scope of this MS.

A: Thank you

1.7 Candidate genes

The revised manuscript makes it clear that these are just a subset of the candidate genes and that they were selected manually based on having interesting predicted molecular functions. I'm satisfied with the changes.

A: Thank you

1.8 CRISPR miniscreen

This figure is much improved and far more interpretable with the key findings moved into the main text figure from the supplements. I'm generally happy with it. I remain concerned about Fig 5G, where my previous concerns have not been addressed in the rebuttal. From a data-viz perspective, I find it confusing to use a diverging color map for p-value data, at least not without having it centered on a threshold of interest like $-\log_{10}(0.05)$. It's also confusing to have different scales for the three assays.

Beyond that, I'm still having a hard time reconciling the $-\log_{10}$ (FDR) p-values shown in 5G with the p-values indicated with brackets in 5C-E and the Venn diagram in F. Why do so many more genes seem to have strong statistical support for a defect in Tbx1/10 expression in 5G vs 5E? Are these different statistical tests? This section would also benefit from a more detailed treatment in the methods of how the images were analyzed, how the orientation,

asymmetry and expression data were reduced to categorical phenotypes, and which statistics were used where.

A: We wish to point out that FDR values shown are not the same as statistical p-values. FDR reflects the rate of type I errors in the total number of incidences in which the null hypothesis is rejected. The less ambiguous nature of scoring presence or absence of easily observable spots of nascent *Tbx1/10* transcripts may lend itself to a clearer classification of types leading to a more robust certainty of classification (5G), however this should not be taken as indication of statistical significance of the total set of observations (5E). For the Venn diagram we only use the genes that we found to cause a statistically significant difference using Fisher's exact test.

We added the following to the methods:

"Fisher's exact test was used to detect significant changes in the three categories with an FDR with a cutoff of 0.05 to determine rate of Type I errors."

1.9

No concerns as currently presented.

A: Thank you

1.10

Much improved, though I note a few small things lower down.

A: Thank you

I'll also note a few things that resonated with me in response to Reviewer 2's comments:

2.1

I like the velocity analysis though I'd suggest that the text should be more direct about how these are pseudovelocities and not calculated in the conventional way.

These values were calculated by using reads that we could infer to be coming from spliced vs non-spliced transcripts. We did not include the pseudovelocity in this paper.

The question of how many sequential regulatory states exist in this lineage and whether they are 'stable' or not is an interesting one. The text should still perhaps be a bit more circumspect in places about how ad hoc clusters of transcriptional profiles may not map perfectly to the underlying regulatory states. Fig2J for example arguably suggests that there are 3 distinct states, not 5.

A: Yes, if we consider stability a necessary element of a state. Others (e.g. Morris and Martinez-Arias, whom we cite) have considered "transition states", which we agree is a bit of an oxymoron. It may be a bit semantic and up for discussion.

2.4

I'll defer to Reviewer 2 but my sense is that a bit more could be done to explain and justify the denoising.

A: We confess being a bit uncertain as to what is requested here, and could use guidance from the editor.

2.8

I think this an important point from reviewer 2 that it is indeed quite surprising to have so many hits in the CRISPR screen. I'd note that A) the CRISPR screen didn't randomly sample mature genes but instead focused on TFs and signal transduction molecules, and B) apart

from DEPDC1, most of the phenotypes seem to be quite subtle. These details are more clear in the revised manuscript (assuming the stats etc in 5G get properly explained...)

A: Thank you

small concerns:

line 79 typo: 'a providing...'

corrected, thank you

line 390 'Remarkably' seems like the wrong word here given that you could easily expect altered division orientation to cause the observed cell fate defects. I'd recommend 'Notably' as an alternative. changed

line 549 S-phase marker? Label? added, thank you

line 599 I recommend making it clear (I think...) that you are talking about cell cycle phase transitions and not other sorts of phase transitions. Clarified thank you

Figure 2-S1 I believe the figure legends are misordered. Corrected. Thank you

Referee #2:

The manuscript has improved a lot and will be impactful. A small number of issues still remain.

Major comments:

1. The availability of the authors' datasets via ShinyApp is appreciated. However, the current analysis of cell clusters remains rudimentary and lacks a thorough comparison with previously published Ciona embryo atlases. Given the existence of established references, it is essential to align cluster annotations with prior studies to ensure consistent interpretation across datasets. The authors can perform label transfer or similar comparative analyses. This would facilitate clearer integration of their findings within the broader context of Ciona cell atlas efforts, help avoid redundancy and highlight the novelty.

A: We appreciate the interest and agree that our whole embryo dataset is richer than presented in details here; however, here we focus on using it to (1) extend the proof of principle for the transgene-based barcoding and multiplexing and (2) extend the concept of multipotent progenitor maturation to other lineages. We wish to argue that these two points are supported by the analysis presented, and we made the dataset available, including through an interface that already facilitates its use by colleagues in the field.

A more complete and extensive analysis, while generally warranted, is beyond the scope of this study and we are preparing another manuscript more directly focused integrating this with previous and other datasets.

2. A key issue that remains unaddressed is the identification of transcription factors (TFs) corresponding to the enriched motifs. Because multiple TFs can recognize the same motif- and not all are necessarily expressed- it is important to evaluate both motif enrichment and TF expression levels. While SCENIC+ requires paired ATAC-seq data, the original SCENIC framework or scMTNI can be applied using transcriptomic data alone. Alternatively, the authors should plot the expression levels of the transcription factors and highlight the motif-expression concordant ones.

A: Regarding the latter point, we attempted to present such connections between TF gene expression, motifs and gene expression clusters extensively in new Figure 7 and the corresponding Appendix Figures S14 to S20, which we renamed after the preferred EMBO Journal nomenclature, and summarized in Figure 10.

Regarding the former point of GRN modeling using high-throughput dataset, we have experience from our collaboration with the group of Dr. Richard Bonneau, who developed the framework Inferrelator, and learned - sometimes the hard way - that such predictions, especially using only expression data, are notoriously unreliable and would require extensive post-hoc experimental validations. This would also extend well beyond the scope of this manuscript.

Importantly, while we agree that additional modelling would be great, we wish to argue that the absence of GRN modeling does not undermine the key conclusions of our paper, which are supported by simpler but reliable analyses.

Minor comments:

1. Citation 46,49,51 and others have been officially published (not just biorxived) and please update it. Updated, with Paperpile issues for Tjarnberg et al. Thank you
Thank you
2. "FACS" is misspelled as "FAC" in three instances and should be corrected!! (such as Line 160)
L. 160 spells "FAC-sorting", the S in FACS stands for sorting, so we wish not to write something that would mean "fluorescence activated cell sorting sorting", nor FACSing, but we'll follow the editor's guidance on this.
3. The term "pseudogenes" in line 808 is used incorrectly. It conflicts with the established concept of pseudogenes. Consider using "spike-ins" or "artificial genes" instead.
changed to artificial genes

[External email] Make sure you recognize the sender's email address before you click links, open attachments, or get involved in financial transactions. Contact IT-support BRITA if you have any questions.

Dear Lionel,

Thank you for submitting the final revised version and addressing the remaining editorial points. I am now pleased to inform you that your manuscript has been accepted for publication. Congratulations on a great study!

Before we forward your manuscript to our publishers, I would like to propose some edits in the manuscript abstract and synopsis (please also see the attached file). I am afraid that I had to significantly shorten the provided text, especially in the abstract, to comply with our style guidelines. I have also written a short blurb that will accompany the title of your manuscript in our online system. Please take a look and let me know if any corrections or adjustments are needed.

Blurb:

Transcriptome changes in the craniopharyngeal progenitors of the tunicate *Ciona intestinalis* enable asymmetric division and correct cell fate determinant expression in the progeny.

Synopsis

During development, multipotent cardiopharyngeal progenitors express both cardiac and pharyngeal muscle transcriptional programs. This study in the tunicate *Ciona* shows that cell cycle-regulated transcriptome maturation in multipotent cardiopharyngeal progenitors is necessary for asymmetric cell division and daughter cell fate establishment.

- Cell-cycle progression and mitosis promote transcriptome maturation in multipotent progenitors and de novo gene expression in fate-restricted precursors.
- Genes that peak in the late G2 mature state are required for asymmetric cell division and polarized expression of the cardiopharyngeal fate determinant *Tbx1/10*.
- In the whole embryo, endomesodermal fates show marked canalization by lineage and show signs of progenitor transcriptome maturation prior to cell fate branching.

If you have any questions, please do not hesitate to contact the Editorial Office or me directly. Thank you for your contribution to The EMBO Journal!

With best wishes,

Ieva

Ieva Gailite, PhD
Senior Scientific Editor
The EMBO Journal
Meyerohofstrasse 1
D-69117 Heidelberg
Tel: +4962218891309
i.gailite@embojournal.org
